# Progress and new challenges in image-based profiling

Erik Serrano[1], John Peters [2], Jesko Wagner [3], Rebecca E Graham[4], Zhenghao Chen[5], Brian Y Feng [5], Gisele Miranda[6], Alexandr A Kalinin [7], Loan Vulliard[8], Jenna Tomkinson [1], Cameron Mattson[1], Michael J Lippincott[1], Ziqi Kang [9], Divya Sitani [10], Dave Bunten[1], Srijit Seal [7], Neil O Carragher [11], Anne E Carpenter [7], Shantanu Singh [7], Paula A Marin Zapata[12], Juan C Caicedo [2] & Gregory P Way [1]✉

## Abstract

For over two decades, image-based profiling has revolutionized cell phenotype analysis. Image-based profiling processes rich, high-throughput, microscopy data into thousands of unbiased measurements that reveal phenotypic patterns powerful for drug discovery, functional genomics, and cell state classification. Here, we review the evolving computational landscape of image-based profiling, detailing the bioinformatics processes involved from feature extraction to normalization and batch correction. We discuss how deep learning has fundamentally reshaped the field. We examine key methodological advancements, such as single-cell analysis, the development of robust similarity metrics, and the expansion into new modalities like optical pooled screening, temporal imaging, and 3D organoid profiling. We also highlight the growth of public benchmarks and open-source software ecosystems as a key driver for fostering reproducibility and collaboration. Despite these advances, the field still faces substantial challenges, particularly in developing methods for emerging temporal and 3D data modalities, establishing robust quality control standards and workflows, and interpreting the processed features. By focusing on the technical evolution of image-based profiling rather than the wide-ranging biological applications, our aim with this review is to provide researchers with a roadmap for navigating the progress and new challenges in this rapidly advancing domain.

**Keywords** Image-Based Profiling; Cell Profiling; Phenotypic Screening; Deep Learning; Feature Extraction
**Subject Category** Methods & Resources

## Introduction

Image-based profiling is a powerful and scalable approach for characterizing cell phenotypes by extracting and processing high-content readouts from microscopy images of cells (Way et al, 2023). By converting cell microscopy image data into high-dimensional numerical representations, image-based profiling enables the systematic analysis of cell responses to genetic or chemical perturbations at single-cell resolution (Fig. 1) (Caicedo et al, 2017; Scheeder et al, 2018). This approach has become increasingly central to a wide range of biomedical applications, including phenotypic drug discovery, mechanism of action (MoA) prediction, functional genomics, and disease modeling (Seal et al, 2024a; Krentzel et al, 2025; Chandrasekaran et al, 2020). In phenotypic drug discovery, image-based profiles can be used to group compounds by MoAs, detect off-target effects, and identify subtle phenotypic changes that might be overlooked by molecular assays (Berg, 2021). This allows exploration of compound bioactivity in a target-agnostic manner, uncovering new therapeutic candidates and repurposing opportunities (Seal et al, 2024a; Way et al, 2023; Scheeder et al, 2018). In functional genomics, image-based profiling quantifies the phenotypic impact of genetic perturbations (e.g., CRISPR knock-outs or RNAi), helping map gene function and pathway relationships at single-cell resolution (Caicedo et al, 2016; Ramezani et al, 2025; Way et al, 2021). Furthermore, in disease modeling, morphological features derived from patient-derived cells can be used to distinguish disease subtypes, identify biomarkers, elucidate disease mechanisms or screen for phenotype-correcting compounds (Caicedo et al, 2022; Betge et al, 2022; Schiff et al, 2022; Travers et al, 2025; German et al, 2021).

The landmark review by Caicedo et al, (2017) was collaboratively written by members of the then newly-founded CytoData Society, dedicated to the informatics of image-based profiling. This review established the foundational standards for image-based profiling methods, outlining critical steps such as illumination

[1]Department of Biomedical Informatics, University of Colorado Anschutz, Aurora, CO, USA. [2]Morgridge Institute for Research, University of Wisconsin-Madison, Madison, WI, USA. [3]MRC Human Genetics Unit, Institute of Genetics and Cancer, University of Edinburgh, Edinburgh, United Kingdom. [4]Centre for Clinical Brain Sciences, University of Edinburgh, Edinburgh, United Kingdom. [5]Calico Life Sciences, South San Francisco, CA, USA. [6]Department of Computational Science and Technology, Science for Life Laboratory, KTH Royal Institute of Technology, Stockholm, Sweden. [7]Imaging Platform, Broad Institute of MIT and Harvard, Cambridge, MA, USA. [8]Systems Immunology and Single-Cell Biology, German Cancer Research Center (DKFZ), Heidelberg, Germany. [9]Research Program in Systems Oncology, University of Helsinki, Helsinki, Finland. [10]Department of Systems Medicine, German Center for Neurodegenerative Diseases (DZNE), Bonn, Germany. [11]Cancer Research UK Scotland Centre, Institute of Genetics and Cancer, University of Edinburgh, Edinburgh, United Kingdom. [12]Bayer AG, Berlin, Germany. ✉E-mail: gregory.way@cuanschutz.edu

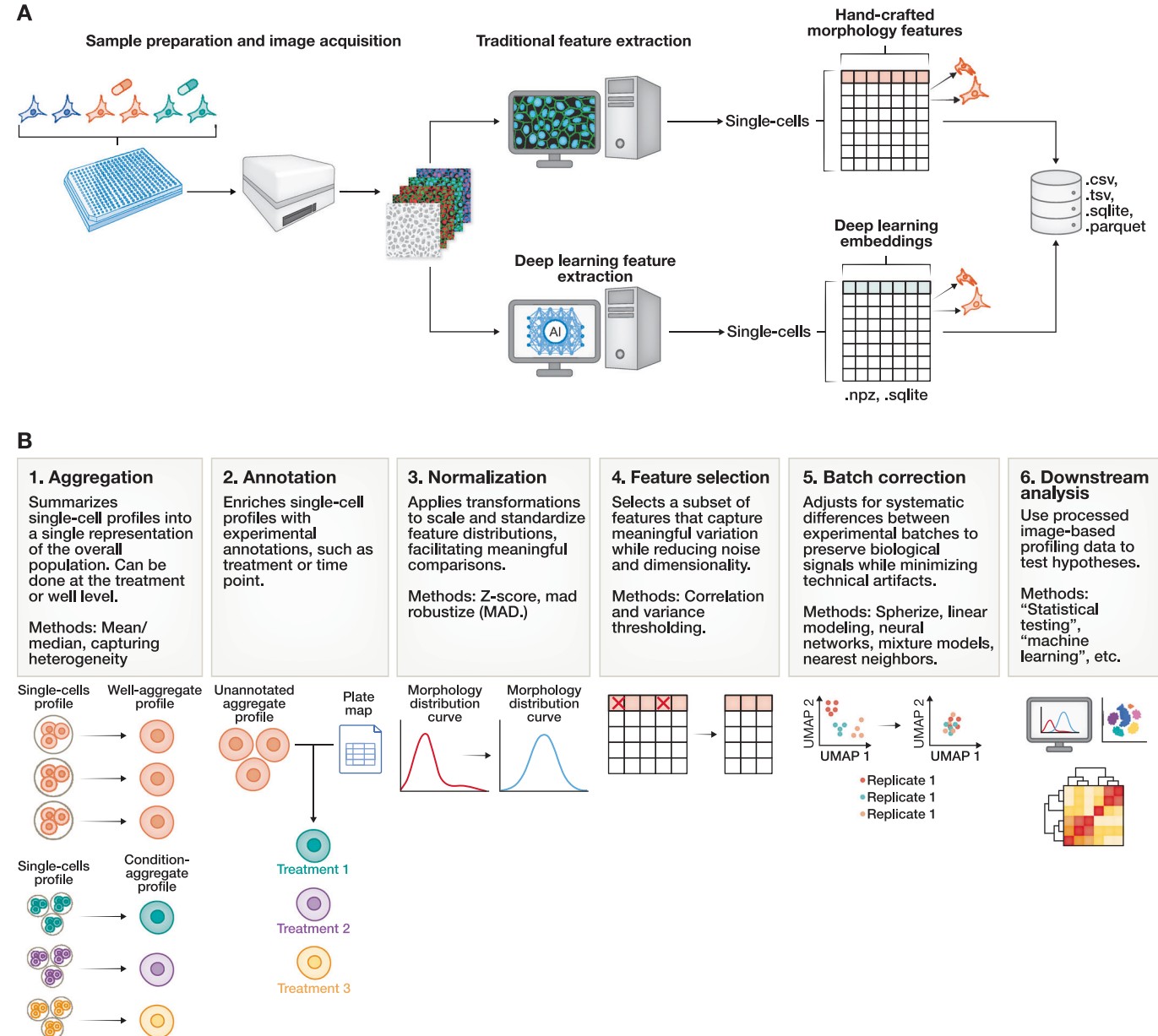

**Figure 1. The comprehensive image-based profiling workflow.**

A complete image-based profiling workflow spans from (**A**) experimental design, sample preparations, and image acquisition to the generation of digitized cellular representations and (**B**) their downstream processing steps.

correction, cell segmentation, feature quantification, feature processing, and profile generation. Since then, the field has advanced rapidly, driven by progress in both experimental protocols such as live-cell high-content assays (Moraes-Lacerda et al, 2025; Garcia-Fossa et al, 2025; Cottet et al, 2023), 3D spheroids and organoid models (Lukonin et al, 2021; Bian et al, 2021; Ringers et al, 2025), complementary profiling modalities (Dagher et al, 2025), and rapidly advancing computational tools, including the rise of deep learning, data integration methods, and reproducible software (Box 1).

In this review, we provide a comprehensive overview of the recent developments as well as current and emerging challenges that have

shaped the field of image-based profiling over the past decade. We begin with an overview of the image-based profiling pipeline. We highlight the importance of experimental design (i.e., including the use of biological replicates, thoughtful plate layout, etc.) and discuss widely adopted assays such as Cell Painting. We then transition to discussing the computational core of image-based profiling, detailing approaches for feature extraction from microscopy images using both traditional computer vision methods and more recent deep learning-based techniques. Throughout, we emphasize how pipeline innovations have significantly expanded the capabilities of image-based profiling while also introducing new technical and conceptual challenges. We also discuss emerging methods that analyze image-based profiling

**Box 1:  Highlighting recent progress in image-based profiling**

*Recent progress in image-based profiling. This box outlines three key advances in image-based profiling methods. We elaborate on all of these methods and more in the remainder of the review*

1. **Powerful deep learning models now enable automatic extraction of biologically meaningful features from raw image data.** Nevertheless, recent studies continue to show similar performance of traditional, hand-crafted features. For example, (A) the count of cells or mitochondria outperformed CNNs to predict assay outcomes (Seal et al, 2025); (B) Mechanism of action (MoA) prediction using K-nearest neighbors on CellProfiler features (F1 = 0.41) was comparable to DeepProfiler (F1 = 0.42); and (C) Mean average precision (mAP) increased 29% when classifying chemical perturbations using a self-supervised DINO framework compared to hand-crafted CellProfiler features (Kim et al, 2025). Still, more often, deep-learned features outperform classical ones. For example, zero-shot image-to-image MoA classification using contrastive learning (multimodal) CLOOME achieved 61.3% in top-10 accuracy compared to 24.5% with CellProfiler. Architectures such as convolutional neural networks (CNNs) (Alex Krizhevsky et al, 2017) and vision transformers (ViTs) (Dosovitskiy et al, 2020) capture both fine-grained and higher-order morphological patterns (Krentzel et al, 2025). These models, trained using weakly or self-supervised learning (SSL), scale effectively to large unlabeled datasets (Chai et al, 2024).

2. **Integrating with omics technologies, image-based profiling can enhance biological interpretation by linking molecular mechanisms to phenotypes.** This linkage is especially relevant in the context of high-throughput screens (Watson et al, 2022; Lim et al, 2024). Tools such as MorphLink (Huang et al, 2025), iIMPACT (Jiang et al, 2024), Phenonaut (Shave et al, 2023), and CellDART (Bae et al, 2022) exemplify this growing ecosystem of multimodal analysis platforms. While transcriptomics, proteomics, and metabolomics provide insight into molecular composition and regulation, image-based profiling captures phenotypic manifestations of these molecular processes (Schneider et al, 2022; Krentzel et al, 2025). In addition, complementary profiling modalities, such as the high-plex secretome profiling,

have been shown to enhance mechanistic insights when combined with image-based assays like Cell Painting (Dagher et al, 2025). Therefore, by combining molecular with phenotypic readouts, image-based profiling is revealing more comprehensive portraits of cell state, drug MoA, and disease.

3. **The rise of robust image-based profiling software and datasets is evidence of its growing utility and impact across academia and industry.** This emerging ecosystem now supports every step in the image-based profiling informatics pipeline, such as image segmentation, feature extraction, data processing, and batch effect correction. Open-source software tools such as CellProfiler (Stirling et al, 2021), BioProfiling (Vulliard et al, 2022), Pycytominer (Serrano et al, 2025), DeepProfiler (Moshkov et al, 2024), and others provide researchers with free-to-use, flexible platforms for image-based profiling. Commercial systems, such as HC StratoMineR (Omta et al, 2016), have complemented open-source strategies by widening industry accessibility to image-based profiling. Large-scale data initiatives have also accelerated software development by providing a test-bed suitable for real-world applications. For example, the JUMP Cell Painting Consortium (Chandrasekaran et al, 2023), released a large, publicly accessible image-based profiling dataset, and the Cell Painting Gallery (Weisbart et al, 2024), stores publicly accessible Cell Painting images alongside raw and processed image-based profiles (Weisbart et al, 2024). The availability of these datasets has driven rapid development in a wide range of methodological research, including new computational tools such as scmorph (Wagner et al, 2025), PhenoProfiler (Li et al, 2025), CytoSummaryNet (van Dijk et al, 2024), and SPACe (Stossi et al, 2024), as well as extensive benchmarking of existing algorithms (Arevalo et al, 2024; Li et al, 2025; Yan et al, 2025; Kim et al, 2025; Tian et al, 2023; Shpigler et al, 2025). This open science model has not only democratized access to high-content image analysis but has also fostered community standards, reproducibility, and rapid innovation in the field.

readouts, including the creation of similarity metrics and evaluation frameworks designed to better interpret and compare profiles. This review does not focus on specific biological applications, which have been discussed extensively elsewhere (Seal et al, 2024b; Tang et al, 2024; Chandrasekaran et al, 2020; Walton et al, 2022). Instead, our goal is to examine the methodological and computational infrastructure that underpins modern image-based profiling. We aim to offer a broad yet detailed perspective on the current state of the field, the innovations propelling it forward, and the challenges that remain on the horizon.

# From experimental design to bioinformatics processing: a comprehensive workflow for image-based profiling

## Experimental design and execution for image-based profiling

Image-based profiling begins with carefully designed experiments aimed at generating high-quality, reproducible microscopy data that accurately capture cell phenotypes (Fig. 1A). The experimental workflow starts with sample preparation, where the inclusion of experimental controls and biological replicates is essential to

distinguish true phenotypic effects from technical variation. Negative controls, such as untreated, vehicle-treated (e.g., DMSO), or non-targetting scrambled CRISPR guides are required to establish baselines for normalization. Most image-based profiling experiments test a range of perturbations (e.g., chemical compounds and gene knockouts) across multiple replicates. Replicates are essential for measuring both the magnitude and consistency of phenotypic effects. Biological replicates capture natural variability and improve statistical power, while technical replicates help account for intra-plate noise like spatial effects (Way et al, 2022a; Birmingham et al, 2009). Reliable estimation of negative control distributions (i.e., mean and standard deviation) depends on having a sufficient number of replicates. Plate layout also constrains replicate number; for example, in a standard 96-well plate using the inner 60 wells, six replicates per treatment with flanking positive controls is common. Higher-density formats, like 384-well plates, allow for even more replicates. We recommend including as many replicates as feasible, which is typically a minimum of four for treatments and at least nine for controls (Cimini et al, 2023).

Maintaining consistent seeding density and media conditions is equally critical, as these variables influence cell morphology, health, and growth uniformity. Inconsistent conditions can introduce artifacts like overcrowding or edge effects (Auld et al, 2020). A few

experimental tricks can reduce differences in cell growth along outer rows and columns ("edge effects" or "plate layout effects") (Lundholt et al, 2003; Tanner, 2017). Plate format selection (e.g., 96, 384-, or 1536-well plates) and strategic control/treatment distribution can greatly impact data quality and reproducibility (Caicedo et al, 2017; Auld et al, 2020; Cimini et al, 2023). For example, high-well-count plates (e.g., 1536-well) minimize data loss from discarding edge wells affected by edge effects, compared to lower-density formats (e.g., 96-well). Great strides in experimental design are also being made by incorporating multiple cell lines and including reference compounds (Elliott et al, 2024). These approaches enhance biological mechanistic understanding and broaden the applicability of image-based profiling, equally benefiting single-cell image-based profiling (Boyd et al, 2019; Dahlin et al, 2023).

Following plate preparation, automated high-throughput imaging systems perform image acquisition. Fluorescence microscopy is predominant, particularly in the Cell Painting assay, which stains multiple subcellular compartments across five to six channels to provide a rich and unbiased representation of cellular morphology (Seal et al, 2024b; Bray et al, 2016). We discuss alternative image-based profiling assays in the section "Emerging focuses and challenges in image-based profiling". Additionally, label-free microscopy techniques, including brightfield or phase-contrast imaging, are increasingly explored for their non-invasive nature and compatibility with live-cell experiments (Cross-Zamirski et al, 2022; Vicar et al, 2019).

After images are acquired and features extracted, quality control becomes an important final step in the experimental workflow, ensuring that the earlier design choices have effectively minimized technical noise. Visualization techniques are right for this assessment. Heatmaps of sample-level correlations can reveal batch effects, where blocks of high similarity correspond to specific plates or experimental dates rather than biological treatments (Caicedo et al, 2017). At a finer resolution, UMAPs of single-cell data, colored by metadata such as plate ID or well position, can highlight "islands" of cells artificially separated by technical variables or experimental details rather than biological treatments (Caicedo et al, 2017; Arevalo et al, 2024). To assess the significance of batch effects, matched samples can be compared across batches, plate rows and columns using the mean average precision framework by Kalinin et al, 2025. Detecting these sources of variation early is critical, as it guides normalization and batch correction strategies and ensures that downstream analyses yield robust and biologically meaningful results.

## Transforming microscopy images into high-dimensional image-based profiles

Following image acquisition, the image-based profiling workflow transitions to a computational focus. The workflow transforms raw microscopy images into quantitative, high-dimensional representations of cell morphology. The canonical multi-step workflow, thoroughly reviewed in (Caicedo et al, 2017), provides a foundational framework for this process. In the years since, while the fundamental steps remain similar, substantial advancements, particularly driven by deep learning, have modernized each component of this pipeline (Way et al, 2023; Driscoll and Zaritsky, 2021). We focus our discussion on these advances, and, while we

discuss each step, we primarily focus on the downstream image-based profiling rather than upstream steps (e.g., segmentation). It is also important to note that many deep learning approaches may skip some of these upstream steps to process embeddings directly from raw microscopy images (Moshkov et al, 2024; Donovan-Maiye et al, 2022; Zhou et al, 2024).

### Whole-image quality control
Quality control (QC) at the whole image level removes images that are out of focus or that contain large smudges/debris. These poor-quality images decrease segmentation quality and introduce noise into downstream image-based profiling readouts. CellProfiler offers automated QC through its "MeasureImageQuality" module, which computes intensity-based metrics that can reveal common artifacts such as blurring, oversaturation, or uneven illumination (Bray and Carpenter, 2018). Alternatively, more systematic and scalable approaches leverage tools like CellProfiler Analyst, which can use these metrics to train machine learning classifiers that distinguish high-quality images from those with defects (Dao et al, 2016). Although using machine learning classifiers requires additional annotation and modeling effort, it enables more robust and automated QC across large datasets.

### Illumination correction
Illumination correction is typically the next step. Illumination anomalies exist across the images' field of view, due to uneven background lighting and signal variations from optical artifacts. Correction involves learning the pattern and correcting it so that downstream measurements reflect biological differences rather than technical noise (Singh et al, 2014; Caicedo et al, 2017). Notably, however, many modern deep learning pipelines often bypass this explicit correction step, as models can learn to be robust to such illumination variations during training, representing a significant leap (Wang et al, 2021; Guo et al, 2025; Wang et al, 2023b; Li et al, 2022; Rai et al, 2022).

### Cell segmentation
Segmentation is a critical step in most image-based profiling workflows. Classical approaches, implemented in tools like CellProfiler (Stirling et al, 2021) and Ilastik (Berg et al, 2019), use image analysis techniques such as intensity thresholding and watershed algorithms to delineate individual cells from the background (Wang et al, 2024b). However, the field is witnessing a shift to deep learning, a transition catalyzed by foundational architectures like U-Net, which uses convolutional neural networks (CNNs) to achieve superior segmentation accuracy (Ronneberger et al, 2015; O'Shea and Nash, 2015; Chen and Murphy, 2023; Schmidt et al, 2018). This advancement paved the way for cell segmentation tools such as StarDist (Schmidt et al, 2018; Weigert and Schmidt, 2022), and Cellpose (Stringer et al, 2020) that offer models capable of identifying cell borders with minimal parameter tuning. Building on the success in segmentation, deep learning models have been extended to tackle the more complex challenge of cell tracking, often by performing both tasks simultaneously in end-to-end frameworks (Chen et al, 2021a; Schwartz et al, 2019). More recently, the advent of foundation models has introduced transformer-based architectures to segmentation. Tools like CellSAM (Marks et al, 2025) and CellViT (Hörst et al, 2024, 2026) leverage these models to further improve performance

and generalizability across diverse imaging conditions. Some of these powerful architectures are being applied to end-to-end models that perform both segmentation and tracking of cells over time (O'Connor and Dunlop, 2025). Despite significant performance improvements over the last two decades, challenges remain, especially in generalizing models to new data types and imaging modalities. Accurately segmenting cells with low signal-to-noise ratios, complex morphologies, or high cell densities, especially within large 3D datasets, continues to be a difficult task within the field (Maška et al, 2023; Edlund et al, 2021).

### Feature extraction

In classical workflows, hundreds to thousands of hand-crafted features are extracted from segmented objects, quantifying aspects like size, shape, texture, intensity, and spatial relationships (Caicedo et al, 2017). However, this does not always translate to clear biological interpretability. Nevertheless, these features provide a mathematically well-defined representation of cell morphology, reflecting certain single-cell characteristics. These classical approaches have been instrumental in establishing image-based profiling as a scalable strategy (Forsgren et al, 2024). Their mathematical transparency allows for a clear understanding of feature measurement, supporting hypothesis-driven research (Driscoll and Zaritsky, 2021). However, the feature set might miss particular phenotypic changes, and can be prone to redundancy and noise. Deep learning (DL) methods have recently introduced a shift in image-based profiling feature extraction (Table 1). Unlike traditional approaches relying on pre-defined features, DL models learn relevant morphological representations directly from raw pixel data; the methods often inherently produce features that are less redundant. These learned representations, often referred to as "embeddings", are extracted via architectures like convolutional neural networks (CNNs) or vision transformers (ViTs), frequently outperform classical features in downstream tasks, a topic we explore further in section three.

### Aggregation and annotation

Aggregating and annotating profiles with associated metadata is a critical step after feature extraction (Fig. 1B). First, single-cell image-based profiles are typically aggregated to generate a population-level representation at the level of a well, field of view (FOV), or treatment (Caicedo et al, 2017; Arevalo et al, 2024). Aggregation is most commonly performed using the median across all single-cell features, generating a single representative vector that summarizes the morphological profile of a cell population under a given condition. Additionally, alternative aggregation strategies extend beyond medians, incorporating distribution-based descriptors, cluster-derived subpopulation proportions, and learned aggregation functions from deep representation models (Doron et al, 2023; Frey et al, 2025; van Dijk et al, 2024; Yao et al, 2024; Hu et al, 2025; Pearson et al, 2022; Garcia-Fossa et al, 2023; Rohban et al, 2019; Yao et al. 2024). Aggregation is useful for capturing general trends across populations that may exhibit substantial heterogeneity or contain outliers. This facilitates comparison across multiple experimental conditions (Rezvani et al, 2022). In the annotation step, metadata is merged with the associated aggregated profiles, which provides contextual information, enabling meaningful downstream interpretation and other analyses (Way et al, 2023). Key metadata include screen layouts, plate maps, sample

types, treatment details (e.g., siRNA/shRNA IDs, gene targets, CRISPR guides, compounds), control types, replicate counts, concentrations and time points, which are particularly vital for temporal studies (Cimini et al, 2023; Caicedo et al, 2017; Garcia-Fossa et al, 2025; Graham et al, 2025). Importantly, efforts toward maintaining metadata annotation ensure datasets are well-documented, shareable, and integratable across studies, thereby promoting reproducibility and long-term utility under Findable, Accessible, Interoperable, and Reusable (FAIR) principles (Wilkinson et al, 2016; Way et al, 2023; Williams et al, 2017). Metadata is essential for downstream processes, including normalization, batch correction, and quality control filtering (Caicedo et al, 2017). Despite its importance, metadata annotation faces challenges such as incomplete records, manual errors, software incompatibilities, and a lack of standardization, which risks loss of context when comparing across experiments.

### Normalization

Normalization is the next fundamental process in image-based profiling experiments, ensuring measurements from different plates, runs/batches, or instruments are comparable and reliable (Caicedo et al, 2017; Arevalo et al, 2024; Pylvänäinen et al, 2025). Normalization also adjusts raw feature values to achieve consistent statistical properties, which are required for downstream processing. Below, we distinguish between two types of normalization procedures: plate-level normalization and batch correction.

Plate level normalization.    Two common plate-level normalization strategies are whole-plate and control-based normalization. Both strategies normalize all samples only present within a single experimental plate. In whole-plate normalization, normalization is performed based on the statistics of the full plate. This method is computationally simple and effectively corrects for subtle, plate-specific artifacts. However, if the plate contains non-representative or homogeneous samples (e.g., all wells having similar phenotypes), this method may obscure biological signals. Control-based normalization leverages stable reference samples, such as untreated controls, to anchor normalization across different plates and batches, enhancing comparability by mitigating technical variation introduced between experimental runs (Risso et al, 2014). However, its effectiveness relies on having a sufficient number of control wells per plate to ensure stable and unbiased estimates (Seal et al, 2024b).

Z-score scaling, or standardizing, is a typical approach that centers each feature by subtracting its mean and dividing by its standard deviation (across the whole experiment, or a subset, such as a plate or batch), resulting in a scale of zero mean with unit variance (Caicedo et al, 2017). However, we recommend using median absolute deviation (MAD) normalization, as it accounts for outliers and heavy skew, providing a more robust representation of the underlying morphological feature distribution (Chen et al, 2021b). Quantile normalization enforces a common feature distribution across samples and can be effective when global properties are expected to be similar (Zhao et al, 2020; Caicedo et al, 2017). However, it should be applied cautiously, as it may obscure genuine biological variation by removing meaningful global shifts. Additional transformations, such as logarithmic or Box-Cox transformations, can address high dynamic ranges or skewed distributions by compressing large values and stabilizing variance, facilitating downstream statistical tests or machine-learning methods (Durbin et al, 2002;

**Table 1. Comparing feature extraction approaches: Classical features vs. deep learning representations.**

| Feature attribute | Classical approach (e.g., CellProfiler) | Deep learning approach (e.g., CNNs, ViTs) |
|---|---|---|
| Feature type | Relies on "hand-crafted" or "manually engineered" cell morphological descriptors. These features, such as size, shape, and texture, are predefined and calculated using classical, deterministic algorithms. | Enables automatic extraction of biologically relevant embeddings learned directly from raw image data. |
| Interpretability | This mathematical approach ensures each feature is well-defined, enabling a clear understanding of how features are measured. Still, this does not always translate directly to biological meaning. | Embeddings are challenging to interpret as they are typically anonymous latent variables without explicit names or meanings. Emerging methodologies attempt to enable an understanding of what each embedding captures in images (which itself may be a challenge to translate directly to biological meaning). |
| Feature redundancy | Features are often highly correlated, leading to significant redundancy (e.g., multiple features measuring different aspects of "size"). This necessitates a feature selection step to remove redundant information. | Embeddings are typically uncorrelated or have low redundancy. This happens when models are optimized to learn a compact representation where each feature captures an independent axis of variation, making feature selection less important. |
| Dependence on segmentation | Requires an explicit cell segmentation step to delineate individual cells before feature extraction can occur. | Can bypass cell segmentation entirely to compute image embeddings from patches across the image (or centered on each cell, after object detection). |
| Post-processing | Many features are typically highly correlated and thus require feature selection. | Requires much less tuning. Once gathered, learned embeddings are typically just normalized and batch corrected, with feature selection not generally being applicable. |
| Performance | These approaches were instrumental in establishing image-based profiling as a scalable and interpretable strategy. In a few instances, they perform equal to or better than deep learning strategies (Wong et al, 2023; Doron et al, 2023). | Learned embeddings usually outperform classical features in downstream tasks Li et al, 2025; Moshkov et al, 2024; Kim et al, 2025, presumably because they can capture nuanced and unbiased descriptors that manual features may miss, although potentially also because they can be taught to mitigate batch effects and other technical artifacts Li et al, 2025; Tang et al, 2024 |

This table compares manually engineered cell morphology-based features with learned deep neural network embeddings across key attributes, including feature type, interpretability, redundancy, dependence on segmentation, post-processing requirements, and downstream performance.

Huber et al, 2002). Handling zeros or negative numbers, such as by adding a small constant before applying a log transform, is also important. Ultimately, the choice of normalization and transformation methods should be guided by dataset characteristics and biological context, balancing comparability across samples with the preservation of meaningful variation (Pylvänäinen et al, 2025).

Batch correction. Batch effect correction is a specific kind of normalization that minimizes variation from technical inconsistencies among experimental batches, such as fluctuations in instrument calibration, reagent quality, or experimental handling (e.g., plate position). Subtle differences across experimental runs or between labs can introduce confounding signals that obscure biological variation (Luecken et al, 2022; Arevalo et al, 2024), making robust batch correction critical for revealing genuine biological patterns.

Current techniques for detecting technical artifacts include correlation heatmaps, where block-like patterns may indicate plate effects (i.e., spatial biases across wells), and principal component analysis (PCA), where separation along components may reflect inter-plate variation or batch membership rather than biological signal (Arevalo et al, 2024). Additionally, intra-plate or intra-batch normalization methods, such as applying $z$-score scaling within each batch, can mitigate such effects (Arevalo et al, 2024). More sophisticated methods such as Spherizing (Kessy et al, 2018), ComBat (Johnson et al, 2007), canonical correlation analysis (CCA) (Hotelling, 1936), and deep learning (Lin and Lu, 2022) are used for large datasets, offering flexibility in modeling and correcting batch variation.

ComBat employs a Bayesian framework to model batch effects as additive and multiplicative noise, while preserving much of the intrinsic variability in biological readouts. Initially designed for microarray gene expression data, it has been adopted in both single-cell genomics and image-based profiling contexts. Another popular algorithm, Harmony, iteratively performs a mixture-modeling approach that maximizes dataset diversity to learn and apply a cell-specific linear correction, removing batch effects while preserving biological distinctions (Korsunsky et al, 2019). Alternatively, some methods directly transform the data instead of modeling it. For example, Sphering computes a whitening transformation, which uses negative control replicate wells that are assumed to represent purely technical variability to apply this transformation across the dataset (Michael Ando et al, 2017). Related PCA/SVD-based approaches such as Typical Variation Normalization (TVN) operate similarly. TVN first learns the principal components generated from control samples to capture typical, batch-driven variation, then applies a sphering transformation in that space to reweight and decorrelate the data (Wang et al, 2024a; Kim et al, 2025).

Regardless of the chosen method, post-correction reassessment is vital to confirm the preservation of true biological signals and minimization of technical noise. However, the increasing scale and complexity of modern datasets have pushed the field to adapt more sophisticated methods from other domains, as simple plate-level normalization is often insufficient. A recent study evaluated the effectiveness of ten batch correction techniques from scRNA-seq on image-based cell profiling data by analyzing the JUMP Cell Painting dataset (Arevalo et al, 2024). The authors concluded that none of the evaluated methods could adequately remove batch effects in the most complex cases involving multiple microscope types and a large number of compounds, underscoring the ongoing challenge of robust batch correction and the need for new methods to be developed.

### Feature selection

Next, feature selection aims to retain features that capture the most meaningful biological variation. Image-based profiling typically generates hundreds to thousands of features per well (or single cell), describing diverse measurements of size, shape, texture, intensity, and spatial organization (Arevalo et al, 2024). However, not all features are equally informative; especially with classical image processing, as many may be redundant, noisy, weakly informative, or influenced by technical artifacts (Gopalakrishnan et al, 2024; Moshkov et al, 2024; Rezvani et al, 2022). Feature selection addresses this by systematically identifying and retaining a subset of features that best represent the biological signal, thereby improving interpretability, reducing computational burden, and enhancing the robustness of downstream analyses (Chandrasekaran et al, 2020; Caicedo et al, 2017). Common approaches include low-variance filtering to remove invariant features, correlation-based filtering to reduce redundancy among highly correlated features, and statistical selection methods identifying features significantly associated with experimental conditions (e.g., via $t$-tests, ANOVA, etc.) (Caicedo et al, 2017). Another practical approach is to select features based on their technical reproducibility, retaining only those that show high correlation across biological replicates under replicate conditions (Christoforow et al, 2019; Schneidewind et al, 2019; Akbarzadeh et al, 2022). Machine learning approaches, such as random forests, recursive feature elimination, LASSO, or elastic net regression, can rank or prioritize features based on predictive importance (Chandrasekaran et al, 2020). However, using machine learning models can be computationally intensive, and the selected features are often specific to the dataset trained on, and thus not generalizing well to new data or tasks (Way et al, 2023; Seal et al, 2024b). Furthermore, some analytical approaches do not require feature selection (e.g., per-feature linear modeling), as they assess each feature independently to identify significant associations, thereby bypassing the need for feature reduction (Kelley et al, 2023). Despite these advances, feature selection still remains challenging, where overly aggressive filtering risks discarding subtle but biologically relevant signals, while insufficient filtering may retain noisy or redundant features that dilute true biological variation and increase overfitting risk (Teschendorff, 2019; Riley, 2019). Furthermore, achieving stable, reproducible feature selection across different batches of data, replicates, or datasets remains a persistent difficulty, limiting generalizability (Dong et al, 2022; Tian et al, 2019). Deep learning approaches have emerged as an alternative, offering the ability to learn rich, high-level feature representations directly from image data (Moshkov et al, 2024; Lu et al, 2019; Tang et al, 2024; Liu et al, 2024b). These methods show promise in addressing challenges of redundancy, noise, and reproducibility by learning more compact and informative embeddings. While deep learning does not eliminate the need for thoughtful evaluation, it represents a paradigm shift in how feature representation and selection are approached in image-based profiling. This move toward learning compact representations directly from pixels, which often bypasses the need for explicit post hoc feature selection, marks one of the most significant evolutions from the classical, multi-step workflows described by (Caicedo et al, 2017; Tang et al, 2024; Chandrasekaran et al, 2020). We will explore the emergence of deep learning-based feature extraction and its transformative impact in greater depth in the next section.

# Advances in deep learning for image-based profiling

Traditionally reliant on manually-engineered morphological descriptors, image-based profiling is undergoing a paradigm shift toward automated, data-driven feature extraction powered by deep learning. Deep learning approaches, which started as proofs-of-concept, are now a practical and increasingly central tool for image-based profiling (Scheeder et al, 2018). One of the earliest and most widespread applications of deep learning has been cell segmentation, where neural networks replace classical image-processing algorithms to achieve greater accuracy, robustness, and scalability across diverse imaging modalities and experimental setups (see section "Cell segmentation") (Ronneberger et al, 2015; Chen et al, 2025; He et al, 2017; Schmidt et al, 2018). Beyond segmentation, deep learning also produces features directly from images (whether segmented or not), promising more comprehensive, nuanced, and unbiased descriptors of cell structure, state, and subtle phenotypic variations that manual features may miss (Seal et al, 2024a; Doron et al, 2023; Pfaendler et al, 2023). This transition reflects advances in the broader field of representation learning, which is the process of learning high-dimensional, informative features directly from raw data without human-defined attributes. These advances are reshaping image-based profiling pipelines, enabling scalable, reproducible, and biologically meaningful analyses. The following sections explore these specific methods in greater depth (Fig. 2).

## Deep learning architectures for image-based profiling

Neural networks are complex models characterized by architectures that comprise millions of parameters, which are randomly initialized before training begins. Through successful training with appropriate data and effective learning strategies, the architectures evolve into functional models. A model encompasses both the network's computational structure and its learned parameters, specifically tailored to address the problem for which it was trained.

There are two major neural network architectural families evaluated in image-based profiling: convolutional neural networks (CNNs) (Alex Krizhevsky et al, 2017) and vision transformers (ViTs) (Dosovitskiy et al, 2020) (Fig. 2A, B). The building blocks of the CNN are convolutional filters, which are learned operators that perform localized convolutional transformations to extract features in a hierarchical manner. The combination of filter sizes and pooling operations across multiple layers enables the projection of the image into a feature representation that can be used for downstream tasks. By using backbone designs such as ResNets (He et al, 2016), EfficientNets (Tan and Le, 2019), and DenseNets (Huang et al, 2017), CNNs learn increasingly descriptive features while maintaining relatively low computational demands. CNNs were widely used in early proof-of-principle experiments for image-based profiling (Caicedo et al, 2018; Kraus et al, 2016) and continue to be an effective option for feature extraction in the field (Bushiri Pwesombo et al, 2025; Razdaibiedina et al, 2024; Sivanandan et al, 2025; Wong et al, 2023; Moshkov et al, 2024).

Recently, the transformer-encoder architecture has gained prominence in image-based profiling following the introduction of the vision transformer (ViT) (Fig. 2B) (Dosovitskiy et al, 2020). Initially designed for processing sequences of text in natural

**A**

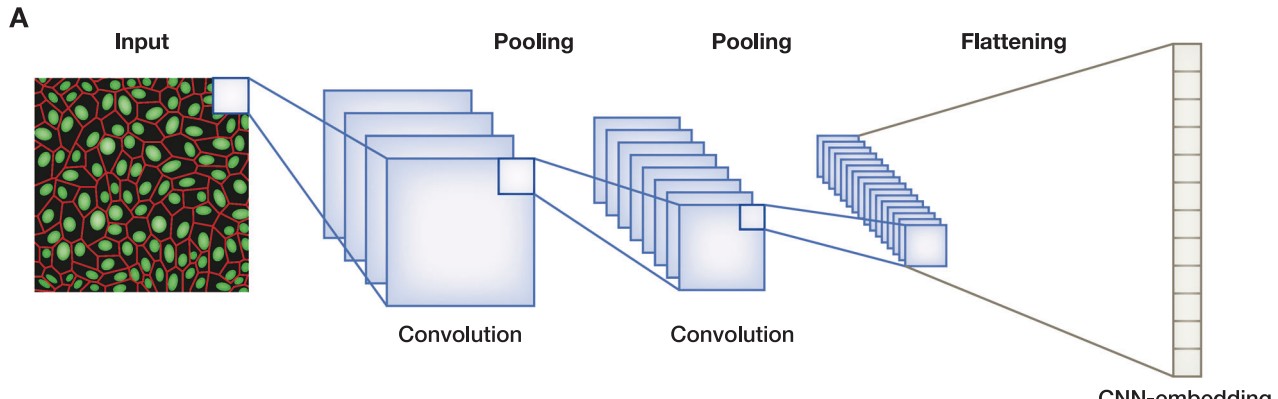

**B**

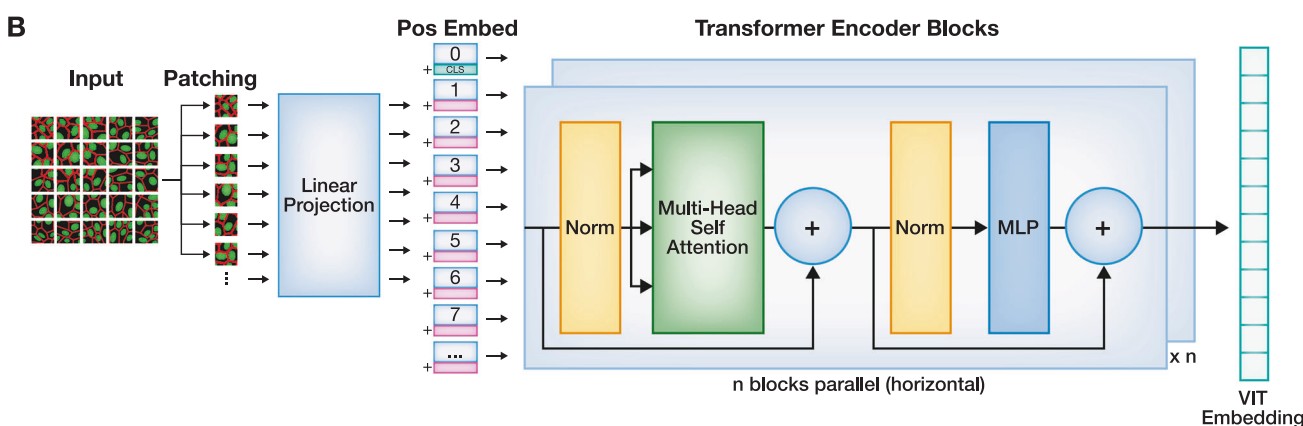

**C**

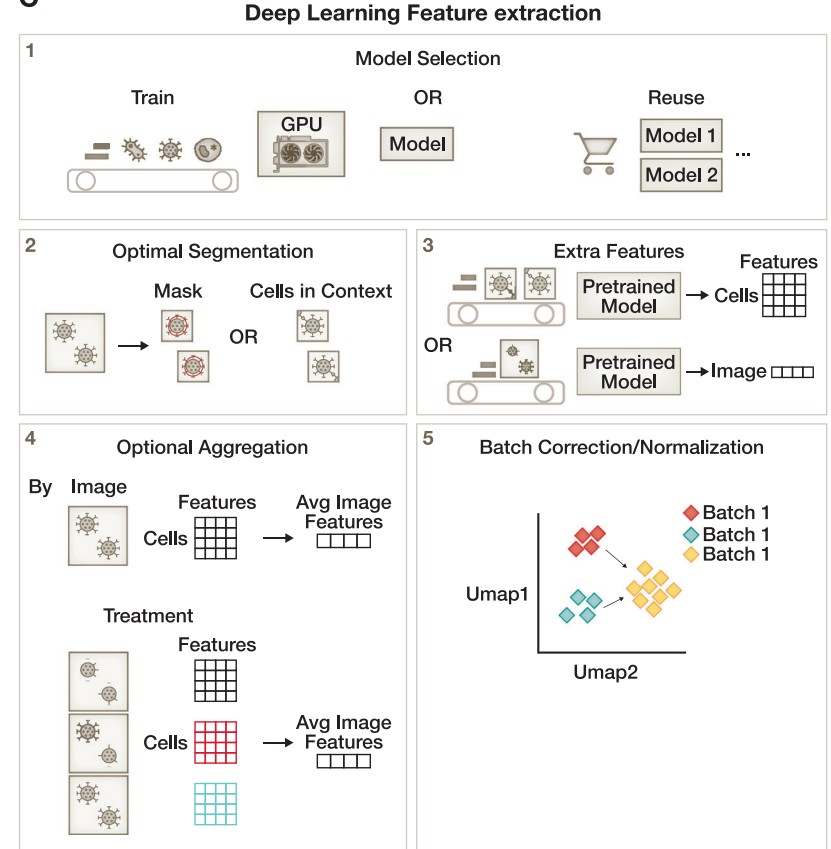

**Figure 2. Deep learning in image-based profiling.**

There are two primary model architectures for deep learning-based feature extraction: (A) Convolutional neural networks (CNNs) process images through stacked convolutional layers to extract local features in a hierarchical manner. (B) Vision transformers (ViTs) process images by dividing them into patches, which are linearly embedded and combined with positional encodings to retain spatial information. Patch embeddings (and an aggregating class token) are then processed by a series of transformer blocks that leverage self-attention mechanisms for global context understanding. The features of the class token are then used for downstream analysis. (C) A generalized pipeline for deep learning in image-based profiling consists of three core steps and two optional steps: (1) model selection, where users choose an architecture and training strategy to construct a model or adopt a pre-trained one; (2) optional segmentation, where cell centroids or masks are produced; (3) feature extraction, producing either image-level embeddings from full fields of views or single-cell representations guided via previously calculated segmentations; (4) optional aggregation, commonly applied to single-cell data by combining features by treatment, well, or image; and (5) normalization and batch correction, a crucial step to account for technical variation and ensure comparability across experiments.

language, transformers adapt to visual data by partitioning an input image into non-overlapping patches, which are then flattened and linearly embedded to produce a sequence of vectors. These vectors are processed in parallel through self-attention blocks (Vaswani et al, 2017), enabling the model to capture global relationships among all image patches at once. This approach allows ViTs to learn complex visual representations, making them particularly powerful in computer vision tasks. Furthermore, the scaling properties of ViTs have made the architecture especially useful when used with large datasets (Zhai et al, 2022). As a result, Vision Transformers have become increasingly used for deep learning feature extraction in recent morphological profiling applications (Bao et al, 2023; Dee et al, 2024; Gao et al, 2025a; Krispin et al, 2025; Bourriez et al, 2024; Kenyon-Dean et al, 2024; Yao et al, 2024; Kim et al, 2025; Cross-Zamirski et al, 2022; Doron et al, 2023; Gupta et al, 2024). Additionally, emerging hybrid architectures that combine CNNs and transformers are being explored, as shown by (Gao et al, 2025b).

## Learning strategies for deep learning in image-based profiling

A common paradigm for training neural networks is supervised learning, which minimizes the discrepancy between model outputs and their corresponding ground truth labels. This approach is especially prevalent in classification tasks, where a model learns to assign predefined labels to new samples. In image-based profiling, labels often represent biological annotations of the cell response to perturbations. However, these labels are frequently unavailable due to challenges in accurate human interpretation, the need for lengthy and costly additional experimentation, or the exploratory nature of profiling experiments. Notable exceptions include large-scale protein localization labels collected through crowdsourcing (Sullivan et al, 2018). Additionally, drug screening applications have leveraged MoA annotations (Godinez et al, 2017; Wong et al, 2023) or bioassay data linked to the measured compounds to train supervised models (Hofmarcher et al, 2019).

Weakly supervised learning has been explored as an alternative strategy to train neural networks with some level of supervision (Caicedo et al, 2018). In this case, models use easily obtained annotations that contain noisy labels, which, while not necessarily corresponding to clearly distinguishable classes, still offer accessible and biologically meaningful signals. These annotations can be used to design an auxiliary task. Although predicting the label is not the end goal, the task helps mitigate label noise and guides representation learning under uncertainty. In microscopy applications, the most common source of weak supervision is the perturbation annotation, which is readily available by design in

nearly all biological experiments. While the perturbation label (e.g., gene or compound) is biologically meaningful, it is a weak annotation because it does not directly reveal its phenotypic effect. In many cases, the phenotypic effect may be neutral or undetectable, resulting in challenges for model training, often addressed with further data curation. Weakly supervised training has been used in Cell Painting experiments that showed between 12 and 23% improvement compared to classical CellProfiler features when matching compounds with the same MoA (Moshkov et al, 2024; Caicedo et al, 2018). Interestingly, models performed better when classical methods were used to remove perturbations that did not produce a strong phenotype, such that the representation was trained on samples with definite phenotypic changes. Weakly supervised training has also been applied to optical pooled screens (Celik et al, 2024; Yao et al, 2024; Sivanandan et al, 2025) and protein localization analysis (Krispin et al, 2025; Kobayashi et al, 2022; Razdaibiedina et al, 2024). In these applications, image-level annotations (e.g., perturbation or protein labels) are treated as proxies for single-cell supervision, improving representation learning.

Self-supervised learning is a training strategy that relies on unannotated image data by defining objective functions that guide the model to learn meaningful representations from the data itself (Ericsson et al, 2021). In this setting, the model learns through an auxiliary task (e.g., predicting missing parts of a microscopy image or distinguishing between augmented views) that is derived entirely from the data itself (Kim et al, 2025). Unlike in weakly supervised learning, where auxiliary tasks are based on noisy or indirect labels, self-supervised auxiliary tasks do not rely on external annotations and instead exploit structural properties of the input data (e.g., image patches). Self-supervision has recently evolved to yield state-of-the-art results in general-purpose computer vision applications (Caron et al, 2021; Chen et al, 2020; He et al, 2022; Kim et al, 2025; Dai et al, 2025) demonstrating that image representations can be obtained without the external (and incomplete) guidance of annotations beyond always available sample-level metadata (Lu et al, 2019; Lafarge et al, 2019; Janssens et al, 2021). Common strategies for self-supervised learning have been applied in the context of image-based profiling, including the Masked Auto-Encoder (MAE) (He et al, 2022), DIstillation with NO labels (DINO) (Dosovitskiy et al, 2020), and contrastive learning (Chen et al, 2020). MAE applications for image-based profiling train models to reconstruct an input microscopy image from partially masked patches, encouraging the model to learn spatial and semantic structure (Kobayashi et al, 2022; Kraus et al, 2024; Lamiable et al, 2023). DINO applications use teacher-student frameworks where the student networks learn to match the output of the teachers on different augmented views of the same

microscopy image, promoting the learning of consistent, label-free semantic representations (Yao et al, 2024; Sivanandan et al, 2025; Fonnegra et al, 2023; Pfaendler et al, 2023; Kim et al, 2025; Morelli et al, 2025). Self-supervised learning methods, particularly DINO trained on multisource data, demonstrated substantial improvements in matching compounds with the same biological annotation, achieving up to 61% improvement in mAP scores compared to traditional hand-crafted features from CellProfiler (Kim et al, 2025). Contrastive learning frameworks operate by training models to bring representations of different augmented views of the same microscopy image closer together in embedding space, while pushing apart views from different images and thereby learning features that capture meaningful differences across perturbations without any supervision (Kim et al, 2025; Bushiri Pwesombo et al, 2025).

Recent work has combined self-supervised learning methods with weak supervision to improve robustness to batch effects and biological noise. Specifically, in both DINO and contrastive learning frameworks, two common adaptations have emerged. First, weak labels (e.g., perturbation labels) are used to select positive pairs from images across different batches or technical replicates, ensuring consistency across nuisance variation (Yao et al, 2024; Bushiri Pwesombo et al, 2025; Cross-Zamirski et al, 2022). Second, weak supervision has been used to stabilize training in single-cell applications by averaging multiple cells to form a more robust embedding of the perturbation instead of directly comparing single-cell embeddings (Yao et al, 2024).

## Image preprocessing and preparation for deep learning in image-based profiling

Adequately training deep learning models requires more than just judicious selection of the architecture and learning strategy. Specifically, how the microscopy images are fed into the models is important to consider. Various models may implement different normalization strategies to standardize the images and facilitate training convergence. In natural images, it is common to normalize images for training by computing the mean and standard deviation of pixels across a large dataset. However, dataset-level statistics may not be optimal for fluorescence microscopy, as these images do not share the same statistical properties as natural images, which are often stored in Red Green Blue (RGB) format. In microscopy images, pixel values can span several orders of magnitude, complicating the application of a single normalization value that accurately reflects the distribution across all images (Yayon et al, 2018). Furthermore, microscopy images often contain substantial background pixels that do not represent cellular structures, skewing dataset statistics and leading to normalization that emphasizes background noise (Kochetov and Uttam, 2024). Therefore, the most common practice is to rescale intensities with the statistics of individual images (self-normalization) on a channel-by-channel basis to better preserve the biological signal (Ma et al, 2024).

Models for image-based profiling can be trained on single-cell images or full images of the entire field of view (Fig. 2C). For training and inference on single cells, a segmentation step restricts images to single-cell regions. When using single-cell segmentations, one could use masked cells, where pixels outside the target segmented cell are zeroed, or with appropriately sized crops centered on a cell, capturing the surrounding context. In one study, cells in spatial context yielded better performance (Moshkov et al,

2024), likely due to imperfect masks or from losing intercellular information. Alternatively, deep learning-based feature extraction can bypass cell segmentation entirely to efficiently compute image embeddings from full images or tiled, non cell-centered crops (Kim et al, 2025; Tang et al, 2024; Luna et al, 2024; Dee et al, 2024; Li et al, 2025). These approaches simplify workflow and computational complexity, but do not allow for features at the single-cell level. Lastly, models using full, downsized images or tiled crops potentially carry more information about the background and can be sensitive to variations in cell count (Kim et al, 2025). A recent study compared the performance of cell-centered crops, tiled crops, and full-image resizing, finding that cell-centered crops yielded the most stable performance across datasets, while tiled crops and full-image resizing were more memory and time-efficient (Xun et al, 2024). In summary, this modeling decision involves important tradeoffs.

Training deep learning models often requires various data augmentations to increase sample variation and reduce overfitting. An augmentation function transforms a real image by introducing random distortions that make it look different. Augmentations traditionally used for natural images (Yang et al, 2022) need to be adapted to work with microscopy images (Gopalakrishnan et al, 2024; Aras, 2017). In general, augmentations do little to combat artifacts like batch effects, but instead address low-level noise in the data to benefit training stability and model performance. Common geometric augmentations include horizontal flips, vertical flips, random resized cropping and rotations, which in all cases result in valid cell images without much computational cost. Additionally, photometric augmentations of intensity and contrast changes can also be introduced, which may be helpful in simulating illumination variations due to microscope hardware differences or staining artifacts (Ma et al, 2023). In cell imaging, photometric augmentations are usually applied on a per-channel basis, in contrast to RGB images, where all channels have their attributes (brightness, contrast, saturation, and hue) transformed simultaneously (i.e., color jittering). For example, selecting images to augment their brightness and intensity at an 80% sampling rate and subsequently doing these augmentations per channel at a 40% rate improved performance over using these augmentations over all channels (Dee et al, 2024). However, geometric random resizing operations, widely used for natural images, can be detrimental for cell images (Kim et al, 2025; Gopalakrishnan et al, 2024). Nevertheless, augmenting intensity shifts in a contrastive learning framework (SimCLR) had the most positive impact (Kim et al, 2025). This may occur because, although stain intensity is a biologically relevant trait, it can be masked by technical noise, leaving the augmentation overall beneficial. Sampling and dropping channels hierarchically was explored as a way to make models robust to missing channels (Bao et al, 2023), and other noise-injection augmentations have been studied by taking inspiration from the physics behind microscopy (Liu et al, 2025). Recent work has explored unique augmentations to microscopy, such as width and height shifts and shearing transformations (Sharma et al, 2025).

## Post-processing features after deep learning training in image-based profiling

After pre-processing and training, deep learning features require post-processing steps for downstream analyses. As previously

discussed, deep learning features can be acquired at multiple resolutions. For example, when derived from segmented single cells, features are measured at the single-cell level. Single-cell features are typically aggregated (e.g., median) across all cells within an FOV or treatment before normalization and further downstream analysis (Xun et al, 2024; Kraus et al, 2024; Sharma et al, 2025; Gupta et al, 2024). It is postulated that this is beneficial due to deep learning-based, single-cell features can be especially noisy, suggesting that between-batches cell state/cycle and technical variance may be amplified (Yao et al, 2024). Additionally, not all cells may exhibit the perturbation effect, which, depending on the assay, may also increase noise (Yao et al, 2024). Alternatively, some models trained against entire FOVs may simply be used to get aggregated features in one shot (Wong et al, 2023; Li et al, 2025; Dee et al, 2024). Similar to hand-crafted features, applying a plate normalization step using MAD-robustize is recommended. This technique transforms embeddings by subtracting the median value and dividing by the mean absolute deviation (MAD), which has been shown to improve the biological signal of deep learning features across various models (Kim et al, 2025). Feature selection, on the other hand, might be unnecessary since only weak correlations between deep learning features are generally observed. For example, the variance and correlation feature-selection methods removed nearly zero features from self-supervised learning embeddings (Kim et al, 2025). Instead, researchers must select a particular output embedding size in advance, aiming to strike a balance that minimizes feature redundancy while preserving expressivity. Some form of dimensionality reduction may still be needed, however, if the embedding space of the particular model used is especially large, to ease computational tractability during statistical analysis. Batch effect correction is also an important step for reducing technical variation introduced in deep learning features across experimental conditions (Arevalo et al, 2024). Models often learn batch effects if they are helpful towards minimizing the learning objective, and therefore they must be mitigated (Sypetkowski et al, 2023; Moshkov et al, 2024) In general, applying a sphering transform to deep learning features followed by robust z-scoring against plate-level statistics yields the most effective post-processing strategy for self-supervised features. Importantly, the order of operations is critical, as performing z-scoring before sphering was found to substantially degrade performance in most cases (Kim et al, 2025).

## Interpreting deep learning features for image-based profiling

Deep learning features present a unique challenge for interpretation, as they are generally anonymous latent variables that do not have explicit names or meaning. This contrasts with traditional image-based profiling, which relies on predefined features extracted using tools like CellProfiler. These features are categorized by image channel, feature type (e.g., shape, texture, correlation), and cell compartment (e.g., nucleus or cytoplasm) and are therefore more interpretable. The challenge of deciphering deep learning models has spurred the growth of explainable AI (XAI), where a key distinction is made between "explainability" (describing the cause for a decision) and "interpretability" (translating an explanation into domain-specific insight) (Rotem and Zaritsky, 2024). Methods for visual interpretability in deep learning for image-based profiling

generally fall into two categories: pixel-level explanations and feature space interpretation. Pixel-level explanations reveal how individual pixels contribute to model predictions, using techniques such as GradCAM (Longo et al, 2024; Gopalakrishnan et al, 2024; Rotem and Zaritsky, 2024). In transformer-based models, the attention mechanism across layers can provide insights into which regions of the input image are most influential in shaping learned representations (Pfaendler et al, 2023). While these methods provide detailed insights at the individual image level, they are often difficult to generalize or aggregate across perturbations, limiting their utility for downstream biological interpretation. Extending these ideas, attribution methods like SHapley Additive exPlanations (SHAP) (Lundberg and Lee, 2017) can also be applied. For example, in a high-throughput context, these methods can explain why a treated cell's profile is "anomalous" by identifying which specific inter-feature dependencies, learned from control cells, were "broken" by the treatment (Shpigler et al, 2025). In contrast, feature space interpretation focuses on understanding the model's embedding space by decoding learned features into images that visually represent cell phenotypes. Generative models have also been explored to gain understanding of how images are organized in the latent space of deep learning models, including variational autoencoders (Lafarge et al, 2019), generative adversarial networks (GAN) (Goldsborough et al, 2017), and conditional GANs (Goldsborough et al, 2017; Lamiable et al, 2023; Fonnegra et al, 2023). For example, an interpretability technique, sometimes called "morphing," uses generative models to create counterfactual in silico images by intentionally altering the model's latent features, which can make the learned phenotypic properties visually apparent (Rotem et al, 2024; Zaritsky et al, 2021). However, further research in this field is needed to deepen our understanding of phenotypic variation of cells in microscopy images and to better connect image-derived representations with other sources of biological knowledge. This remains a fertile and promising frontier for advancing image-based profiling.

## Foundation models for image-based profiling

In recent years, the concept of foundation models has been introduced in image-based profiling. Foundation models refer to large models trained on large amounts of often diverse data, which can be adapted to solve a wide range of tasks that generalize to multiple datasets (Bommasani et al, 2021). Early work repurposed trained, natural-image CNNs by pseudo-RGB channel mapping (Michael Ando et al, 2017; Caicedo et al, 2022; Weisbart et al, 2024). Furthermore, Cell Painting CNN (Moshkov et al, 2024) is a generalizable model trained with weakly supervised learning on five datasets from the Cell Painting Gallery (Moshkov et al, 2024; Weisbart et al, 2024), demonstrating that high technical and biological variation during training are beneficial for downstream performance. More recently, self-supervised learning with vision transformer networks has become the dominant approach for training foundation models in image-based profiling. Several studies (Kim et al, 2025; Morelli et al, 2025; Borowa et al, 2024), conducted comprehensive evaluations of self-supervised learning algorithms for morphological profiling on various datasets, and demonstrated how the approaches can both accelerate processing and improve the accuracy of phenotypic analysis. Kenyon-Dean et al presented a masked autoencoder-based strategy for scaling

**Table 2. Comparing classical features with deep learning on key performance metrics.**

| Metric | Classical features | Deep learning |
|---|---|---|
| Performance | Robust baseline results. | ~20% improvement relative to classical features |
| Usability | Published pipelines and user-friendly software are available to reproduce results for many profiling tasks. | Requires knowledge of the torch ecosystem and the creation of custom extraction scripts. Requires further programming and computing resources to fine-tune/train models. |
| Interpretability | Features are named measurements taken from images, which may provide insights in some cases. | Features are unnamed latent variables. Requires additional methods (e.g., generative models) or further data analysis. |
| Data requirements | No minimum data requirements. | Pretrained models have no minimum data requirements. Fine-tuning and training may need thousands to tens of thousands of samples |
| Runtime | Varies. Needs CPU compute. GPU acceleration is typically not available. For large-scale studies, cluster-based parallelization is needed. | Up to 50x faster for predictions due to GPU acceleration. |
| Hidden costs | Illumination correction, segmentation and feature selection. | Optional segmentation. May require fine-tuning or training |

This table summarizes the main differences between classical morphology-based feature extraction and deep learning–based representations in image-based profiling, emphasizing the trade-offs in performance, usability, interpretability, data requirements, runtime, and hidden costs.

ViT models to ~1.9B parameters and datasets with ~16 M images, and showed how performance improves at that scale (Kenyon-Dean et al, 2024).

Channel adaptation is a challenging hurdle for foundation models. Most models, including those described above, have been trained using the five-channel Cell Painting assay, which is the most common assay for image-based profiling. However, imaging panels can vary with experimental goals, resulting in images with a diverse number of channels and other technical configurations. This poses a challenge to create models that can be reused across experiments because the established practice in computer vision is to fix the image channels ahead of time for neural network training. For example, SubCell models (Gupta et al, 2024), which are foundation models of cell morphology trained with the Human Protein Atlas dataset (Gupta et al, 2024; Thul et al, 2017), present a collection of eight models that can be chosen and configured depending on the number and types of channels in an experiment. Several strategies have emerged to yield foundation models that can adapt to varied numbers of channels. To incentivize the investigation of technical innovations in this field, a benchmark was created for channel adaptive models in microscopy imaging, CHAMMI (Chen et al, 2023). Moreover, CytoImageNet created a diverse dataset of microscopy images and trained a model that averages all channels in a single gray-scale image (Hua et al, 2021). Additionally, Microsnoop (Xun et al, 2024) proposed a strategy where a model is trained to look at one channel at a time, which has been followed by uniDINO (Morelli et al, 2025) and bag-of-channels (BoC) DINO (De Lorenci et al, 2024). A key disadvantage of these models is that concatenating individual channel embeddings is computationally intensive, results in high-dimensional representations where the number of dimensions scale linearly with the number of channels, and fail to capture spatial correlations between them. However, this channel-wise separation can enhance interpretability by allowing feature attributions to be traced back to specific channels. The channel-agnostic masked autoencoder (CA-MAE) model used to train OpenPhenom-S/16 (Kraus et al, 2024) can simultaneously look at all channels in an adaptive way to produce image features. Novel transformer architectures have been developed recently, including ChannelViT (Bao et al, 2023), Chada-ViT (Bourriez et al, 2024), and ChA-MAEViT (Pham et al, 2025),

which incorporate architectural and training innovations to yield improved and more robust channel adaptive capabilities. While these architectures are promising, large-scale foundation models for image-based profiling are still to be trained with these innovations.

Foundation modeling is seen as a "holy grail" in image-based profiling due to its potential to produce highly generalizable representations across diverse biological contexts. However, training such models from scratch requires substantial computational resources, access to large and diverse datasets, extensive benchmarking, and there are many unsolved challenges. A more practical alternative may be to adopt foundational architectures (i.e., models pretrained on large-scale datasets), which can then be fine-tuned for specific applications or datasets within image-based profiling (Ji et al, 2024).

## Comparative analysis of classical features and deep learning

Previous studies have benchmarked the improvements of using deep learning compared to classical image-based profiling features (Doron et al, 2023; Sanchez-Fernandez et al, 2023; Kim et al, 2025; Wong et al, 2023; Moshkov et al, 2024; Kraus et al, 2025; Morelli et al, 2025; Pawlowski et al, 2016; Hofmarcher et al, 2019; Gao et al, 2025a; Doan et al, 2021; Kraus et al, 2024). However, the specific added benefit remains a case-by-case basis and an open research question. We provide a summary of performance metrics comparing deep learning and classical features (Table 2). Specifically, benchmarks include comparing performance in the main biological task, usability, interpretability of features, data requirements, and runtime. On average, these benchmarking studies showcased a relative improvement of ~20% for deep learning approaches over classical features. We note that this does not account for learning paradigm (SSL, WSL, or SL), model architecture or other factors that would prevent this from being a robust quantitative measure of deep learning performance for image-based profiling.

There are also other various trade-offs, unrelated to performance. For example, classical features are most often implemented with graphical user interfaces that facilitate configuration and streamlined preparation of data processing (Stirling et al, 2021; Schindelin et al, 2012). On the other hand, deep learning solutions rarely provide a common interface and will need to be adapted to a

given user's data via custom scripts, requiring additional programming expertise and other challenges to implementation. Furthermore, classical features can be applied to all scales of data: They have no minimum nor maximum data requirements. Deep learning models, on the other hand, require thousands of images to adequately fine-tune or tens of thousands of images to train from scratch (Steiner et al, 2021). Nevertheless, a trained deep learning model can extract representations much quicker than classical approaches. For example, a 50x speed improvement towards generating embeddings via deep learning was reported in (Kim et al, 2025). Lastly, classical features likely require additional pre or post processing, such as segmentation, illumination correction, and feature selection, and these costs should be considered. Conversely, certain deep learning approaches may sidestep some or all preprocessing steps. For instance, segmentation can be skipped by processing image patches (or whole FOVs) instead, which results in less computation work, while yielding similar performance (Kim et al, 2025; Xun et al, 2024).

# Emerging focuses and challenges in image-based profiling

## Single-cell image-based profiling

### Methods for analyzing single-cell profiles

In image-based profiling experiments, single-cell morphological descriptors are typically averaged to the image, well, or treatment level to create an aggregated morphological "fingerprint" (Caicedo et al, 2017). Aggregating data in well-level or treatment-level profiles offers significant advantages by creating a robust and stable representation that minimizes the impact of irrelevant single-cell variability and technical noise. Furthermore, it dramatically reduces the volume of data points, ensuring that downstream computational analyses, like comparing thousands of compound profiles, remain scalable and efficient (Caicedo et al, 2017; Chandrasekaran et al, 2020). For many downstream applications, such as predicting a compound's mechanism of action (MoA), these aggregated profiles have proven highly effective (Scheeder et al, 2018). However, averaging leads to a loss of information about cell heterogeneity and the diversity of cell states that are present (Mattiazzi Usaj et al, 2021; Pearson et al, 2022). As a result, the profiles will be less effective at quantifying changes in situations where this heterogeneity matters, such as co-cultures and other dynamic phenotypic events (e.g., mitosis, differentiation, non-synchronous signaling, and transcription (Sinning et al, 2025; van Dijk et al, 2024; Yin et al, 2020)). Aggregating single cells to the population level, therefore, leads to misrepresentation of the underlying biology and prevents the effective use of image-based profiles for many biologically important research questions (Rohban et al, 2019).

Single-cell image-based profiling offers an effective, affordable, high-content alternative to other single-cell profiling technologies, such as single-cell RNA-seq. Over the last few years, innovations in the image-based profiling field yielded methods that capture heterogeneity, subpopulations, and cell state transitions (Fig. 3A) (Graham et al, 2025; Högel-Starck et al, 2024; van Dijk et al, 2024; Feldman et al, 2019). These methods have been applied to identify drug toxicity, cell types, cell cycle stages, and cell health alterations

in rare subpopulations. They also enabled the quantification of the dynamic effects of perturbations on mixed cell types and subpopulations (Tomkinson et al, 2024; Tegtmeyer et al, 2025; Geng and Kidder, 2025; Okoro et al, 2025). Beyond opening up new insights into heterogeneous morphologies, there is also emerging evidence that single-cell analysis of image-based profiles may improve the prediction of MoAs compared to population-level image-based profiling analysis (Stossi et al, 2024).

Methodologically, the only differences between traditional aggregation methods and emerging single-cell strategies is to skip the aggregation step and to analyze profiles of single cells, instead of populations (Fig. 3B). Best practices are still emerging, but often either features or lower dimensional representations are compared using metrics such as the Kolmogorov–Smirnov statistic and the Earth Mover's distance (Fig. 3C) (Pearson et al, 2022; Ljosa et al, 2013; Stossi et al, 2024; Rubner et al, 2002; Loo et al, 2007; Young et al, 2007; Hutz et al, 2013). In specific applications, there may be knowledge of distinct morphologies in an experiment, which single-cell image-based profiling can incorporate by using supervised classifiers to identify morphological states of cells (Högel-Starck et al, 2024; Heigwer et al, 2023; Frey et al, 2025; Aghayev et al, 2023). Whereas population-level image-based profiles miss changes in subpopulations and cell states due to aggregation, these approaches allow quantification and comparison of distinct morphologies.

The enhanced content and biological understanding of single-cell image-based profiling come with tradeoffs. Specifically, the primary trade-off is between the improved biological signal detection and discovery of subtle phenotypes offered by preserving single-cell heterogeneity, versus the computational efficiency, scalability, and robustness to noise provided by aggregate-level representations (van Dijk et al, 2024; Palma et al, 2025; Garcia-Fossa et al, 2023; Stossi et al, 2024). For example, single-cell image-based profiling has a higher data processing burden. Large image sets like JUMP Cell Painting contain over 1 billion cells, which is beyond the practical application for many algorithms; even computing pairwise similarities among all cells becomes infeasible. Existing algorithms (especially those from the single-cell mRNA profiling field with much smaller experiments) may not scale sufficiently to allow single-cell analysis of image-based profiling experiments, though this challenge is being tackled by new tools such as SPACe (Stossi et al, 2024), scmorph (Wagner et al, 2025), and CytoTable (Bunten et al, 2025) (Fig. 3C). Furthermore, the diversity of research questions addressed with single-cell image-based profiling to date is reflected in the wide variety of single-cell image-based profiling analysis methods. Consolidating approaches into best practices and the development of benchmark datasets will establish single-cell image-based profiling analysis as a valuable addition to the field. Taken together, single-cell analysis has the potential to dramatically increase the insights gained from image-based profiling experiments, but methodological challenges remain (e.g., quality control; discussed below).

### Quality control for single-cell profiles

Using single cells for image-based profiling makes analysis more prone to issues such as segmentation errors and technical biases (e.g., uneven illumination), which are "averaged out" in population-level image-based profiles (Pearson et al, 2022; Stossi et al, 2023, 2022). Currently, however, there is no standardized single-

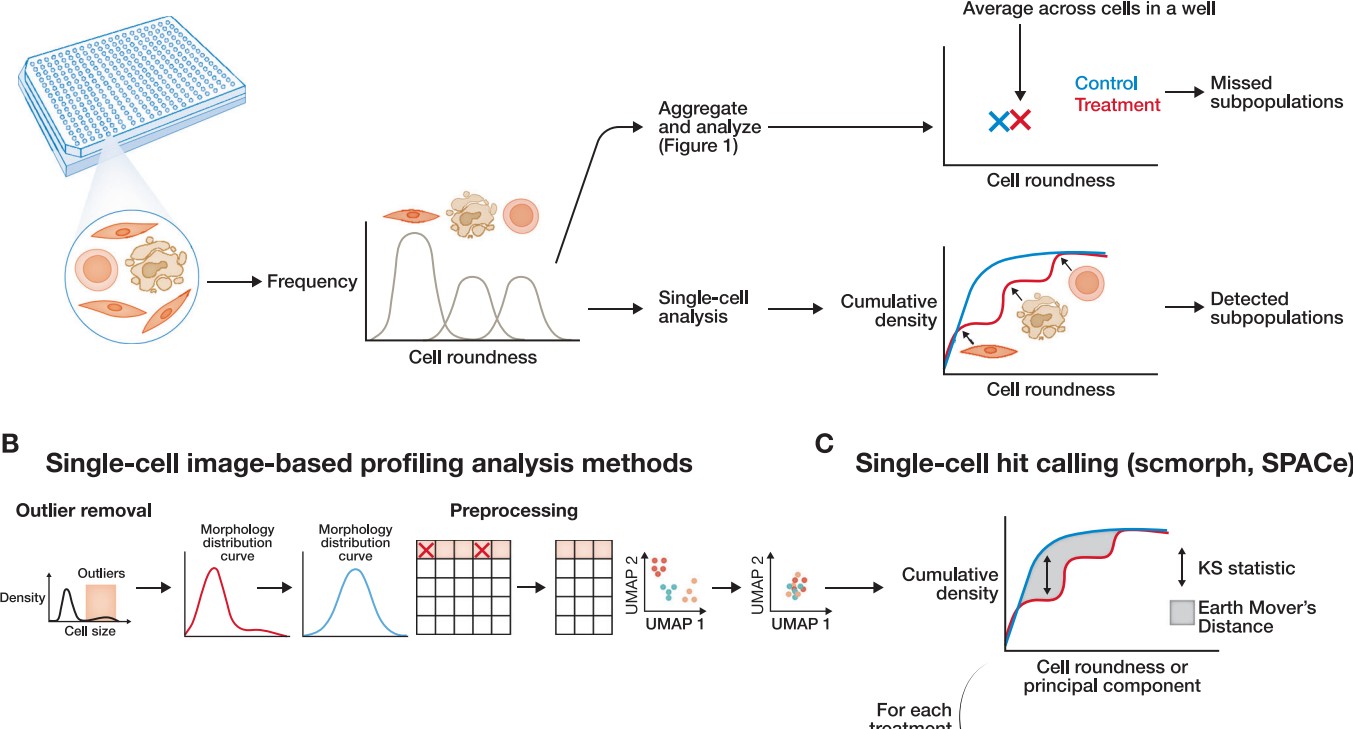

**Figure 3.  Leveraging single-cell resolution in image-based profiling to uncover cellular heterogeneity and improve hit detection.**

(**A**) Comparison between traditional population-averaged profiling and single-cell approaches. While aggregate profiles may obscure heterogeneous cellular responses, single-cell analysis reveals distinct subpopulations, as illustrated by diverging distributions in cumulative density plots. (**B**) Overview of a representative single-cell image-based profiling pipeline, encompassing key stages from image acquisition and feature extraction to cell-level data normalization and quality control. (**C**) Application of statistical tests, such as the Kolmogorov–Smirnov (KS) statistic, and distance-based metrics, such as Earth Mover's Distance (EMD), to cumulative density functions for detecting perturbations that induce significant shifts in single-cell feature distributions, enabling more sensitive and nuanced hit detection.

cell QC approach for image-based profiling, and new approaches are emerging. Newer methods aim to achieve the removal of outliers without the need for training any models. One methodology introduced the development of a two-tiered single-cell quality control methodology based on one extracted feature, endoplasmic reticulum (ER) intensity in the nucleus (Stossi et al, 2022). This study compared ER distributions between treatment and control (DMSO) wells by first trimming extreme values at the fourth and 96th percentiles, followed by normalizations of all wells relative to the DMSO control. Dynamic time warping was applied to adjust the compound ER intensity distribution relative to the DMSO reference. Furthermore, whole experiments failed QC if the earth mover's distance (Rubner et al, 2000) was greater than three standard deviations away from the mean. This method of QC focuses on identifying poor-quality cell profiles in a biological context, removing whole wells or experiments if the response from a compound or control is not as expected (Stossi et al, 2022). A recent advance in single-cell QC also utilizes extracted morphology features, but to specifically detect segmentation problems or other "technical outliers." One such model-free tool, coSMicQC, offers an open-source solution for single-cell QC by leveraging subsets of morphological features, performing z-score normalization, and thresholding based on standard deviations to identify technical outliers (Travers et al, 2025; Tomkinson et al, 2025). It is particularly useful at flagging under- or over-segmented nuclei and missegmented background regions, with AreaShape and intensity features performing well either independently or in combination (Travers et al, 2025). While effective in untreated cells, this approach may overcorrect in perturbed conditions, underscoring the importance of the human-in-the-loop aspect of the software, utilizing tools like CytoDataFrame (Bunten et al, 2025). coSMicQC also includes a contamination detection module that uses nuclear shape and texture features to flag potential mycoplasma contamination, particularly useful when upstream wet-lab tests are inconclusive or overlooked. Single-cell QC in image-based profiling remains challenging as it is difficult to

reliably distinguish biologically meaningful outlier phenotypes (i.e., "interesting cells") from analysis-inappropriate cells or segmentation artifacts (Caicedo et al, 2017). Therefore, establishing standardized methods to quantitatively report the impact of QC on downstream results will be critical for navigating the trade-off between removing artifacts and preserving true biological signals (Subramanian et al, 2022; Qiu et al, 2020; Tomkinson et al, 2025).

## Optical pooled screening

Optical Pooled Screening (OPS) has recently emerged as a transformative technology to combine with image-based profiling, enabling high-throughput, single-cell resolution mapping of genotype-to-phenotype relationships at an unprecedented scale (Walton et al, 2022). Traditional pooled genetic screens, while scalable, relied on unidimensional readouts (e.g., cell viability or reporter expression), where perturbation effects are collapsed into scalar scores that average out cell heterogeneity, thus masking the complexity of cell responses. OPS with image-based profiling overcomes these limitations by integrating high-content imaging (HCI), capturing rich morphological, and spatial features from microscopy data, including subcellular structures, protein localization, and dynamic behaviors (Walton et al, 2022; Kudo et al, 2024; Feldman et al, 2019). These visual phenotypes are then linked to specific genetic perturbations through in situ genotyping methods such as fluorescence in situ hybridization (FISH) or in situ sequencing (ISS), offering a multidimensional view of cellular states. While OPS studies typically involved targeted visual readouts exploring particular phenotypes of interest (and therefore looking to identify a small number of "hits" from among thousands of tested genes, in a conventional screening format), it has also been used for image-based profiling (Funk et al, 2022) and was recently adapted to be compatible with the Cell Painting assay by using cleavable, destainable fluorophores linked via disulfide bonds, which are removed using the reducing agent TCEP (Ramezani et al, 2025). This innovation enables ISS (and potentially other rounds of phenotypic staining) to be successful after Cell Painting. The versatility of OPS has catalyzed the development of specialized platforms to expand its applications, such as measuring multi-dimensional phenotypes, such as subcellular organization, protein localization, and morphological dynamics, that would be lost in conventional unidimensional screens (Walton et al, 2022; Sivanandan et al, 2025). For example, CellPaint-POSH (Sivanandan et al, 2025) adapts the Cell Painting assay to incorporate RNA signal by replacing traditional mitochondrial dyes (e.g., Mito-Tracker) with RNA-friendly alternatives such as Mitoprobe and introducing reverse transcription prior to staining to preserve RNA integrity during in situ sequencing ISS.

Processing OPS data builds upon foundational steps in classical image-based profiling (i.e., illumination correction, segmentation, and feature extraction) while introducing specialized components for single-cell barcode deconvolution for each cell. Barcode deconvolution links each cell to its corresponding perturbation by performing in situ genotyping. After barcode deconvolution, standard image-based profiling workflows can be applied, with the exception of the normalization step. Because multiple perturbations exist within the same well, normalization must account for cell-level rather than well-level perturbation assignments, requiring adapted strategies to avoid mixing signals across

conditions. One strategy normalizes features at the single-cell level using non-targeting control cells within the same well, applying median absolute deviation (MAD) scaling to reduce the influence of outliers (Carlson et al, 2023). To correct for plate-to-plate variation, another approach aggregates single-cell data by guide within each plate, followed by z-score normalization at the guide level on a per-plate basis (Ramezani et al, 2025). In optical enrichment-based screens, imaging is used to sort cells by phenotype, and downstream sequencing provides sgRNA abundance as the final readout. In these cases, a "phenotypic score" is computed for each sgRNA, representing the normalized log-fold change in abundance between phenotypically selected and control cell populations (Yan et al, 2021). Unlike conventional pooled screening methods, OPS workflows must preserve single-cell resolution and phenotypic diversity across all processing steps, requiring scalable, robust pipelines tailored to the unique demands of high-dimensional image-based data (Di Bernardo et al, 2025).

Nevertheless, OPS has also motivated several customized computational and assay advancements. For example, to improve disentanglement of biological signal from noise, Multi-ContrastiveVAE (mcVAE) employs a contrastive variational autoencoder framework that isolates perturbation-specific phenotypes from intrinsic cellular heterogeneity and technical artifacts in OPS images (Wang et al, 2023c) Similarly, GRAPE uses generative adversarial networks (GANs) to learn biologically meaningful, style-invariant representations by disentangling technical confounders, although its performance is constrained by limited training data and high computational requirements (Bigverdi et al, 2024). Enhancements on the assay side have expanded the OPS scope as well. For example, the CRISPRmAP platform integrates ISS with multiplexed immunofluorescence and RNA detection, enabling barcode readout across diverse cell and even tissue environments (Gu et al, 2024, 2023). However, RNA degradation and suboptimal detection efficiency continue to pose challenges, particularly in complex tissue contexts (Gu et al, 2023). More recently, Perturb-View has enhanced the scalability of OPS by introducing signal amplification techniques that reduce imaging time and enable simultaneous detection of proteins, RNAs, and morphological features (Kudo et al, 2024). PerturbView still contends with technical issues such as barcode diffusion and the requirement for large sample sizes in tissue-like environments.

While applying OPS with image-based profiling enables high-dimensional, high-throughput phenotypic screening at single-cell resolution, it also presents substantial experimental and computational challenges that can render the choice between pooling and arraying context- and experiment-dependent (Bock et al, 2022). The construction and validation of large-scale perturbation libraries remain labor- and resource-intensive (Walton et al, 2022). Barcode delivery, expression efficiency, and transcript stability can introduce variability that affects recovery fidelity. Additional technical constraints, including spectral overlap between phenotyping dyes and sequencing fluorophores, as well as RNA degradation during staining, must be meticulously controlled to preserve data quality. Computationally, OPS generates vast volumes of imaging data, often reaching terabyte scales per screen, necessitating robust and scalable pipelines for barcode decoding, feature normalization, and batch correction (Labitigan et al, 2024). Nevertheless, OPS marks a paradigm shift in image-based profiling, offering a unique framework for genome-wide, multimodal interrogation of cellular phenotypes, with advantages of

unprecedented scale and more consistent technical artifacts across perturbations, which can provide more sensitivity.

## Temporal and 3D image-based profiling

The traditional image-based profiling approach measures cells in two spatial dimensions (X, Y) at a single time point and up to five spectral dimensions on a standard fluorescence microscope (Bray et al, 2016; Way et al, 2021). However, this traditional approach measures only a single snapshot of cells and restricts measurements to a single slice (whether through maximum projection of multiple z slices or analyzing a single z slice). While introducing a temporal and the third spatial dimension increases experimental and computational complexities, it also increases the possibility of deriving important biological insights. Multiple studies have shown that only live-cell imaging can capture dynamic cell processes that otherwise a static method would not (Wang et al, 2020; Padovani et al, 2022; Gladkova et al, 2024).

Temporal imaging (typically live-cell) provides more biological context and circumvents the survivorship bias of static image-based profiles. In other words, typical fixed time point image profiles capture only the cells still attached to the surface at the end of the experiment, whereas time-lapse imaging, for example, capturing images every hour, can capture the profiles of cells that die in addition to those that did not, giving more biological insight into the perturbation (Lippincott et al, 2025a). Cell Painting requires fixation of cells, thus advances in techniques for observing live-cells are critical for time-lapse imaging. Typically, minimally toxic live-cell dyes can be used to perform "Live cell painting" such as ChromaLive and Acridine orange (Sivagurunathan et al, 2025; Cottet et al, 2023; Garcia-Fossa et al, 2025). Alternatively, cells can be genetically engineered to express proteins fused to fluorescent proteins or markers. These approaches can be combined with advances in spinning disk confocal and light sheet microscopy that make the acquisition of large datasets over time possible (Dent et al, 2024). Additionally, these technologies now enable high-throughput screens where temporal information, such as signaling dynamics, serves as the primary readout instead of a single static endpoint (Goglia et al, 2020). The complex trajectories of cells have necessitated analytical tools like CODEX, which uses deep learning to explore and classify diverse signaling dynamics landscapes (Jacques et al, 2021).

Temporal imaging presents new informatics challenges, depending on the approach taken. One option is to simply capture population profiles from live cells at multiple time points (perhaps with one fixed time point at the end, allowing more staining options). This can be more informative than a single endpoint while minimizing microscopy time. The other option is to take time points frequently enough to enable single-cell tracking, such that each profile represents the same cell over time. Many cell tracking software exist, such as TrackMate (Ershov et al, 2022) and Ultrack (Bragantini et al, 2024), however, there remain both major technical and computational hurdles for incorporating these single-cell tracking methods into high-throughput temporal image-based profiling pipelines (Lefebvre et al, 2025; Holme et al, 2023). While widely used single-cell tracking tools, like TrackMate, offer scripting capabilities for automated "headless" execution, achieving sufficiently low error rates across large-scale datasets remains a major technical hurdle (Cayuela López et al, 2023). In smaller experiments, tracking errors (e.g., identity swaps or lost

tracks) are typically corrected through manual curation using the software's graphical user interface (GUI). However, in high-throughput screening, where datasets contain billions of cells, such manual intervention is impossible. Consequently, the compounding nature of tracking errors remains the primary barrier to scalability. Some limitations of single-cell tracking can be addressed through thoughtful experimental design, for instance, using shorter time intervals to better track highly motile cells (Yang et al, 2020).

Like the temporal dimension, the third spatial dimension (depth) remains underexplored in image-based profiling. 3D image-based profiling holds the promise of revealing more nuanced and detailed cellular signatures than its 2D counterpart, and it expands high-content measurements to in vivo, ex vivo, in vitro organoids, and in situ tissue samples (Ong et al, 2025; Chen et al, 2018). Indeed, rapid development in the field is beginning to realize this potential, with new protocols adapting 2D assays like Cell Painting for 3D spheroids and enabling high-content morphological profiling of these models (Jeremiasse et al, 2024; Bozal et al, 2024; Ringers et al, 2025; Huang et al, 2024; Lukonin et al, 2021; Viana et al, 2023; Ramm et al, 2022; Betge et al, 2022). This progress is further supported by the creation of dedicated high-content screening software that uses automated AI-driven analysis to facilitate robust, high-throughput phenotyping in 3D models (Diosdi et al, 2025; Kok et al, 2025; Madan et al, 2025). However, the two large limiting factors to 3D image-based profiling are the ability to (1) accurately, and quickly segment single cells and (2) extract morphology features. 3D image-based profiling is also essential for accurately capturing complex cell shapes, such as neurons or macrophages with extended protrusions, that may appear disconnected or fragmented in a 2D slice. While there are many tools for 3D segmentation (Wang et al, 2022; Mahmud et al, 2023; Perera et al, 2024), they each require large annotated ground truth datasets, large compute requirements such as GPU resources, and some lack computational complexity and resources scalability. Furthermore, extracting morphology features is done differently for 3D images compared to 2D images. Most 3D features are not hand-drawn features using traditional computer vision principles (Ong et al, 2025). However, there has been substantial work in the geospatial information services field using point clouds as a means for inputs into deep learning transformer-based frameworks to extract non-interpretable learned tokens for representation of single cells in 3D space (Yu et al, 2022; Krentzel et al, 2025). Lastly, advances in 3D microscopy techniques, such as lattice light-sheet microscopy, oblique plane microscopy and other super-resolution methods, have made imaging in three dimensions more advantageous and information-rich than ever before (Chen et al, 2014). However, the relatively low throughput of these advanced systems currently remains a barrier for generating the large-scale experiments that are central to adapting image-based profiling solutions, representing an active area of research and development (Danial, 2025; Bond et al, 2022).

While holding much promise, temporal and spatial image-based profiling incur increased computational demands, non-standard bioinformatics data processing pipelines, and significant time and data storage requirements. Key barriers for 3D image-based profiling include a lack of robust methods for voxel-based morphological analysis, higher computational demands, less intuitive feature extraction pipelines, and a relative scarcity of tools tailored to 3D cell data.

## Virtual staining for image-based profiling

Although fluorescence imaging remains the dominant modality for capturing cell phenotypes in image-based profiling, it has several limitations. Fluorescence dyes can be phototoxic, most staining procedures such as antibodies require fixing cells, multi-channel optical setups can be expensive, staining is sensitive to protocol variability, and fluorescence is limited in multiplexing due to the need to minimize spectral overlap between fluorescence channels. These constraints motivated the development of machine learning techniques, collectively termed "virtual staining" or in silico stain prediction, that predict fluorescence signals from label-free modalities such as brightfield or phase-contrast imaging, which are simpler, lower-cost, and compatible with live-cell imaging (Cross-Zamirski et al, 2022; Ounkomol et al, 2018; Liu et al, 2025; Le and Lundberg, 2024). This strategy is distinct from directly profiling label-free images; although it is possible virtual staining improves information content in the images, so far its central motivation is to translate label-free data into a format that is compatible with the vast ecosystem of established tools, datasets, and pre-trained models originally developed for fluorescent assays like Cell Painting (Cross-Zamirski et al, 2022; Seal et al, 2024b; Tonks et al, 2023). To perform virtual staining, several architectures have been proposed, including CNNs, GANs, and diffusion models in U-Net frameworks (Xing et al, 2024; Ronneberger et al, 2015; Elmalam et al, 2024; Zaritsky et al, 2021; Navidi et al, 2024). Image-based profiles can then be derived from these virtually stained images, enabling downstream analyses using existing pipelines.

A key challenge for this emerging approach is to demonstrate that profiles from virtually stained images can reliably yield the same biological insights as those from real stains in downstream applications, or at least improved insights relative to the label-free images themselves. To our knowledge, virtual staining has not been used in any published studies for image-based profiling applications. However, recent work has extensively used image-based profiling pipelines to rigorously evaluate the biological faithfulness of virtual staining (Follain et al, 2024; Cross-Zamirski et al, 2023; Le and Lundberg, 2024; De Vries et al, 2025). For example, a multi-level evaluation framework that uses CellProfiler-derived features and high-throughput screening quality metrics like the robust Z prime (RZ') factor to demonstrate that while virtual nuclei and cytoplasm stains are reliable, other more complex structures like DNA-damage spots are not accurately reproduced (Tonks et al, 2023). Other studies have used morphological profiles to show that class-guided diffusion models, which incorporate experimental metadata, can improve performance in downstream tasks like mechanism of action prediction (Cross-Zamirski et al, 2023). Crucially, the generalizability of these models remains a central question; a systematic analysis by Tonks et al revealed that model performance on unseen cell types and phenotypes is highly variable and depends on the biological context of the training data, underscoring the complexity of creating universally applicable models (Tonks et al, 2025).

Nevertheless, image-based profiles of virtually stained images have served as metrics to evaluate virtual staining model performance (Cross-Zamirski et al, 2022; Barteneva and Vorobjev, 2015; Wieslander et al, 2021; Tonks et al, 2023). Additionally, biological characteristics such as cell-type, morphological measurement, and morphological compartment influence model training

dynamics and performance (Cross-Zamirski et al, 2022; Tonks et al, 2025). Therefore, it may improve generalizability to incorporate image-based profiling as a perceptual loss during model training. A perceptual loss leverages a pre-trained network to extract feature maps from both the generated and target images, and computes the loss based on differences in these high-level feature representations to capture semantic and structural discrepancies beyond pixel-wise differences. In other words, minimizing the differences between virtual and ground-truth image-based profiles may preserve biologically relevant structures in virtually stained images, potentially improving robustness to experimental variability. The substantial promise of virtual staining, offering lower cost, reduced perturbation, and greater scalability, makes the continued validation and refinement of these methods a crucial and exciting frontier in image-based profiling.

## Integration of omics

Image-based profiling and omics technologies are increasingly valuable in biomedical research. Although both aim to characterize cell states and responses to stimuli, they provide complementary perspectives (Table 3). Omics technologies capture quantitative information of molecular features and abundances, which allows researchers to identify pathways, regulatory networks, and molecular alterations related to cell states and stimulations (Baysoy et al, 2023). While morphological profiling may not offer the same depth of molecular information as other omics approaches, it possesses several unique advantages that make it a valuable tool in biological research. Most dramatic is the cost; in an academic environment, image-based profiling costs roughly ~$2 per well when processing an entire plate of prepared cells, including all labor, instrumentation, reagents and informatics. This is more than 10x cheaper per sample than bulk mRNA and protein profiling methods, and 1000x cheaper for single-cell resolution. While scRNA-seq provides richer molecular resolution, image-based profiling provides phenotypic and spatial context, which are often critical for functional interpretation. This, therefore, impacts the scale and nature of biological experiments that can be designed. The type of information gained by image-based profiling is also substantially distinct relative to omics. Image-based profiles measure cellular phenotypes (morphology, protein localization, etc.). When directly compared, this information has roughly similar power to predict cell state as high-throughput mRNA profiling using LINCS (Moshkov et al, 2023; Seal et al, 2022; Haghighi et al, 2022; Way et al, 2022a) as well as BBBC047 (Bray et al, 2017; Tian et al, 2023; Haghighi et al, 2022; Hofmarcher et al, 2019; Seal et al, 2025) or ~200-protein secretome profiling (Dagher et al, 2025) using tasks such as predicting a chemical's activity in a given assay, or a compound's mechanism of action, each modality typically predicts a unique subset but with substantial overlap. Image data can reveal functional states, such as changes in protein localization, organelle structure, or cell morphology, that are difficult to capture from molecular measurements alone (Driscoll and Zaritsky, 2021; Way et al, 2022a). Therefore, compared with omics analysis, image-based profiling has great advantages in the initial screening of potential drug candidates and studying the phenotypic consequences of perturbations, particularly at scale (Watson et al, 2022).

Recent advances in assays that combine imaging and omics by jointly profiling both modalities within the same sample enable

**Table 3.**  Comparison of omics and image-based profiling modalities.

|  | Omics technologies | Image-based profiling |
|---|---|---|
| **Data** | Molecular sequences, abundances, or concentrations (DNA, RNA, proteins, metabolites) | Visual images of cells and tissues from microscopy (brightfield, fluorescence, etc.) |
| **Features** | Molecular identity, quantity, interactions | Morphology, phenotype, dynamics of cellular processes, and quantity of stained components |
| **Spatial** | Lost in bulk sequencing; spatial omics technologies retain positional information in tissue but rely on indirect inference of structure (e.g., molecular barcodes, transcript counts). | Can be directly visualized in cells and tissues via microscopy, enabling cellular and subcellular resolution of morphology and marker localization. |
| **Temporal** | Captured through time-series experiments (different samples at various time points), or inferred statistically, but typically cannot provide continuous dynamic information | Possibility to directly capture live-cell images, allowing for the observation of dynamic processes in real time |
| **Scalability** | High-cost high-throughput molecular profiling | Inexpensive high-throughput with automated microscopy and high-content screening. At least 10x cheaper per sample than bulk mRNA and protein profiling methods, and 1000x for single-cell resolution |
| **Destructive** | Typically, destructive as cells are lysed to extract molecules, some single-cell omics methods can be non-destructive (e.g., Live-seq). Some techniques allow supernatants to be analyzed, which is non-destructive | Can be non-destructive (live-cell imaging), allowing for longitudinal studies on the same cells; or destructive (fixed-cell imaging) |
| **Examples** | Bulk and single-cell RNA-seq, whole-genome sequencing (WGS), mass spectrometry (MS)-based proteomics and metabolomics | Fluorescence microscopy, confocal microscopy, high-content screening (HCS), live-cell imaging, quantitative phase imaging |

This table contrasts molecular omics technologies with microscopy-based image profiling across data type, feature representation, spatial and temporal resolution, scalability, destructiveness, and typical applications.

more comprehensive cellular characterization than either modality can achieve alone (Alieva et al, 2023). One widely adopted approach is imaging-driven omics, which includes behavior-guided transcriptomic methods like Image-seq, used to isolate and sequence cells with specific phenotypic behaviors (Haase et al, 2022), and Live-seq, which enables RNA extraction from live cells while preserving their viability and capturing behavioral dynamics via temporal imaging (Chen et al, 2022). These strategies enhance our understanding of how cellular phenotypes evolve in response to perturbations and aid in mapping underlying molecular pathways (Liu et al, 2024c). A growing body of work has successfully applied image-omics integration to uncover novel biological insights across diverse systems (Yoon et al, 2024; Moshkov et al, 2023; Watson et al, 2022; Tang et al, 2024). For example, combining 3D imaging with transcriptomic data revealed 27 novel behavior-specific gene signatures in engineered T cells, which are undetectable in unimodal analyses (Dekkers et al, 2022). Similarly, image-guided genomics applied to invasive cancer populations uncovered atypical VEGF-driven angiogenic signaling in specific regions of tumor invasion (Konen et al, 2017). Recent computational frameworks further advance this integration; for instance, iIMPACT enhances spatial domain and gene expression analyses by integrating histological images with molecular atlases (Jiang et al, 2024). Additionally, deep learning models like CellDART (Bae et al, 2022) and STACI (Zhang et al, 2022) facilitate the integration of spatial transcriptomics and imaging, improving cell-type deconvolution, gene imputation, and joint biomarker discovery (Luo et al, 2024).

Together, these developments position image-based profiling as a powerful modality for capturing cell phenotypes with high spatial and single-cell resolution and alongside omics. When integrated with genomics, transcriptomics, proteomics, and other omics layers, imaging enables researchers to link phenotypic changes to underlying molecular mechanisms (Nassiri and McCall, 2018). This synergy enhances biomarker discovery, improves the understanding of disease mechanisms, and supports more precise evaluation of therapeutic responses and genetic perturbations. As analytical

frameworks continue to evolve, image-based profiling is emerging as a foundational platform for multi-omics integration and systems-level biological insight.

## Batch correction for image-based profiling

Batch effect correction has long been a cornerstone of large-scale biological data analysis, from transcriptomics to image-based profiling (Dataset EV1). These methods aim to align data from different experimental conditions or instruments so that variation due to calibration, reagents, or processing steps does not over-shadow true biological signals (Arevalo et al, 2024). Traditional approaches, such as ComBat (Johnson et al, 2007; Arevalo et al, 2024) and Sphering (Michael Ando et al, 2017; Arevalo et al, 2024), remain widely used but face limitations when applied to image-based profiling. ComBat assumes additive and multiplicative linear batch effects, which may oversimplify complex, nonlinear technical variation present in large-scale imaging datasets (Johnson et al, 2007). Sphering requires reliable negative control samples in each batch, which can be problematic when controls are missing, poorly distributed, or biologically variable (Michael Ando et al, 2017). Representation-learning methods like Mutual Nearest Neighbors (MNN) (Haghverdi et al, 2018), fastMNN (Haghverdi et al, 2018), Scanorama (Hie et al, 2019), and Seurat (CCA/RPCA) (Stuart et al, 2019) rely on overlapping biological subpopulations to align batches. While effective in single-cell transcriptomics, these approaches struggle when image-based profiles lack shared states or when global phenotypic shifts occur (Arevalo et al, 2024). Collectively, these challenges highlight a need for more flexible, biologically informed, and scalable strategies tailored to microscopy data.

Deep learning-based approaches are emerging as powerful alternatives for mitigating batch effects in image-based profiling, particularly in high-dimensional, large-scale datasets. Techniques developed in single-cell transcriptomics offer a roadmap for their application to imaging data (Danino et al, 2024; Yu et al, 2024;

Tran et al, 2020; Lotfollahi et al, 2022; De Donno et al, 2023). For example, DESC reduces batch variation through iterative latent-space clustering while preserving predefined biological signals (Li et al, 2020; Luecken et al, 2022), whereas BERMUDA uses autoencoders and transfer learning to align shared cell types across batches (Wang et al, 2019; Guo et al, 2022; Zhan et al, 2024). Probabilistic models like scVI (Lopez et al, 2018) balance batch-effect mitigation with preservation of nuanced biological variation, while scArches (Lotfollahi et al, 2022) and scPoli (De Donno et al, 2023) extend these ideas for transfer learning and multi-scale integration. Despite their promise, these algorithms require adaptation for image-based profiling, where batch characteristics differ from transcriptomic data.

Emerging approaches increasingly integrate batch correction directly into feature extraction rather than relying on post hoc adjustments. Self-supervised and semi-supervised frameworks leverage experimental metadata, such as treatment or compound labels, to enhance learning of biologically relevant features while controlling for batch effects. WS-DINO samples global and local crops from different images with the same treatment, achieving state-of-the-art MoA prediction performance on the BBBC021 dataset (Cross-Zamirski et al, 2022). SemiSupCon combines supervised contrastive learning on annotated pharmacological classes with self-supervised learning on unlabeled datasets (Bushiri Pwesombo et al, 2025), and Batch Effect Normalization (BEN) aligns experimental batches during training and inference to improve generalization (Lin and Lu, 2022).

Other strategies extend these ideas using domain adaptation, set-level consistency, or style-transfer. CODA (Haslum et al, 2023) adapts feature extractors to new batches without retraining classifiers, while CDCL (Haslum et al, 2022) enforces consistency across images with shared biological signals. Set-DINO aggregates embeddings from sets of cells sharing the same perturbation across batches (Yao et al, 2024), and style-transfer-based methods such as IST (Pernice et al, 2023) and IMPA (Palma et al, 2025) generate synthetic images or disentangle content and style to remove batch-specific artifacts. Collectively, these methods highlight a trend toward dynamic, batch-aware representation learning in microscopy datasets.

In summary, emerging batch correction methods for image-based profiling are shifting from static normalization to integrated, batch-aware feature learning. By leveraging metadata, self-supervised learning, set-level consistency, or style transfer, these approaches reduce reliance on post hoc corrections and improve generalization. Challenges remain, including tuning requirements, dependence on control labels or large datasets, and the need for standardized benchmarks to evaluate performance across imaging platforms. Overcoming these hurdles is essential for translating these methods into robust, scalable tools for high-content screening and other microscopy-based applications.

## Profile similarity metrics

Image-based profiles serve as compact representations of cell state, enabling systematic comparisons across diverse experimental conditions. Advances in feature extraction, spanning both classical approaches and deep learning-based embeddings, have greatly expanded the capacity of image-based profiling to detect subtle and complex phenotypic effects. Central to these efforts is the measurement of profile similarity, which provides a quantitative foundation for grouping related perturbations, inferring MoAs, and assessing experimental reproducibility. The ability to distinguish biologically meaningful variation from technical noise depends critically on the choice of similarity metric and its alignment with the structure of the data. As profiling experiments continue to grow in scale, resolution, and complexity, similarity metrics have evolved to meet increasing demands for sensitivity, robustness, scalability, and interpretability.

Correlation-based metrics provide a foundational approach for quantifying phenotypic similarity, offering statistically grounded and interpretable measures to group perturbations by bioactivity. Pearson correlation is widely used for capturing linear associations and its invariance to feature scale (Caicedo et al, 2017; Pahl et al, 2023), but is limited in datasets with nonlinear relationships or redundant features (Janse et al, 2021). Non-parametric alternatives, such as Spearman's rank correlation and Kendall's tau, relax linearity assumptions, improve robustness to outliers, and increase reliability in small or tied datasets (Stossi et al, 2024; Caicedo et al, 2022; Schober et al, 2018; Cimini et al, 2023). These approaches broaden the analytical landscape for profile comparison, though trade-offs remain in sensitivity, computational efficiency, and interpretability.

Distance-based metrics offer a complementary paradigm by treating profiles as vectors in multidimensional feature space and quantifying phenotypic divergence. Euclidean distance, while intuitive and widely applied, suffers from the "curse of dimensionality" and sensitivity to outliers (Caicedo et al, 2017). Manhattan distance provides a more robust alternative but ignores feature correlations (Reisen et al, 2013; Aggarwal et al, 2001). Mahalanobis distance mitigates these issues by accounting for feature covariance, down-weighting redundancy, and adjusting for scale differences (Gao et al, 2025a; Hughes et al, 2020; Nyffeler et al, 2021; Hutz et al, 2013), though reliable covariance estimation can be challenging in high-dimensional or sparse datasets and may require regularization or robust statistical approaches (Etherington, 2021; Leys et al, 2018; Vulliard et al, 2022). More recent distance-based approaches, including cosine similarity and Earth Mover's Distance (EMD), have addressed some of these limitations. Cosine similarity is widely adopted in high-dimensional latent spaces, capturing angular relationships rather than absolute distances (Moshkov et al, 2024; Leys et al, 2018). EMD, in contrast, compares entire distributions of profiles between conditions, making it sensitive to subtle shifts in phenotypic populations that mean-based measures may overlook. Both metrics have shown stability across diverse datasets and pharmacological screens (Ljosa et al, 2013; van Dijk et al, 2024; Gupta et al, 2024; Wang et al, 2023a; Stossi et al, 2024), but limitations remain: cosine similarity is insensitive to magnitude, single-cell heterogeneity, and feature weighting, while EMD is computationally expensive, less interpretable, and cannot directly confirm mechanisms of action (van Dijk et al, 2024; Gupta et al, 2025; Meng et al, 2024; Riccardo et al, 2021).

Phenotypic separability can also be evaluated directly using classification models, where the ability to distinguish treatment classes provides empirical evidence of morphological differences in high-dimensional space (Doron et al, 2023; Murthy et al, 2025; Mousavikhamene et al, 2021). This approach excels at detecting subtle, distributed phenotypic effects but requires substantial labeled data, may generalize poorly across contexts, and is less

interpretable than similarity-based methods. Recent advances, including feature attribution, model-agnostic explainability, and biologically informed architectures, have improved both interpretability and generalization (Kim et al, 2025; Razdaibiedina et al, 2024; Lamiable et al, 2023; Lotfollahi et al, 2023). As machine learning evolves, classification approaches are poised to play an increasingly important role in morphological profiling, particularly for fine-grained discrimination and predictive modeling across heterogeneous datasets.

Recent frameworks have further advanced the evaluation of perturbations, enabling simultaneous assessment of compound efficacy and specificity. CNN-based *phenoprints* projected onto disease axes quantify *on-perturbation* (reversal of disease phenotype) and *off-perturbation* (unrelated morphological changes) scores (Heiser et al, 2020; Cuccarese et al, 2020; Victors et al, 2025). Complementary retrieval-based metrics, such as mean average precision (mAP), rank profiles by similarity and assess replicate consistency to capture subtle morphological differences while accounting for experimental noise (Kalinin et al, 2025; Chandrasekaran et al, 2024a; Ramezani et al, 2025; Ringers et al, 2025; Lippincott et al, 2025b). While these approaches reduce assumptions of linearity and improve sensitivity, their performance depends on preprocessing, choice of similarity metric, and the quality of embeddings; they may also underrepresent effect magnitude or be limited in significance testing for small sample sizes.

The effectiveness of distance metrics can depend on the strength of the induced phenotype (Moshkov et al, 2024; Shpigler et al, 2025). Strong perturbations generally produce well-separated profiles across all distance measures, whereas weak or subtle perturbations can yield nuanced shifts that are difficult to capture, especially for metrics sensitive to noise or dominated by irrelevant features (Chandrasekaran et al, 2020). In such cases, cosine similarity and Mahalanobis distance, particularly when combined with dimensionality reduction or robust feature selection, can enhance the detection of subtle yet biologically meaningful phenotypic changes. Thus, careful selection and application of distance-based metrics is essential, tailored to the data structure and resolution of biological effects under investigation.

# Public datasets and benchmarking

Given the technical complexity of imaging-based profiling, systematic and unbiased evaluation procedures are essential to assess the performance, reliability, and biological relevance of image-based profiling pipelines and algorithms (Celik et al, 2024; Kraus et al, 2025). These evaluations generally focus on two aspects of image-based profiling: (1) evaluating the performance of specific steps or components in an analysis pipeline (e.g., batch correction, feature extraction/representation learning, image-quality enhancement, etc.) and (2) quantifying the accuracy of inferred biological relationships such as gene–gene or drug–gene interactions and the "perturbative maps" derived from analyzing image-based profiles (Celik et al, 2024; Kraus et al, 2025; Ewald et al, 2025). As models and methods proliferate, comparing methods will guide researchers toward the most effective solutions for their objectives. Benchmarks will also foster standardization, thereby facilitating data and pipeline sharing, integrating findings across studies, and enabling large-scale meta-analyses (Celik et al, 2024; Arevalo et al, 2024).

## The state of benchmarking in image-based profiling

### Availability of benchmarking datasets

The availability of large, FAIR-compliant (Wilkinson et al, 2016) datasets has significantly accelerated progress in method development for image-based profiling (Dataset EV1). These datasets not only serve as real-world benchmarks for testing, optimizing, and validating computational methods, but also enable software developers to assess tool performance across diverse, real-world experimental conditions. The richness and complexity of publicly available image-based profiling data also drive the development of domain-specific solutions for advanced tasks such as single-cell analysis, batch harmonization, and machine learning–based phenotype classification.

Recently published datasets such as RxRx (Fay et al, 2023), JUMP-CP (Chandrasekaran et al, 2023), CPJUMP1 (Chandrasekaran et al, 2024b), and EU-OPENSCREEN (Wolff et al, 2025) have provided large, annotated datasets that can be used to train and evaluate models (Dataset EV1). In particular, the JUMP-CP dataset includes sentinel plates, referred to as Target 2 plates, which were imaged in every single batch across multiple laboratories, making them valuable resources for evaluating and benchmarking batch correction methods. However, the substantial size of these datasets (over 100TB) challenges the accessibility of computational resources. This has motivated the recent development of compressed benchmarking datasets such as RxRx3-Core (Kraus et al, 2025). These large resources also release embeddings for single cells or perturbations, which are substantially smaller and more manageable than the raw microscopy images.

Additionally, such large datasets may contain hidden technical artifacts that complicate the interpretability of benchmark metrics. For example, the open reading frame (ORF) subset of the CPJUMP1 dataset (Chandrasekaran et al, 2024b; Pahl et al, 2023) shows strong well-position effects that are confounded with perturbation effects (as a given ORF is in the same position across all replicates). Furthermore, the majority of ORFs were not profiled across multiple experimental batches, further confounding batch-level effects with the effects of individual ORFs (Chandrasekaran et al, 2024b). These benchmark datasets also generally lack standardized train-validation-test splits, which further introduces variance in benchmark metrics and potential bias in estimation of model generalization if train-test splits are not carefully constructed.

Many benchmarking tasks, such as evaluating the accuracy of predictions regarding gene–gene interactions, drug-target interactions or identifying specific perturbation classes, require the availability of ground truth annotations. The reliability of any image-based profiling benchmark is inherently limited by the accuracy and completeness of the ground truth data it utilizes. In practice, these annotations are often sourced from databases of protein-protein interactions, protein complexes (e.g., CORUM (Giurgiu et al, 2018), String (Szklarczyk et al, 2022), Reactome (Milacic et al, 2023)), gene ontology (Aleksander et al, 2023), and other gene set annotations or public databases of small molecules (e.g., ChEMBL (Gaulton et al, 2012)). However, such labels may suffer from false positives, for example, when perturbations may not induce a detectable phenotype in the assayed context, or false negatives, for example, when interactions have simply not been assayed. Recognizing this bottleneck, community efforts like the

Broad Drug Repurposing Hub (Corsello et al, 2017) and European Chemical Biology Database (Škuta et al, 2025) have become instrumental in providing more deeply curated and reliable annotations for model validation. When integrated with benchmarks, ground truth labels will accelerate image-based profiling method development. However, new strategies are needed to generate and validate ground truth annotations, potentially integrating orthogonal omics data or establishing community annotation projects to expand the scope of evaluation tasks available (Celik et al, 2024).

### Benchmarking traditional features against deep learning

The advent of deep learning has introduced powerful new paradigms for feature extraction in imaging-based profiling. For a full discussion on deep learning in image-based profiling, refer to the section "Advances in deep learning for image-based profiling". Briefly, self-supervised or semi-supervised methods such as DINO (Caron et al, 2021), MAE (He et al, 2022; Kraus et al, 2024), and SimCLR (Chen et al, 2020) learn complex feature representations directly from microscopy images (Doron et al, 2023). Several studies have benchmarked the performance of deep learning or SSL methods, including against traditional CellProfiler (Carpenter et al, 2006) features. These benchmarks cover a variety of tasks, including assessing the clustering of perturbations that are known to have similar MoAs or phenotypic effects and assessing the predictive power of learned features on various downstream biological tasks, such as the classification of drug targets or MoAs of small molecules.

Celik et al, (2024) proposed a general framework (EFAAR) describing a standardized pipeline for building perturbative maps from phenotypic profiles, and proposed several metrics for evaluating the quality of perturbative maps produced by such pipelines (Celik et al, 2024). Subsequently, the same group released a benchmarking dataset, RxRx3-Core, along with a reference implementation of their benchmarking approach. Kraus et al evaluated several proprietary MAE models against traditional CellProfiler (Carpenter et al, 2006) features and found that the MAE models are better at recovering drug-target interactions and separating perturbations from negative controls (Kraus et al, 2025).

In contrast, Ewald et al (2025) compared CellProfiler features, a convolutional neural network trained specifically on Cell Painting images, and a pretrained self-supervised vision transformer model. They evaluated these methods on predicting cytotoxicity and mode-of-action of small molecules with known liver toxicity in primary human hepatocytes, finding relatively similar performance across methods for predicting assay endpoints (Ewald et al, 2025). Tomkinson et al evaluated CellProfiler features against DeepProfiler features in predicting 15 different single-cell phenotypes labeled by the MitoCheck consortium (Neumann et al, 2010), and found that combining the two feature spaces yielded the best results for 9/15 labels (Tomkinson et al, 2024).

Finally, Frey et al (2025) compared CellProfiler features against DeepProfiler (pretrained CNN) (Moshkov et al, 2024) and a DINO ViT model (Caron et al, 2021) on distinguishing six mechanisms of cell death induced by 50 different small molecules at the single-cell level (Frey et al, 2025). Frey et al found that features derived from CellProfiler and DeepProfiler generally outperformed features derived from the DINO ViT when evaluated on classification accuracy and F1 score of predicting the mechanism of cell death,

but the DINO ViT features were better at capturing single-cell heterogeneity and resolving perturbation-specific or dose-dependent effects, which may account for the lower accuracy of such features on the coarse classification task (Frey et al, 2025).

### Benchmarking batch correction methods

Celik et al (2024) also evaluated the effectiveness of total variation normalization (TVN) as a batch correction method against simple centering and scaling methods and found that TVN improved both separation of perturbations from negative controls and recall of known gene–gene interactions (Celik et al, 2024). Beyond PCA and TVN-based approaches, Arevalo et al (2024) compared seven batch correction techniques developed for scRNA-seq in five different scenarios on the JUMP-CP dataset (Chandrasekaran et al, 2024b) and found that Harmony performed the best of all methods benchmarked in their study (Arevalo et al, 2024).

### Establishing common evaluation metrics

A critical gap in establishing common benchmarks in image-based profiling is the need for standardized evaluation tasks and metrics. A variety of metrics are currently employed, but there remains a lack of consensus on which metrics are most appropriate for evaluating the different aspects of performance. The different aspects include the ability of image-based profiled derived representations to capture biologically relevant information, measures of replicate consistency and signal strength, and improvements to image quality for generative modeling. We summarize commonly used metrics, illustrating this variety, in (Dataset EV2).

Importantly, even when there is consensus on an evaluation metric (e.g., mAP for evaluating the ability of representations to capture known relationships between perturbations), specific technical choices in the evaluation protocol can significantly influence the conclusion of evaluation metrics. These protocol details should therefore also be carefully described, or ideally, standardized within the field.

Benchmarking efforts are crucial for advancing the field of imaging-based profiling by providing researchers with evidence-based guidance on the selection of appropriate computational methods for their specific needs. Recent efforts combined a dataset designed for evaluation of a specific benchmarking task (predicting drug-target interactions), with reference embeddings and benchmarking code, have greatly advanced the current state of benchmarking in image-based profiling (Kraus et al, 2025). Future research should focus on more comprehensive benchmarking studies that assess the combined impact of different method choices across the entire analysis pipeline. There is also a need to drive the adoption of standardized metrics and evaluation protocols within image-based profiling to facilitate comparison of methods. Lastly, as deep learning-based methods become increasingly prevalent, addressing the interpretability and explainability of these approaches will be a critical area for future benchmarking efforts.

## Software for image-based profiling

Image-based profiling is advancing rapidly. This advance is driven by the convergence of open science initiatives, the adoption of

FAIR (Findable, Accessible, Interoperable, and Reusable) data principles, robust software engineering practices, and the widespread use of open-source tools and platforms (Wilkinson et al, 2016). This collective emphasis on transparency, availability, and methodological rigor has significantly accelerated the development and dissemination of computational tools, reshaping how image-based profiling data are generated, processed, and interpreted. Equally transformative is the community-led momentum toward standardizing data formats and analytical workflows, an essential step for improving interoperability, reproducibility, and scalability across diverse experimental contexts.

Standardized data formats are essential for processing image-based profiling datasets. The wide variety of imaging platforms and analytical tools, ranging from open-source frameworks such as CellProfiler (Carpenter et al, 2006) and DeepProfiler (Moshkov et al, 2024) to proprietary commercial systems like Revvity's Harmony and Molecular Devices' IN Carta, produce a heterogeneous landscape of image analysis options and data outputs. This diversity underscores the critical need for common, well-documented formats that support transparent data exchange, reduce the burden of format conversion, and facilitate seamless integration with downstream bioinformatics pipelines. In this context, the adoption of open standards such as OME-TIFF (Leigh et al, 2017) and OME-Zarr (Moore et al, 2023) has been transformative. OME-Zarr, with its chunked, cloud-native architecture, is particularly well-suited for large-scale image-based profiling, enabling efficient, on-demand access to subsets of high-volume datasets. Community-driven initiatives such as the Cell Painting Gallery are actively converting legacy datasets to OME-Zarr, improving both usability and alignment with FAIR principles (Weisbart et al, 2024). Complementary efforts to standardize morphology feature-level numerical data include tools like CytoTable, which convert extracted morphological features from a variety of image analysis outputs into Apache Parquet, a highly efficient, columnar storage format optimized for processing profiles (Bunten et al, 2025).

In parallel with efforts to standardize data formats, there is growing momentum to develop software toward best practices for processing image-based profiles. A typical analysis pipeline following feature extraction involves several interconnected steps, including metadata annotation, normalization, single-cell to well-level aggregation, batch correction, and feature selection, that benefit from modular, reproducible implementations (Caicedo et al, 2017). General-purpose frameworks such as Pycytominer (Serrano et al, 2025) and BioProfiling.jl (Vulliard et al, 2022) have been instrumental in this regard. Pycytominer, a Python-based toolkit, offers a flexible, well-documented API that supports data ingestion from tools like CellProfiler (Stirling et al, 2021) and commercial platforms, enabling researchers to build customizable and reproducible workflows (Serrano et al, 2025). It provides core functionality for data aggregation, annotation, normalization, and feature selection, along with utility functions such as batch effect correction. Complementing this, BioProfiling.jl, developed in Julia for high-performance computing environments, also offers an end-to-end solution for compiling, and analyzing morphological profiles (Vulliard et al, 2022). It incorporates robust methods for noise reduction, normalization, and statistical testing, including Hellinger distances as well as permutation tests and facilitates integration with external biological data sources such as molecular targets.

Building on these foundations, tools tailored specifically for single-cell morphological profiling have emerged to address the limitations of population-averaged analyses by preserving cellular heterogeneity. The Python package scmorph enables scalable single-cell analysis with features including interpretable batch correction (via scone (Cole et al, 2019)), nonlinear feature selection, and trajectory inference for dynamic biological systems (Wagner et al, 2025). Compatible with outputs from tools like CellProfiler and integrated into the scverse ecosystem (e.g., AnnData, scanpy), scmorph supports seamless incorporation into established single-cell workflows (Virshup et al, 2023, 2021; Wolf et al, 2018). Similarly, SPACe provides a streamlined and efficient pipeline for single-cell analysis of Cell Painting data, using AI-based segmentation, a curated set of biologically interpretable features, and signed earth mover's distance to capture phenotypic variation across individual cells (Stossi et al, 2024; Rubner et al, 2000). Notably, SPACe is optimized for computing resources, capable of running efficiently on consumer-grade hardware, making it especially valuable for labs with limited computational resources. coSMicQC takes a different approach by conducting single-cell quality control by leveraging morphological features from single-cell image-based profiles to identify technical outliers like segmentation errors and potential mycoplasma contamination (Tomkinson et al, 2025). Progress in this area was made possible by the pioneering efforts of HC StratoMineR (Omta et al, 2016) and PhenoRipper (Rajaram et al, 2012), which offered intuitive, web-based platforms for high-content data analysis. For example, designed to support users across a range of expertise levels, HC StratoMineR provided an end-to-end workflow encompassing data filtering, quality control, dimensionality reduction, hit picking, and clustering (Omta et al, 2016; Sexton et al, 2023).

Workflow management is essential for ensuring transparency, reproducibility, and scalability in the complex, multi-step pipelines typical of image-based profiling (Wratten et al, 2021; Stoudt et al, 2021). The integration of workflow management systems (WMS) such as Snakemake (Köster and Rahmann, 2012) and Nextflow (Di Tommaso et al, 2017) has orchestrated analyses across diverse computational environments, from local workstations to high-performance clusters and cloud-based infrastructures (Wratten et al, 2021). These systems formalize each analytical step, manage software dependencies, track data provenance, and support automation, which collectively reduces user error and enhances reproducibility (Stoudt et al, 2021). However, there remains a lack of widely adopted workflows specifically tailored to the unique demands of image-based profiling. Recent efforts, such as those documented in the Image-based profiling handbook (Cimini et al, 2019) have begun to address this gap by outlining orchestrated workflows that process raw microscopy images into structured morphological profiles (Cimini et al, 2019). At this time, efforts to develop processing workflows are underway to lower the barrier to entry, enabling non-experts to process image-based profiles and engage with the broader community.

Despite the progress of open-source tooling, community standardization, and FAIR data, challenges remain. For example, chaining together image-based profiling steps into a pipeline still requires bespoke programming, and this lack of domain-specific workflow management implementations limits access and reproducibility (Ziemann et al, 2023; Keefe et al, 2023). Moreover, there is no standardized benchmarking suite for processing tools (Way et al, 2022b; Arevalo et al, 2024). Addressing these issues will require coordinated efforts in workflow implementation, software

**Box 2:  Solving challenges for the next decade of image-based profiling**

*Next decade of challenges for image-based profiling. This box outlines the core challenges that the image-based profiling community must address to drive innovation over the next 10 years*

We have identified six core challenges that will shape the trajectory of image-based profiling research over the next decade. These include:

1. **Developing end-to-end workflows**.

   A core emerging challenge, driven by the rapid proliferation of new tools and complex data types, is the development of modular and interoperable frameworks for constructing, reproducible, end-to-end, workflows for image-based profiling. Despite the growing ecosystem of open-source tools, there remains a lack of flexible, end-to-end solutions for executing the full bioinformatics pipeline, which starts from raw microscopy image acquisition and ends in interpretable image-based profiles. However, assembling such end-to-end workflows is challenging because of poor interoperability across a fragmented ecosystem of tools that often requires extensive, bespoke, human-in-the-loop, and error-prone programming (Djaffardjy et al, 2023; Hu et al, 2021). This challenge is further compounded by the emergence of temporal, 3D, and virtually stained datasets, each introducing additional complexity through unique data formats, modality-specific processing requirements, and a lack of harmonized computational approaches (Arora et al, 2023). This topic is gaining urgency not only because of the challenges workflows would solve, but also because recent advances now make solutions attainable. Foundational tools such as Pycytominer (Serrano et al, 2025), BioProfiling.jl (Vulliard et al, 2022), scmorph (Wagner et al, 2025), and DeepProfiler (Moshkov et al, 2024) are maturing, and community resources such as the Image-Based Profiling Handbook (Cimini et al, 2019) provide valuable best-practice guidelines. The widespread availability of FAIR-compliant (Wilkinson et al, 2016) datasets also creates ideal conditions for testing and validating new workflows. Solving this challenge will not only streamline high-throughput image-based profiling for experts and non-experts alike, but will also enhance reproducibility, promote interoperability, and enable more scalable, integrated analyses. A community-driven focus on creating modular, interoperable, and well-documented tools for building these pipelines can empower broader participation, reduce redundant efforts, and accelerate both methodological development and biological insight.

2. **Fortifying methodological infrastructure for temporal, 3D, and virtual staining imaging modalities**.

   Most image-based profiling today uses static 2D imaging. This is rapidly changing, however, as recent advances in microscopy hardware, assays (e.g., live-cell painting), generative AI models, and scalable computation are making these complex modalities now feasible for high-throughput profiling. Cellular processes are inherently dynamic and spatially structured (Eismann et al, 2020; Garcia-Fossa et al, 2025). For example, temporal image-based profiling will capture cell dynamics, revealing processes such as cell division, migration, and differentiation (Hur et al, 2024; Alieva et al, 2023). 3D image-based profiling enables more physiologically relevant analysis in tissues and organoids (Liu et al, 2024a; Edwards et al, 2020; Chelebian et al, 2025). Furthermore, virtual staining is a promising strategy that aims to translate low-cost, low-phototoxicity, label-free images (e.g., brightfield) into fluorescence-like images. While this prediction only approximates the ground truth offered by real reagents, a primary motivation is to make these data compatible with the vast ecosystem of analysis tools and pre-trained models originally developed for fluorescence microscopy (Ichita et al, 2025). These methodological frontiers present unique computational and infrastructure challenges, including the need for high-quality cell tracking algorithms, accurate 3D segmentation, new opportunities for high-content featurization, and large annotated training sets for virtual staining. The image-based profiling field is in a good position to solve these challenges, through the development of scalable, accessible, and standardized computational strategies for these rich data types. Doing so will unlock a more nuanced view of cell behavior, reduce experimental burden, and enable more powerful models of health and disease.

3. **Bridging profiles to mechanisms through interpretable features**

   The interpretability of morphology features, ranging from classical hand-crafted measurements to those derived from deep learning, remains a key challenge in image-based profiling (Garcia-Fossa et al, 2023). This is an emerging topic as a critical bottleneck because the field's rapid adoption of powerful deep learning models, which often act as "black boxes," has outpaced our ability to interpret what they are learning. While the field has been highly productive using classical features that are not always directly interpretable, moving from powerful descriptive profiles to robust explanatory frameworks requires a deeper understanding of what features represent (Chandrasekaran et al, 2020; Moshkov et al, 2024; Garcia-Fossa et al, 2023). Classical features, though defined, are generated in large numbers, with many being noisy, redundant, or susceptible to technical artifacts that obscure biological signals (Murthy et al, 2024; Caicedo et al, 2017). Deep learning features pose an even greater interpretability barrier, typically manifesting as "anonymous latent variables" with no explicit meaning nor obvious mapping from trained model to trained model. This black-box nature makes it difficult to link phenotypic profiles to specific biological mechanisms (Seal et al, 2024a; Foroughi Pour et al, 2022). The core challenge lies in developing and validating methods to reveal what these complex models are capturing, disentangling biologically meaningful variation from technical noise. However, it is crucial to recognize that the potential for mechanistic understanding may be fundamentally constrained by the input data itself. The low-resolution or static 2D images common in large-scale screens may not contain the granular information needed to explain dynamic cellular processes, a limitation that improved feature interpretation alone cannot overcome (Gordonov et al, 2016; Yoshida et al, 2023; Chandrasekaran *et al*, 2020). The field is scaling to increasingly complex datasets, including 3D, temporal, and virtually stained modalities, which may provide sufficiently interpretable, but they are generating high-dimensional data with minimal intuitive guidance without additional methodology layered on. Explainable AI methods such as GradCAM (Longo et al, 2024) and attention-based visualization (Vaswani et al, 2017; Pfaendler et al, 2023; Doron et al, 2023; Morelli et al, 2025) are gaining traction in image-based profiling contexts, helping to clarify how models make phenotypic distinctions. Addressing this challenge will be highly valuable, as it will increase confidence in AI-driven findings and enable mechanistically grounded hypothesis generation. Ultimately, combining more interpretable features with richer data from high-resolution or time-lapse imaging offers a transformative path to shift image-based profiling from a powerful descriptive method toward a robust explanatory framework.

4. **Establishing rigorous quality control standards**.

   Quality control (QC) is an underdeveloped area in image-based profiling. This challenge is emerging now as the field's focus is shifting from population-level analyses to high-resolution single-cell resolution. QC removes poor images and poorly segmented single cells to improve biological interpretations - without QC, findings may be spurious or misleading. QC is often omitted for population-level analyses using robust statistics (Cimini et al, 2023). Currently, there is no universally accepted set of QC criteria for either whole-image or single-cell level assessments. Existing QC implementations are frequently ad hoc, relying on dataset-specific heuristics and often requiring substantial hands-on time and computational resources (Qiu et al, 2020). However, recent advances such as coSMicQC (Tomkinson et al, 2025) and SPACe (Stossi et al, 2022), demonstrate that systematic QC not only improves signal detection but also enhances the reliability of downstream analyses. Importantly, beyond being a technical necessity for single-cell analysis, accurate segmentation preserves the integrity of the cell as the fundamental unit of function (Marks et al, 2025). This enables single-cell representations that capture biologically meaningful variability and foster interpretable connections between morphology and mechanism (Chen and Murphy, 2023). As imaging data scales in size and complexity, and as analyses increasingly leverage single-cell

over bulk profiles, the volume of low-quality segmentations and imaging artifacts will rise, underscoring the need for streamlined and reproducible QC protocols. The field is now well-positioned to define and adopt practical, generalizable QC standards that can be integrated directly into end-to-end workflows. Doing so will elevate confidence in morphological profiling outputs, facilitate robust cross-study comparisons, and ensure the integrity of data used to drive biological discovery.

5. **Building systematic benchmarking frameworks**.

As image-based profiling methodology evolves, establishing systematic benchmarking practices is essential for users to select and optimize methods, and to ensure methodological reliability and biological relevance. This challenge has become particularly urgent, as the recent release of massive, high-quality public datasets now provides the common ground necessary to compare the concurrent explosion of new computational methods and deep learning models. A wide array of evaluation approaches have been proposed, but the lack of standardized evaluation metrics, well-defined benchmarks, and high-quality ground truth annotated data has limited comparability and practical deployment. The tide is beginning to turn here with recent benchmarks signaling the field's commitment to standardizing evaluation and accelerating innovation (Chen et al, 2023). Early public datasets contain confounding factors, and computational demands often hinder reproducibility. Fortunately, a wave of new datasets (see section "Public datasets and benchmarking") is addressing this. Resources like Recursion's RxRx datasets collection (Sypetkowski et al, 2023; Fay et al, 2023; Kraus et al, 2025), JUMP Cell Painting (Chandrasekaran et al, 2023), and the Cell Painting Gallery (Weisbart et al, 2024) providing higher-quality labels for model training and validation. Frameworks like EFAAR (Celik et al, 2024) also offer structured ways to assess and compare models. This is a timely opportunity for the community to coalesce around shared metrics, datasets and annotation standards. Doing so will establish common evaluation standards, facilitate fair model comparisons, encourage robust method development, and ultimately guide researchers toward best-in-class tools for their specific needs.

6. **Reaping the rewards of AI and foundation models, while accounting for high costs**

A major frontier and growing success in image-based profiling is the rapid adoption and evolution of deep learning models, which have moved beyond proof-of-concept to become common in the field. Increasingly, these models are being used to scale profiling efforts, improve generalizability across experiments, adapt to varying fluorescent channel configurations, and enhance the interpretability of complex cellular phenotypes (Pratapa et al, 2021; Tang et al, 2024). The next decade will see similar advances using AI. Furthermore, foundation models trained on large, diverse imaging datasets are now being actively developed and deployed, reflecting a shift toward more robust, flexible, and general-purpose AI systems for image-based profiling (Kraus et al, 2024; Kenyon-Dean et al, 2024; Gupta et al, 2024; Ji et al, 2024). This topic is emerging as a critical frontier now because the field has finally overcome two major historical barriers that previously constrained progress: the lack of massive-scale public datasets and the high cost of computation. The recent emergence of large, standardized datasets (such as the JUMP-CP and RxRx) and more accessible high-performance computing are the specific advances that have directly enabled the development of these powerful, data-hungry models. In turn, the very success of these models now acts as a powerful trigger for further data standardization and sharing, as the community recognizes the value of combining resources to build even more generalizable AI. Continued advancement will democratize access to state-of-the-art models, support reproducible and interpretable analyses, and empower researchers to derive deeper, more transparent insights from image-based profiling data.

# Future challenges

Over the past two decades, image-based profiling has evolved from a niche technique into a cornerstone of high-throughput, quantitative cell biology. Initially catalyzed by the foundational standards proposed by the nascent CytoData society, as captured in Caicedo et al (2017), the recent transformation has been driven by advances in computational methods, including deep learning, and a shift toward community-driven open source science. These developments have extended the reach and impact of image-based profiling across increasingly diverse research areas and applications.

We have identified a set of core challenges in image-based profiling that require focused attention from our field (Box 2). As image-based profiling scales in scope and impact, its long-term utility depends on the development of processing methods that are reliable, interpretable, and reproducible across diverse contexts. The decade ahead holds immense promise for image-based profiling, but realizing that promise depends on our collective resolve to confront and overcome the challenges that remain. Standardizing and managing end-to-end workflows, embracing the richness of temporal, 3D, and virtual staining modalities, ensuring the interpretability of features, establishing rigorous benchmarking and quality control standards, and advancing AI for greater scalability, adaptability, and interpretability are not isolated goals. Together, they form a cohesive foundation for the next generation of image-based

engineering, and community training to build a more robust and reproducible ecosystem.

profiling and new frontiers for biological discovery in general. Achieving this vision will require the continued dedication of researchers, software engineers, and cross-disciplinary consortia working in unison. By fostering innovation through open collaboration, transparency, and reproducibility-aware design, the field can evolve from a powerful profiling technique into an intelligent, unified system for understanding cell biology. One that not only captures the complexity of cellular phenotypes with unprecedented fidelity, but also empowers researchers to uncover hidden patterns, generate robust hypotheses, and drive meaningful breakthroughs in health and disease. Now is the moment to act. With sustained momentum, image-based profiling will redefine how we explore, understand, and ultimately transform biology.

# Peer review information

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

## Acknowledgements

We would like to thank the CytoData Society for their support and contributions to the development of this paper. Research reported in this publication was supported by the National Library of Medicine of the NIH under award number T15LM009451 to ES and 5T15LM007359 to JP. Funding support also by National Institutes NIH grant R35 GM122547 to AEC, the Human Frontier Science Program (RGY0081/2019 to SS), The Gilbert Family Foundation (923014 to GPW), Alex's Lemonade Stand Foundation "A" Award and Tap Cancer Out (Grant # 23–28306 to GPW), American Heart Association Collaborative Sciences Award (24CSA1255857 to GPW), the Medical Research Council (MC_ST_00035 to JW), (MR/Ro15635/1 to NOC), the National Science Foundation (Grant No. 2348683 to JCC), the Chan Zuckerberg Initiative grant (DAF2021-225261, https://doi.org/10.37921/644085ggkbos to GM), an advised fund of Silicon Valley Community Foundation (funder https://doi.org/10.13039/100014989 to GM).

## Author contributions

**Erik Serrano**: Conceptualization; Data curation; Formal analysis; Investigation; Visualization; Methodology; Writing—original draft; Project administration; Writing—review and editing. **John Peters**: Investigation; Visualization; Methodology; Writing—original draft; Writing—review and editing. **Jesko Wagner**: Investigation; Visualization; Writing—original draft; Writing—review and editing. **Rebecca E Graham**: Investigation; Visualization; Writing—original draft; Writing—review and editing. **Zhenghao Chen**: Investigation; Writing—original draft. **Brian Y Feng**: Investigation; Writing—original draft. **Gisele Miranda**: Investigation; Writing—original draft. **Alexandr A Kalinin**: Investigation; Writing—original draft. **Loan Vulliard**: Investigation; Writing—original draft. **Jenna Tomkinson**: Investigation; Writing—original draft; Writing—review and editing. **Cameron Mattson**: Investigation; Writing—original draft. **Michael J Lippincott**: Investigation; Writing—original draft. **Ziqi Kang**: Investigation; Writing—original draft. **Divya Sitani**: Investigation; Writing—original draft. **Dave Bunten**: Investigation; Writing—original draft. **Srijit Seal**:

Investigation; Writing—original draft. **Neil O Carragher**: Supervision; Writing—review and editing. **Anne E Carpenter**: Conceptualization; Resources; Project administration; Writing—review and editing. **Shantanu Singh**: Conceptualization; Resources; Project administration; Writing—review and editing. **Paula A Marin Zapata**: Conceptualization; Investigation; Writing—original draft. **Juan C Caicedo**: Conceptualization; Resources; Supervision; Investigation; Writing—original draft; Project administration; Writing—review and editing. **Gregory P Way**: Conceptualization; Resources; Supervision; Funding acquisition; Visualization; Methodology; Writing—original draft; Project administration; Writing—review and editing.

## Disclosure and competing interests statement

NOC is co-founder, shareholder, and management consultant for PhenoTherapeutics Ltd. SS and AEC serve as scientific advisors for companies that use image-based profiling and Cell Painting (AEC: Recursion, SyzOnc, Quiver Bioscience, SS: Waypoint Bio, Dewpoint Therapeutics, Deepcell) and receive honoraria for occasional scientific visits to pharmaceutical and biotechnology companies. The remaining authors declare no competing interests.

