## [Peer Review File · Molecular Systems Biology]

Progress and new challenges in image-based profiling

Erik Serrano, John Peters, Jesko Wagner, Rebecca Graham, Zhenghao Chen, Brian Feng, Gisele Miranda, Alexandr Kalinin, Loan Vulliard, Jenna Tomkinson, Cameron Mattson, Michael Lippincott, Ziqi Kang, Divya Sitani, Dave Bunten, Srijit Seal, Neil Carragher, Anne Carpenter, Shantanu Singh, Paula Marin Zapata, Juan Caicedo, and Gregory Way

Corresponding author(s): Gregory Way (Gregory.way@cuanschutz.edu)

Review Timeline:

Submission Date:	28th Jul 25
Editorial Decision:	15th Sep 25
Revision Received:	13th Nov 25
Editorial Decision:	12th Dec 25
Revision Received:	15th Dec 25
Accepted:	16th Dec 25

Editor: Jingyi Hou

Transaction Report:

15th Sep 2025

Manuscript Number: MSB-2024-12819

Title: Progress and new challenges in image-based profiling

Author: Erik Serrano

John Peters

Jesko Wagner

Rebecca Graham

Zhenghao Chen

Brian Feng

Gisele Miranda

Alexandr Kalinin

Loan Vulliard

Jenna Tomkinson

Cameron Mattson

Michael Lippincott

Ziqi Kang

Divya Sitani

Dave Bunten

Srijit Seal

Neil Carragher

Anne Carpenter

Shantanu Singh

Paula Marin Zapata

Juan Caicedo

Gregory Way

Dear Greg,

Thank you for submitting your interesting and timely Review article to Molecular Systems Biology. We have now received reports from three reviewers who agreed to evaluate your manuscript.

All three reviewers found your manuscript to be relevant, timely, and very well-written. However, they also raised several points and suggestions that could help further strengthen the paper. We strongly encourage you to carefully consider all the feedback during your revision, while leaving it to your discretion whether or not to incorporate specific suggestions.

On a more editorial level:

1. Please provide the manuscript file in .docx format. All figures should be removed from the manuscript and uploaded as separate figure files, with figure legends placed below the References section in the manuscript file. Our graphic designer will redraw the figures after the revision is complete.
2. Include up to five keywords in the manuscript file.
3. Remove the "Authors' Contributions" section from the manuscript.
4. Add the missing figure callouts for Figure 2C and Figures 3A-C.
5. Tables 2 and 4 should be renamed to Table EV1 and EV2, respectively, with updated callouts in the manuscript. These should be uploaded as Expanded View Content.
6. Rename the "Conflict of Interest" section to "Disclosure and Competing Interests Statement."
7. Ensure that tables in the manuscript are listed in consecutive order as Table 1, 2, 3, with appropriate callouts in the text.
8. Correct the section order in the manuscript to the following:
Title Page - Abstract - Keywords - Introduction - Acknowledgements (if any) - Disclosure and Competing Interests Statement - References - Figure Legends - Table(s)

When submitting a revised version of your manuscript, please attach a covering letter giving details of the way in which you have handled each of the points raised by the referees. A revised manuscript will be once again subject to review.

I look forward to receiving your revised manuscript soon.

Kind regards,
Jingyi

Jingyi Hou, PhD
Senior Editor
Molecular Systems Biology

*** PLEASE NOTE *** As part of the EMBO Press transparent editorial process initiative (see our Editorial at <https://dx.doi.org/10.1038/msb.2010.72>), Molecular Systems Biology publishes online a Review Process File with each accepted manuscripts. This file will be published in conjunction with your paper and will include the anonymous referee reports, your point-by-point response and all pertinent correspondence relating to the manuscript. If you do NOT want this File to be published, please inform the editorial office at contact@molsystbiol.org within 14 days upon receipt of the present letter.

Reviewer #1:

Summary & General remarks

The authors present a comprehensive review of the data analysis approaches and challenges for image-based profiling. Since the last in-depth review by Caicedo et al. (2017), numerous new deep learning approaches have significantly improved this field. Thus, this review is important and timely. Overall, the review is well written and rests on a large and highly relevant corpus of papers. There is a clear structure. Although the formatting of the headlines is broken and needs to be fixed.

The manuscript is of very high quality. However, I still have a number of points where I would like to engage in a discussion with the authors. Therefore I would like to ask the authors for comments and revisions.

Major points

Section II

Experimental design and execution for image-based profiling

Lack of discussion of assay quality control. The authors have a section "Experimental design and execution for image-based profiling". There is very little mention of comprehensive and/or quantitative QC on the data acquisition side. Of course, this is not the most important section for this review. However, I believe data quality is highly relevant for analysis overall, and good QC starts at the acquisition. The only section kind of dedicated to it is "Whole-image QC" - which arguably deals only with filtering data - but one should create good data in the first place. In any case this addresses only the microscopy-image acquisition issues, while important, in our experience, they are the least pressing issues in a high-content screen. I think it is important to mention the usage of heatmaps to control for plate artefacts. The same goes for the assessment of batch effects with at least mentioning the usage of UMAPs in this context. No need for an in-depth discussion, but I would define these as very minimal requirements for any profiling assay.

Feature extraction

There is an inconsistency when the authors discuss the feature interpretability of classical features. In the comparison table (Classic approach vs Deep learning approach), they assess: "Interpretability: This mathematical approach ensures each feature is well-defined, enabling a clear understanding of how features are measured. Still, this does not always translate directly to biological meaning". In the text under section "Feature extraction," the Authors write a bit differently: "These features capture biologically important characteristics, reflecting single-cell morphological signatures. These classical approaches have been instrumental in establishing image-based profiling as a scalable and interpretable strategy". The first definition given in the table is much more balanced and accurate, as many features - e.g., texture features - have been defined by computer scientists to solve more generic computer vision tasks. Plus, even if they are mathematically well-defined, most of them lack clear interpretability. I would suggest to tone down the claim that image based profiling with classical features is a clear, interpretable strategy.

Feature selection.

Feature selection is in our hands one of the most problematic and non-reproducible aspects of the classical analysis workflow. It's great that the authors highlight this. Here, I want to mention also a different approach that I came across (disclosure: the reviewer is NOT an author of any of these papers): Christoforow et al. 2019 defined their feature set on technical reproducibility by performing 2 biological repeats of a plate with reference compounds and including features that showed ≥ 0.8 similarity. As this analysis approach has been quite productive in the past, I think it merits a discussion (Christoforow et al. 2019, Schneidewind et al. 2020, Akbarzadeh et al. 2022).

Section IV

Only single-cell image-based profiling has its separate figure. Maybe the other aspects in this section would profit also from illustrations.

Temporal and 3D image-based profiling

"Lastly, advances in 3D microscopy techniques, such as lattice light-sheet microscopy, Oblique Plane Microscopy, and other super-resolution methods, have made imaging in three dimensions more advantageous and information-rich than ever before (Chen et al, 2014)." For the image-based profiling field, large-scale datasets have been central. The lack of throughput of these microscopy methods is still an issue and an active field of research and development. I have not seen convincing systems on the market yet. I feel that this needs to be qualified.

Virtual staining for image-based profiling

I am confused by this section, and I do not fully understand the main focus on virtual staining. There is a real push in the community towards establishing label-free approaches, using brightfield imaging. While I agree that virtual staining, i.e., the prediction of fluorescent signals from the label-free images, is a valid research direction. The benefits at this point are rather unclear, as the authors also admit. This raises the question: why go through this extra step that most probably will also introduce errors or loss of information? Indeed, there are papers that show that training models on BF images directly has similar performance to training on CP images (Harrison et al. 2023). So it is questionable that virtual staining should be the main focus. I would suggest to title the section label-free and discuss it from this point of view with virtual staining as an interesting direction.

Single-cell quality control for image-based profiling

This section shares some content overlap with single-cell image-based profiling and is confusing because of the similar title, content, and, to some extent, similar goal of the text.

Batch correction for image-based profiling & Profile similarity metrics

I found these 2 sections very extensive and a bit unfocused.

The state of benchmarking in image-based profiling

A key factor that drives many analysis developments are the lack of good labels/annotations and the existing labels being problematic due to polypharmacology or dependence on factors such as cell Line, concentration, etc. It's great that the authors highlight this and raise that the community should act on improving this. They also mention it in Box 2. Although it is a bit lost under the aspect of benchmarking overall, addressing multiple attached aspects (e.g., datasets, metrics etc). This lack of good ground truth - the labels, not the datasets - is quite a bottle neck. For instance, for image analysis - i.e., image segmentation - creating good ground truth was a big accelerator.

There are some projects that could additionally be mentioned that work on improving this. For instance, the annotations collected by the Broad Drug Repurposing hub (Corsello et al. 2017) were quite instrumental for our profiling projects. Additionally, the European Chemical Biology Database (ECBD <https://ecbd.eu/> - Skuta et al. 2025) efforts by collecting all kinds of relevant information based on the EU-OPENSOURCE compounds, even negative ones, via its bioprofiling efforts could become valuable (Disclaimer: reviewer is affiliated with EU-OPENSOURCE)

Section VI: Software for image-based profiling

This only focuses on open-source software. There is also commercially available software that is widely used and useful (e.g., Revvity Harmony). I believe some commercial solutions also deserve to be at least mentioned.

Box 2: Solving challenges for the next decade

1. Developing standardized, end-to-end workflows.

I have an issue with the term "standardized workflows". Actually, I am not at all convinced why there is a need for standardized and/or universally adopted workflows. For me, standardized workflows mean that the sequence of components and/or the components of the workflow themselves would be fixed and/or the "same" across different analyses. First of all, would this not imply that all analysis challenges would be solved? Something the authors clearly disproved with the review they have written? The field seems to be still rapidly developing - a call for standardization could hamper this.

I agree to all the points about poor interoperability, fragmented ecosystem, lack of transparent reporting, etc. There are also good solutions raised by the authors, e.g., guidelines, libraries. For sure, if most of the field would agree to only use a few workflows, the technical complexity would be reduced, and naturally, these could become better documented. But that seems rather unlikely to happen, and "limiting" the workflows seems to be a very poor solution to these problems.

From an application viewpoint, it would make sense to standardize the analysis used for pre-clinical assays, e.g., Cell Painting in toxicology as a replacement of animal experiments. Because there would be a regulatory mandate (Piergiorganni et al. 2025). But I would argue this is currently only a minor part of the field.

This first section of this very interesting box leaves me unsatisfied. Sure, there are important technical issues raised, but a call for standardized analysis seems not to be a good solution for any of them. Better would be clear, transparent reporting of used workflows - whatever they might be, creating software with a focus on interoperability, benchmarking with comparable metrics and datasets, and an open discussion in the field in which frameworks/libraries to invest as community resources.

2. Fortifying methodological infrastructure for temporal, 3D, and virtual staining modalities

As above with the virtual staining section: "Virtual staining provides a promising strategy for reducing phototoxicity and experimental cost, though it only approximates ground truth offered from real reagents". Label-free imaging is providing these benefits, e.g., brightfield imaging. To me, it seems unclear if the virtual staining, i.e., the prediction of fluorescent signals from the label-free images, is actually beneficial for the analysis of label-free images. I would suggest rewriting.

3. Ensuring ontological and interpretable feature extraction across experiments and platforms

It's a very important and evocative section. But I am not entirely convinced by it.

A) Centrality of the interpretability problem. Given that the field has been quite productive, even just based on the rather weak interpretability of the classical image features. My impression is that this issue does not have a very high priority.

B) I am also a bit sceptical about the "transformative" potential of the solution to this, at least for the field of image-based profiling. The field has been driven in the past to produce very large microscopy data based on fixed cells and with low resolution. It is a bit hard for me to see how this approach could yield a strong mechanistic understanding of cellular processes - or how they would be altered when perturbed. I say this also as fields that have that expressed goal - e.g., virtual cell - seem to have a preference for high resolution and also time-lapse imaging. Thus, I believe this is also a data problem, not only an analysis issue.

Minor points:

"Single-Cell image-based profiling": "In the future, impact assessment of QC should become a standard in image-based profiling to ensure the highest quality of profiles and downstream analyses (Pratapa et al. 2021)." It's unclear what is meant by "impact assessment". The citation does not use that term.

"Optical pooled screening": "Processing OPS data builds upon foundational steps in classical image-based profiling (i.e., illumination correction, segmentation, and feature extraction) while introducing specialized components for single-cell barcode deconvolution for each cell (see Figure 1)." It's unclear what the Figure reference means - there seems to be no term "single-cell barcode deconvolution" in the figure.

"Many cell tracking algorithms exist, such as TrackMate and Ultrack (Ershov et al, 2022; Bragantini et al, 2024)". TrackMate is a software suite (Fiji plugin) and not a specific algorithm - it implements different segmentation and tracking algorithms.

The term ISS is defined twice in the first paragraph of "Optical pooled screening".

Formatting of the headlines is broken and needs to be fixed.

References:

A. Christoforow, J. Wilke, A. Binici, A. Pahl, C. Ostermann, S. Sievers, H. Waldmann
Design, Synthesis, and Phenotypic Profiling of Pyrano-Furo-Pyridone Pseudo Natural Products
Angew. Chem. Int. Ed., 58 (2019), pp. 14715-14723, 10.1002/anie.201907853

T. Schneidewind, S. Kapoor, G. Garivet, G. Karageorgis, R. Narayan, G. Vendrell-Navarro, A.P. Antonchick, S. Ziegler, H. Waldmann. The pseudo natural product myokinasib is a myosin light chain kinase 1 inhibitor with unprecedented chemotype Cell Chem. Biol., 26 (2019), pp. 512-523

M. Akbarzadeh, I. Deipenwisch, B. Schoelermann, A. Pahl, S. Sievers, S. Ziegler, H. Waldmann. Morphological profiling by means of the Cell Painting assay enables identification of tubulin-targeting compounds Cell Chem. Biol. <https://doi.org/10.1016/j.chembiol.2021.12.009>

Harrison PJ, Gupta A, Rietdijk J, Wieslander H, Carreras-Puigvert J, Georgiev P, et al. (2023) Evaluating the utility of brightfield image data for mechanism of action prediction. PLoS Comput Biol 19(7): e1011323. <https://doi.org/10.1371/journal.pcbi.1011323>

Corsello, S., Bittker, J., Liu, Z. et al. The Drug Repurposing Hub: a next-generation drug library and information resource. Nat Med 23, 405-408 (2017). <https://doi.org/10.1038/nm.4306>

Čtibor Škuta, Tomáš Müller, Milan Voršilák, Martin Popr, Trevor Epp, Katholiki E Skopelitou, Federica Rossella, Katja Herzog, Bahne Stechmann, Philip Gribbon, Petr Bartůněk, ECBD: European chemical biology database, Nucleic Acids Research, 2025; gkae904, <https://doi.org/10.1093/nar/gkae904>

Piergiovanni, M., Mennecozzi, M., Barale-Thomas, E. et al. Bridging imaging-based in vitro methods from biomedical research to regulatory toxicology. Arch Toxicol 99, 1271-1285 (2025). <https://doi.org/10.1007/s00204-024-03922-z>

Reviewer #2:

Dear Editors, Dear Authors,

The review article presented by Erik Serrano and the community around CytoData Society is a timely and state-of-the-art update to their 2017 best practice article. In my opinion, the details, intricacies, and guides for beginners in the field of image-based profiling, as well as high-content/phenotypic screening, are comprehensively covered and extended in upcoming areas of research. What I particularly liked is where the authors centered on approaches/methods/software that have not yet yielded significant improvements over the more traditional way of analyzing cellular imagery. I would also like to highlight that I noticed the authors made a great effort to describe methods and advances in an application-agnostic way, wherever possible, including referencing peripheral articles as well. Set aside that I think this is an excellent read for anybody working or being interested in the booming field of phenotypic profiling, I do have some minor comments, which I list below:

1. In Figure 1, take care that the microscope is not the picture of a specific instrument from some vendor. This should be more generalized.
 - 1.a. The same holds true for software like CellProfiler and DeepProfiler. Though these are excellent examples, there are also many other tools (as the authors explain later), but those two are highlighted particularly, and that shall be avoided.
 - 1.c. The text boxes in Figure 1 are redundant to the main text and can either be actual text boxes or cut from the figure, as well as the "downstream-analysis icon" that is too small for interpretation after all.
2. The entire Section II would benefit significantly if the learnings since the time of Caicedo 2017 et al. were highlighted. Otherwise, it seems like repeating what many people already know very well.
3. Section II a-f, in my view, need a figure with small representative examples, such that also newbies in the field can relate the text to the problem at hand and thus gain a deeper understanding of the methods applied.
4. I'd suggest clustering section IV a little bit differently, such that the different current challenges and topics connect a little bit better e.g.: a,d,e,f,c,b,g

Otherwise, congratulations on a well-written, balanced, and comprehensive review.

Reviewer #3:

Summary and general remarks

Serrano et al. survey the methodological and computational aspects of image-based profiling. The manuscript details the experimental pipeline and computational analyses followed by focusing on three areas. First, a comprehensive description of deep learning in cell profiling, where I especially enjoyed the clear description of weak- and self-supervised learning (Section 3). Second, "emerging" advances and challenges (Section 4) and third, surveying benchmarking and software tools. Given the growing number of cell profiling projects, this review is timely and valuable. However, I believe the manuscript would be strengthened by a more focused and organized structure.

Major points

This review primarily focuses on the analysis of Cell Painting data, using it as the main example throughout the text. I believe it would be beneficial to state this explicitly in the introduction to manage reader expectations. While this focus allows for a detailed discussion of aspects specific to Cell Painting (and similar phenotypic-based screens), it may not fully represent other forms of high-content cell phenotyping. This is also reflected in the terminology used (e.g., "perturbation" for experimental conditions). A broader framing of "cell profiling" would, in my opinion, require substantial revisions to accurately reflect the state of the field.

The following comments are offered as constructive suggestions to strengthen the manuscript. It is, of course, up to the authors to determine if and how these can best be implemented.

In my opinion, the review's broad scope and organization could be streamlined for greater clarity and impact. While a timely overview of the field, the current structure jumps between topics, making the flow less coherent than it could be. I suggest reorganizing the technical discussion into two main sections following the experimental and computational pipeline: one dedicated to "Current Technical Aspects and Best Practices" and a second to "Emerging Topics and the Future". Furthermore, I recommend that the authors transition from a broad survey of the literature to a more focused discussion. Synthesizing the key findings into a more coherent narrative would transform the manuscript from a list of papers into an engaging and insightful review.

To facilitate this, the section on "Current Technical Aspects and Best Practices" could be structured around key questions that are highly relevant to the readership. These topics are touched upon, but a more structured discussion would enhance the manuscript's contribution:

- * **Representations:** A comparative analysis of deep learning versus classic feature extraction would be highly valuable. This could address in what situation each approach is most effective, the trade-off in interpretability, and the magnitude of performance gains from deep learning. This discussion could also be broadened to include the distinction and implications of single-cell versus patch-based representations.

- * **Encoding Cell Heterogeneity:** A discussion on the trade-offs between averaging representations at the well level versus preserving single-cell heterogeneity. What does the current literature say about the downstream performance gained by maintaining a population-level representation?

- * **Normalization and Batch Correction:** It would be highly beneficial to discuss the impact of different normalization and batch correction methods on downstream performance. Are there significant alterations that researchers should be aware of?

- * **Best Practices for Downstream Analyses:** The authors could propose a set of "best practice" measurements for downstream analyses, such as reproducibility or MoA classification.

While some answers may not be conclusive, a structured discussion based on relevant literature would be a great contribution to the field.

For the "Emerging Topics and Future" section, a clearer justification for each topic is needed. It would be helpful to explain what makes these areas "emerging," why they are important now, and what specific advances have enabled their recent development. For example, the authors could make the case that the emergence of "foundation models" are a trigger for data sharing and data standardization.

Minor points

- * The abstract appears to focus on a few specific topics that may not fully align with the broader scope of the review. It would be helpful to ensure the abstract accurately reflects the full content and emphasis of the manuscript.

- * In the Abstract the statement, "Despite these advancements, the field still faces significant challenges requiring innovative solutions", is very broad and not specific. I recommend either specifying what these challenges are or omitting the sentence for greater conciseness.

- * "Since then, the field has advanced rapidly, driven by progress in both experimental protocols (e.g., Cell Painting (Bray et al, 2016), Live Cell Painting (Garcia-Fossa et al, 2024))". To better reflect the recent advancements in experimental protocols, I suggest citing a more contemporary screening technique, such as those involving organoids, instead of Live Cell Painting, which has not yet been widely demonstrated for practical applications. Also, Cell Painting was already referenced in the 2017 paper and thus should probably be excluded in this context.

- * **Box 1 -** To make the benefits of data sharing more concrete, it would be valuable to highlight specific papers that have successfully reused the data mentioned in Box 1 #3.

- * In Table 1, it would be helpful to explicitly state whether deep learning features are uncorrelated (which is mentioned later). A brief explanation of the underlying reasons would greatly benefit the reader. This is one example of a few topics that partly

- appear early in the manuscript and then appear again later. I would highly recommend discussing a topic in full in one place.
- * For a more comprehensive overview, I suggest citing some of the other leading cell segmentation methods and tools available in the field.
 - * In Section 2d on Feature Extraction, I recommend mentioning the more commonly used terms "representation" and "embedding," particularly in the context of deep learning.
 - * The concept of "annotation" (Section 2e) may be new to many readers. For improved flow, it might be more effective to introduce this concept within the section dedicated to software.
 - * "such as incomplete records, manual errors, software incompatibilities, and lack of standardization, risking loss of context when comparing across experiments, for example" - The use of "such as" and "for example" in this sentence is redundant. I suggest removing one of the phrases to make the sentence more concise.
 - * The discussion of the computational pipeline feels unbalanced, with some steps (e.g., normalization) receiving extensive detail while others (e.g., cell segmentation) are only briefly mentioned. A better approach might be to provide a concise overview of the entire pipeline and then select a few key topics for a more in-depth discussion.
 - * "One of the earliest and most widespread applications of deep learning has been cell segmentation, where neural networks replace classical image-processing algorithms to achieve greater accuracy, robustness, and scalability across diverse imaging modalities and experimental setups (Moshkov et al, 2024; Tang et al, 2024).". The references cited in this sentence do not appear to be core cell segmentation papers. I recommend replacing them with more appropriate citations.
 - * When discussing interpretability in microscopy, you could consider the following papers from my lab: [PMID: 39122948], [PMID: 34077708], [PMID: 39191720], including one example of interpretability in screening in <https://www.biorxiv.org/content/10.1101/2024.06.01.595856v1>.
 - * The acronym "IBP" appears suddenly in Figure 3 without prior definition. I recommend either defining it at its first use in the text or consistently using the full phrase "Image-based profiling. My personal preference is for the latter.
 - * I found it hard to follow the flow of section 4 and recommend the authors to revise. I found the style of this section to deviate from the style in the earlier parts of the text.
 - * Temporal information - there are several papers that used live imaging for screening, and might be worth discussing. For example PMID: 33835701, PMID: 32191874.
 - * The statement that "Most 3D features are hand drawn features..." is not accurate. There are many examples of advanced 3D feature extraction from labs like Danuser, Liberali and institutions like the Allen Institute for Cell Science ([PMID: 36599983]). If the focus is on perturbation-based screens, I suggest discussing recent 3D cell or organoid screening efforts.
 - * The section on virtual staining could be enhanced by including additional applications in cell profiling, such as those described in these works: <https://www.biorxiv.org/content/10.1101/2024.09.30.615654v1>, <https://arxiv.org/abs/2407.06979>, <https://ieeexplore.ieee.org/document/10230501>, <https://arxiv.org/abs/2303.08863>, <https://www.biorxiv.org/content/10.1101/2024.05.31.596710v1>, <https://www.biorxiv.org/content/10.1101/2024.12.19.629451v1>. A recent review from our team is available here PMID: 38838549. Diffusion models are worth discussing. In my opinion these papers offer insight beyond perceptual loss.
 - * Table 2 and Table 4 are missing from the provided PDF.
 - * To provide greater clarity and practical guidance, I recommend including a more detailed discussion on the importance of batch correction and the magnitude of performance differences between various methods.
 - * I believe a separate Discussion section is not necessary for a review article. I would suggest integrating the main discussion points into their relevant sections for improved flow.
 - * Box 2 currently frames cell segmentation as a limitation on engineered features. I suggest broadening this perspective to acknowledge that the cell is a fundamental unit of function, and therefore, single-cell representations hold intrinsic value and offer unique opportunities for biologically meaningful interpretability.
 - * References: "Shpigler A, Kolet N, Golan S, Weisbart E & Zaritsky A (2024) Anomaly detection for high-content image-based phenotypic cell profiling. *Bioinformatics*". This is a biorxiv manuscript from my lab. It was not published in *Bioinformatics*. Please be careful with accurate references.

Assaf Zaritsky
Ben-Gurion University of the Negev, Israel

Reviewer 1:

Summary and general remarks

The authors present a comprehensive review of the data analysis approaches and challenges for image-based profiling. Since the last in-depth review by Caicedo et al. (2017), numerous new deep learning approaches have significantly improved this field. Thus, this review is important and timely. Overall, the review is well written and rests on a large and highly relevant corpus of papers. There is a clear structure. Although the formatting of the headlines is broken and needs to be fixed.

The manuscript is of very high quality. However, I still have a number of points where I would like to engage in a discussion with the authors. Therefore I would like to ask the authors for comments and revisions.

We would like to sincerely thank the reviewer for their encouraging and thoughtful evaluation of our manuscript. We are delighted that they found our review to be comprehensive, important, and timely. We greatly appreciate the time and effort they dedicated to providing feedback, and we have carefully addressed all comments and suggestions to further strengthen our manuscript. We fixed the formatting of our headlines in the revision.

Major points

Section II

1. Experimental design and execution for image-based profiling

Lack of discussion of assay quality control. The authors have a section "Experimental design and execution for image-based profiling". There is very little mention of comprehensive and/or quantitative QC on the data acquisition side. Of course, this is not the most important section for this review. However, I believe data quality is highly relevant for analysis overall, and good QC starts at the acquisition. The only section kind of dedicated to it is "Whole-image QC" - which arguably deals only with filtering data - but one should create good data in the first place. In any case this addresses only the microscopy-image acquisition issues, while important, in our experience, they are the least pressing issues in a high-content screen. I think it is important to mention the usage of heatmaps to control for plate artefacts. The same goes for the assessment of batch effects with at least mentioning the usage of UMAPs in this context. No need for an in-depth discussion, but I would define these as very minimal requirements for any profiling assay.

We agree with the reviewer that ensuring high data quality is extremely important for image-based profiling. However, we consider improvements to data quality during acquisition (i.e., changes to laboratory procedures) beyond the scope of this paper.

That said, we should not wish to understate the importance of high quality data, and have added details on assessing technical artifacts after image acquisition. These additions can be found in the last paragraph of the "*Experimental design and execution for image-based profiling*" section where we now write:

"After images are acquired and features extracted, quality control becomes an important final step in the experimental workflow, ensuring that the earlier design choices have effectively

minimized technical noise. Visualization techniques are right for this assessment. Heatmaps of sample-level correlations can reveal batch effects, where blocks of high similarity correspond to specific plates or experimental dates rather than biological treatments (Caicedo *et al*, 2017). At a finer resolution, UMAPs of single-cell data, colored by metadata such as plate ID or well position, can highlight “islands” of cells artificially separated by technical variables or experimental details rather than biological treatments (Caicedo *et al*, 2017; Arevalo *et al*, 2024a). To assess the significance of batch effects, matched samples can be compared across batches, plate rows and columns using the mean average precision framework by (Kalinin *et al*, 2025). Detecting these sources of variation early is critical, as it guides normalization and batch correction strategies and ensures that downstream analyses yield robust and biologically meaningful results.”

2. Feature extraction

There is an inconsistency when the authors discuss the feature interpretability of classical features. In the comparison table (Classic approach vs Deep learning approach), they assess: "Interpretability: This mathematical approach ensures each feature is well-defined, enabling a clear understanding of how features are measured. Still, this does not always translate directly to biological meaning". In the text under section "Feature extraction," the Authors write a bit differently: "These features capture biologically important characteristics, reflecting single-cell morphological signatures. These classical approaches have been instrumental in establishing image-based profiling as a scalable and interpretable strategy". The first definition given in the table is much more balanced and accurate, as many features - e.g., texture features - have been defined by computer scientists to solve more generic computer vision tasks. Plus, even if they are mathematically well-defined, most of them lack clear interpretability. I would suggest to tone down the claim that image based profiling with classical features is a clear, interpretable strategy.

We thank the reviewer for their careful reading and providing this valuable feedback. We agree that toning down this claim is more reflective of the current state of the field. We have revised the text accordingly:

“In classical workflows, hundreds to thousands of hand-crafted features are extracted from segmented objects, quantifying aspects like size, shape, texture, intensity, and spatial relationships (Caicedo *et al*, 2017). However, this does not always translate to clear biological interpretability. Nevertheless, these features provide a mathematically well-defined representation of cell morphology, reflecting certain single-cell characteristics. These classical approaches have been instrumental in establishing image-based profiling as a scalable strategy (Forsgren *et al*, 2024). Their mathematical transparency allows for a clear understanding of feature measurement, supporting hypothesis-driven research (Driscoll & Zaritsky, 2021).”

3. Feature selection.

Feature selection is in our hands one of the most problematic and non-reproducible aspects of the classical analysis workflow. It's great that the authors highlight this. Here, I want to mention also a different approach that I came across (disclosure: the reviewer is NOT an author of any of these papers): Christoforow *et al*. 2019 defined their feature set on technical reproducibility by performing 2 biological repeats of a plate with reference compounds and including features that showed ≥ 0.8 similarity. As this analysis approach has been quite productive in the past, I think it merits a discussion (Christoforow *et al*. 2019, Schneidewind *et al*. 2020, Akbarzadeh *et al*. 2022).

We are grateful to the reviewer for mentioning this important feature selection strategy. We now include this in the 'g. *Feature selection*' section, of our discussion on selecting features based on their technical reproducibility across biological replicates, and have included the suggested citations.

“Another practical approach is to select features based on their technical reproducibility, retaining only those that show high correlation across biological replicates under identical conditions (Christoforow *et al*, 2019; Schneidewind *et al*, 2019; Akbarzadeh *et al*, 2022).”

4. Section IV

Only single-cell image-based profiling has its separate figure. Maybe the other aspects in this section would profit also from illustrations.

The reviewer is correct that we dedicated a figure exclusively to single-cell image-based profiling. This was an intentional choice, as we wanted to specifically highlight this concept because it's a central theme and recent evolution in the field.

We did consider adding illustrations for the other topics, but we felt that it would distract readers from the manuscript's primary narrative. We have also comprehensively described other sections in detail, which we felt did not warrant the same attention. Instead, to guide readers who are interested in an even deeper dive than we provide, we've made sure to add citations to several comprehensive review papers that cover these other areas in great detail.

We are also in communication with the editorial team, who will commission a more professional rendering of these figures, and we are limited in this regard to only specifically chosen topics.

5. Temporal and 3D image-based profiling

"Lastly, advances in 3D microscopy techniques, such as lattice light-sheet microscopy, Oblique Plane Microscopy, and other super-resolution methods, have made imaging in three dimensions more advantageous and information-rich than ever before (Chen *et al*, 2014)." For the image-based profiling field, large-scale datasets have been central. The lack of throughput of these microscopy methods is still an issue and an active field of research and development. I have not seen convincing systems on the market yet. I feel that this needs to be qualified.

We thank the reviewer for describing this crucial point. We agree that while this is an area of significant development, the current throughput limitations of advanced 3D microscopy techniques are a significant barrier to their widespread adoption in large-scale image-based profiling. We have revised the paragraph to add:

“However, the relatively low throughput of these advanced systems currently remains a barrier for generating the large-scale experiments that are central to adapting image-based profiling solutions, representing an active area of research and development (Danial, 2025; Bond *et al*, 2022).”

6. Virtual staining for image-based profiling

I am confused by this section, and I do not fully understand the main focus on virtual staining. There is a real push in the community towards establishing label-free approaches, using

brightfield imaging. While I agree that virtual staining, i.e., the prediction of fluorescent signals from the label-free images, is a valid research direction. The benefits at this point are rather unclear, as the authors also admit. This raises the question: why go through this extra step that most probably will also introduce errors or loss of information? Indeed, there are papers that show that training models on BF images directly has similar performance to training on CP images (Harrison et al. 2023). So it is questionable that virtual staining should be the main focus. I would suggest to title the section label-free and discuss it from this point of view with virtual staining as an interesting direction.

We thank the reviewer for this thoughtful and constructive comment. We agree that the direct analysis of label-free images is a powerful direction for our field, and we appreciate the reviewer's reference to important work demonstrating comparable performance.

Our choice to dedicate a section specifically to *virtual staining* was deliberate, as we see it as a distinct and rapidly developing methodological line of research. While label-free profiling asks, "*What phenotypes can we extract directly from a label-free image?*", the virtual staining community is exploring a complementary question: "*Can we computationally reconstruct the rich, multi-channel fluorescent data that decades of research and a large ecosystem of tools have been built around?*"

As the reviewer correctly notes, and as we acknowledge in the manuscript, the benefits of virtual staining for profiling remain to be fully established, and concerns such as potential information loss or error introduction are important considerations. Nevertheless, because our review aims to capture the breadth of emerging computational strategies, we felt it was valuable to highlight virtual staining as an active area of innovation, while also being transparent about its current limitations and open challenges (we added elaboration of this in the manuscript). In summary, our intent is not to present virtual staining as the definitive focus, but rather as a noteworthy methodological direction that illustrates the diversity of ongoing approaches in the field.

Below are the changes we made in the virtual staining section to more clearly reflect our reasoning:

Sentences 4-6 of the first paragraph:

"This strategy is distinct from directly profiling label-free images; although it is possible the transformation improves information content in the images, so far its central motivation is to translate label-free data into a format that is compatible with the vast ecosystem of established tools, datasets, and pre-trained models originally developed for fluorescent assays like Cell Painting (Cross-Zamirski *et al*, 2022; Seal *et al*, 2024b; Tonks *et al*, 2023). To perform virtual staining, several architectures have been proposed, including CNNs, GANs, and diffusion models in UNet frameworks (Xing *et al*, 2024; Ronneberger *et al*, 2015). Image-based profiles can then be derived from these virtually stained images, enabling downstream analyses using existing pipelines."

First sentence of the second paragraph:

"A key challenge for this emerging approach is to demonstrate that profiles from virtually stained images can reliably yield the same biological insights as those from real stains in downstream applications, or at least improved insights relative to the label-free images themselves. "

Last sentence of the third paragraph:

“The substantial promise of virtual staining, offering lower cost, reduced perturbation, and greater scalability, makes the continued validation and refinement of these methods a crucial and exciting frontier in image-based profiling.”

7. Single-cell quality control for image-based profiling

This section shares some content overlap with single-cell image-based profiling and is confusing because of the similar title, content, and, to some extent, similar goal of the text.

We thank the reviewer for identifying the overlap between these two sections. We agree with this assessment and have merged them into a single, more cohesive section called “*Single-cell image-based profiling*”. Within this section, we unified points and subsequently organized specific content into two distinct subsections: “*Methods for analyzing single-cell profiles*” and “*Quality control for single-cell profiles*.” We believe this new structure eliminates redundancy and better clarifies the relationship between general single-cell analysis methods and the specific challenges of single-cell quality control. Because of the substantial reworking of the two sections, we refer the reviewer to the manuscript file to review the updated text.

8. Batch correction for image-based profiling & Profile similarity metrics

I found these 2 sections very extensive and a bit unfocused.

We thank the reviewer for their careful reading and thoughtful feedback. In response, we have updated the structure of both sections to improve focus and clarity.

For the profile similarity metrics section, we have streamlined content. Our goal was to provide readers with an overview of both widely used and emerging similarity metrics in the field. We removed detailed experimental explanations, retaining only the relevant citations. The revised structure now clearly presents the available methods, outlines their limitations, and highlights recent developments.

For the batch correction section, we focused on current batch correction methods applied to image-based profiling. We reduced extraneous text and reorganized the section to improve clarity. The revised version emphasizes deep learning-based batch correction approaches, emerging methods that integrate batch correction at the image level to avoid post-hoc adjustments, and strategies leveraging domain adaptation.

Overall, these changes aim to make both sections more concise and focused while maintaining comprehensive coverage of the current state of the field. As well, because of the substantial reworking of these two sections, we refer the reviewer to the manuscript file to review the updated text.

9. The state of benchmarking in image-based profiling

A key factor that drives many analysis developments are the lack of good labels/annotations and the existing labels being problematic due to polypharmacology or dependence on factors such as cell Line, concentration, etc. It's great that the authors highlight this and raise that the community should act on improving this. They also mention it in Box 2. Although it is a bit lost under the aspect of benchmarking overall, addressing multiple attached aspects (e.g., datasets,

metrics etc). This lack of good ground truth - the labels, not the datasets - is quite a bottle neck. For instance, for image analysis - i.e., image segmentation - creating good ground truth was a big accelerator.

There are some projects that could additionally be mentioned that work on improving this. For instance, the annotations collected by the Broad Drug Repurposing hub (Corsello et al. 2017) were quite instrumental for our profiling projects. Additionally, the European Chemical Biology Database (ECBD <https://ecbd.eu/> - Skuta et al. 2025) efforts by collecting all kinds of relevant information based on the EU-OPENSOURCE compounds, even negative ones, via its bioprofiling efforts could become valuable (Disclaimer: reviewer is affiliated with EU-OPENSOURCE)

We thank the reviewer for this insightful comment and the suggestion of these valuable resources. We agree that the challenges of generating high-quality ground truth annotations remain a bottleneck for the field, which would be important to state in the relevant sections. We have revised the “*Availability of Benchmark Datasets*” section to highlight this issue strongly and have included the Broad Drug Repurposing Hub and European Chemical Biology Databases as key examples of the community trying to address these issues.

We added in the 4th paragraph:

“Recognizing this bottleneck, community efforts like the Broad Drug Repurposing Hub (Corsello *et al*, 2017) and European Chemical biology (Škuta *et al*, 2025) have become instrumental in providing more deeply curated and reliable annotations for model validation. When integrated with benchmarks, ground truth labels will accelerate image-based profiling method development.”

We also updated the box section within “5. *Building systematic benchmarking frameworks*” to reflect the importance of ground truth labels.

“A wide array of evaluation approaches have been proposed, but the lack of standardized evaluation metrics, well-defined benchmarks, and high-quality ground truth annotated data has limited comparability and practical deployment. The tide is beginning to turn here with recent benchmarks signaling the field’s commitment to standardizing evaluation and accelerating innovation (Chen *et al*, 2023). Early public datasets contain confounding factors, and computational demands often hinder reproducibility. Fortunately, a wave of new datasets (see Section V) is addressing this. Resources like Recursion’s RxRx datasets collection (Sypetkowski *et al*, 2023; Fay *et al*, 2023; Kraus *et al*, 2025), JUMP Cell Painting (Chandrasekaran *et al*, 2023), and the Cell Painting Gallery (Weisbart *et al*, 2024) providing higher-quality labels for model training and validation. Frameworks like EFAAR (Celik *et al*, 2024) also offer structured ways to assess and compare models. This is a timely opportunity for the community to coalesce around shared metrics, datasets and annotation standards.”

10. Section VI: Software for image-based profiling

This only focuses on open-source software. There is also commercially available software that is widely used and useful (e.g., Revvity Harmony). I believe some commercial solutions also deserve to be at least mentioned

We agree that including examples from commercially used software provides a more complete and balanced overview of the image-based profiling field. We have updated the second paragraph of the "*Software for image-based profiling*" section as shown below:

“Standardized data formats are essential for processing image-based profiling datasets. The wide variety of imaging platforms and analytical tools, ranging from open-source frameworks such as CellProfiler (Carpenter *et al*, 2006) and DeepProfiler (Moshkov *et al*, 2024) to proprietary commercial systems like Revvity's Harmony and Molecular Devices' IN Carta, produce a heterogeneous landscape of image analysis options and data outputs.”

Box 2: Solving challenges for the next decade

11. Developing standardized, end-to-end workflows.

I have an issue with the term "standardized workflows". Actually, I am not at all convinced why there is a need for standardized and/or universally adopted workflows. For me, standardized workflows mean that the sequence of components and/or the components of the workflow themselves would be fixed and/or the "same" across different analyses. First of all, would this not imply that all analysis challenges would be solved? Something the authors clearly disproved with the review they have written? The field seems to be still rapidly developing - a call for standardization could hamper this.

I agree to all the points about poor interoperability, fragmented ecosystem, lack of transparent reporting, etc. There are also good solutions raised by the authors, e.g., guidelines, libraries. For sure, if most of the field would agree to only use a few workflows, the technical complexity would be reduced, and naturally, these could become better documented. But that seems rather unlikely to happen, and "limiting" the workflows seems to be a very poor solution to these problems.

From an application viewpoint, it would make sense to standardize the analysis used for pre-clinical assays, e.g., Cell Painting in toxicology as a replacement of animal experiments. Because there would be a regulatory mandate (Piergiorganni *et al*. 2025). But I would argue this is currently only a minor part of the field.

This first section of this very interesting box leaves me unsatisfied. Sure, there are important technical issues raised, but a call for standardized analysis seems not to be a good solution for any of them. Better would be clear, transparent reporting of used workflows - whatever they might be, creating software with a focus on interoperability, benchmarking with comparable metrics and datasets, and an open discussion in the field in which frameworks/libraries to invest as community resources.

We are extremely grateful to the reviewer for this insightful and detailed comment. The reviewer raises a critical point, and their feedback has helped us significantly clarify the core message of this section.

We agree that a call for a single, fixed "standardized workflow" would be premature and could hamper innovation in this rapidly evolving field. We apologize for the lack of clarity. Our intention was not to propose a "one-workflow-to-rule-them-all." Instead, we argue for a more standardized *approach* to building flexible and interoperable end-to-end pipelines. Given the vast software ecosystem, our goal was to emphasize the need for modular tools that can be connected seamlessly to fit diverse analysis scenarios.

To address this, we have substantially revised the paragraph. We've removed the ambiguous term "standardized workflows" and replaced it with more precise language. The text now focuses on:

- The importance of interoperability between different software tools.
- The development of modular components that can be flexibly combined.
- The creation of well-documented end-to-end pipelines that are adaptable to different research questions.

Specifically, we now write:

“A core challenge is the development of modular and interoperable frameworks for constructing, reproducible, end-to-end, workflows for image-based profiling. Despite the growing ecosystem of open-source tools, there remains a lack of flexible, end-to-end solutions for executing the full bioinformatics pipeline, which starts from raw microscopy image acquisition and ends in interpretable image-based profiles. However, assembling such end-to-end workflows is challenging because of poor interoperability across a fragmented ecosystem of tools that often requires extensive, bespoke, human-in-the-loop, and error-prone programming (Djaffardjy *et al*, 2023; Hu *et al*, 2021). This challenge is further compounded by the emergence of temporal, 3D, and virtually stained datasets, each introducing additional complexity through unique data formats, modality-specific processing requirements, and a lack of harmonized computational approaches (Arora *et al*, 2023). Encouragingly, foundational tools such as Pycytominer (Serrano *et al*, 2025), BioProfiling.jl (Vulliard *et al*, 2022), scmorph (Wagner *et al*, 2025), and DeepProfiler (Moshkov *et al*, 2024) are maturing, and community resources such as the Image-Based Profiling Handbook (Cimini *et al*, 2019) provide valuable best-practice guidelines. The widespread availability of FAIR-compliant (Wilkinson *et al*, 2016) datasets also creates ideal conditions for testing and validating new workflows. Solving this challenge will not only streamline high-throughput image-based profiling for experts and non-experts alike, but will also enhance reproducibility, promote interoperability, and enable more scalable, integrated analyses. A community-driven focus on creating modular, interoperable, and well-documented tools for building these pipelines can empower broader participation, reduce redundant efforts, and accelerate both methodological development and biological insight.”

12. Fortifying methodological infrastructure for temporal, 3D, and virtual staining modalities

As above with the virtual staining section: "Virtual staining provides a promising strategy for reducing phototoxicity and experimental cost, though it only approximates ground truth offered from real reagents". Label-free imaging is providing these benefits, e.g., brightfield imaging. To me, it seems unclear if the virtual staining, i.e., the prediction of fluorescent signals from the label-free images, is actually beneficial for the analysis of label-free images. I would suggest rewriting.

We thank the reviewer for this point. The original sentence was imprecise and failed to clearly articulate the motivation behind virtual staining. We have rewritten this part of the paragraph as follows:

“Furthermore, virtual staining is a promising strategy that aims to translate low-cost, low-phototoxicity, label-free images (e.g., brightfield) into fluorescence-like images. While this prediction only approximates the ground truth offered by real reagents, a primary motivation is to make these data compatible with the vast ecosystem of analysis tools and pre-trained models originally developed for fluorescence microscopy.”

The revised text accomplishes two things:

1. It correctly attributes the benefits of reduced phototoxicity and experimental cost to the label-free imaging modalities (e.g., brightfield) that serve as the input for virtual staining.
2. It explicitly states the primary motivation for performing the virtual stain transformation: to make these data compatible with the vast ecosystem of analysis tools and pre-trained models that were originally developed for fluorescence microscopy.

13. Ensuring ontological and interpretable feature extraction across experiments and platforms

It's a very important and evocative section. But I am not entirely convinced by it.

A) Centrality of the interpretability problem. Given that the field has been quite productive, even just based on the rather weak interpretability of the classical image features. My impression is that this issue does not have a very high priority.

B) I am also a bit sceptical about the "transformative" potential of the solution to this, at least for the field of image-based profiling. The field has been driven in the past to produce very large microscopy data based on fixed cells and with low resolution. It is a bit hard for me to see how this approach could yield a strong mechanistic understanding of cellular processes - or how they would be altered when perturbed. I say this also as fields that have that expressed goal - e.g., virtual cell - seem to have a preference for high resolution and also time-lapse imaging. Thus, I believe this is also a data problem, not only an analysis issue.

We thank the reviewer for the thoughtful and constructive feedback. We agree with the assessment that our initial draft overstated the centrality of the interpretability problem without sufficiently acknowledging prior progress in the field or the inherent limitations of the input data itself.

To address these points, we made several key revisions:

First, we revised the section opening to better contextualize the problem. We now highlight that classical feature-based approaches have already achieved substantial success and that challenges in interpretability represent a natural next step in moving from descriptive to more explanatory analyses.

Second, in response to the reviewer comment about this being fundamentally a "data problem," we added a new sentence explicitly acknowledging that the potential for deep mechanistic insight is ultimately constrained by data quality. We now emphasize that improved interpretability, while valuable, cannot overcome limitations in the underlying measurements.

Finally, we adjusted the language around the "transformative" potential of our proposed approach. The revised text now presents this potential as emerging from the combination of more interpretable features and higher-quality data.

3. Bridging profiles to mechanisms through interpretable features

The interpretability of morphology features, ranging from classical hand-crafted measurements to those derived from deep learning, remains a key challenge in image-based profiling (Garcia-Fossa *et al*, 2023). While the field has been highly productive using classical features that are not always directly interpretable, moving from powerful descriptive profiles to robust

explanatory frameworks requires a deeper understanding of what features represent (Chandrasekaran *et al*, 2020; Moshkov *et al*, 2024; Garcia-Fossa *et al*, 2023). Classical features, though defined, are generated in large numbers, with many being noisy, redundant, or susceptible to technical artifacts that obscure biological signals (Murthy *et al*, 2024; Caicedo *et al*, 2017). Furthermore, different image analysis tools inconsistently name features, which depend on the input parameters and make a unifying ontology challenging (Image Feature Variability Results For Five Open-Source Feature Extraction Libraries). Deep learning features pose an even greater interpretability barrier, typically manifesting as “anonymous latent variables” with no explicit meaning nor obvious mapping from trained model to trained model. This black-box nature makes it difficult to link phenotypic profiles to specific biological mechanisms (Seal *et al*, 2024a; Foroughi Pour *et al*, 2022). The core challenge lies in developing and validating methods to reveal what these complex models are capturing, disentangling biologically meaningful variation from technical noise. However, it is crucial to recognize that the potential for mechanistic understanding may be fundamentally constrained by the input data itself. The low-resolution or static 2D images common in large-scale screens may not contain the granular information needed to explain dynamic cellular processes, a limitation that improved feature interpretation alone cannot overcome (Gordonov *et al*, 2016; Yoshida *et al*, 2023; Chandrasekaran *et al*, 2020). The field is scaling to increasingly complex datasets, including 3D, temporal, and virtually stained modalities, which may provide sufficiently interpretable, but they are generating high-dimensional data with minimal intuitive guidance without additional methodology layered on. Explainable AI methods such as GradCAM (Longo *et al*, 2024) and attention-based visualization (Vaswani *et al*, 2017; Pfaendler *et al*, 2023; Doron *et al*, 2023; Morelli *et al*, 2025) are gaining traction in image-based profiling contexts, helping to clarify how models make phenotypic distinctions. Addressing this challenge will be highly valuable, as it will increase confidence in AI-driven findings and enable mechanistically grounded hypothesis generation. Ultimately, combining more interpretable features with richer data from high-resolution or time-lapse imaging offers a transformative path to shift image-based profiling from a powerful descriptive method toward a robust explanatory framework.”

Minor points:

1. "Single-Cell image-based profiling": "In the future, impact assessment of QC should become a standard in image-based profiling to ensure the highest quality of profiles and downstream analyses (Pratapa *et al*. 2021)." It's unclear what is meant by "impact assessment". The citation does not use that term.

We thank the reviewer for highlighting this ambiguity. Our intention was to convey the need to quantitatively measure how QC procedures affect downstream results. We have now updated the sentence for more clarity and the citations as well.

“In the future, a standard practice should be to quantitatively evaluate and report how quality control procedures affect downstream analysis outcomes, ensuring the highest quality of profiles and results (Subramanian *et al*, 2022; Qiu *et al*, 2020)“

2. "Optical pooled screening": "Processing OPS data builds upon foundational steps in classical image-based profiling (i.e., illumination correction, segmentation, and feature extraction) while introducing specialized components for single-cell barcode deconvolution for each cell (see Figure 1)." It's unclear what the Figure reference means - there seems to be no term "single-cell barcode deconvolution" in the figure.

We thank the reviewer for catching this. We agree that the figure reference was unclear, and we have removed it.

3. "Many cell tracking algorithms exist, such as TrackMate and Ultrack (Ershov et al, 2022; Bragantini et al, 2024)". TrackMate is a software suite (Fiji plugin) and not a specific algorithm - it implements different segmentation and tracking algorithms.

We thank the reviewer for this important distinction. We have revised the sentence to refer to 'cell tracking software' to be more accurate.

"Many cell tracking software exist, such as TrackMate and Ultrack (Ershov *et al*, 2022; Bragantini *et al*, 2024), however there remain both major technical and computational hurdles for incorporating these single-cell tracking methods into high throughput temporal image-based profiling pipelines."

4. The term ISS is defined twice in the first paragraph of "Optical pooled screening".

We thank the reviewer for catching this oversight. We have removed the second definition.

5. Formatting of the headlines is broken and needs to be fixed.

We apologize for this inconvenience. The document headings appear correct on our end, but we will confirm with the Molecular Systems editorial team.

References:

A. Christoforow, J. Wilke, A. Binici, A. Pahl, C. Ostermann, S. Sievers, H. Waldmann
Design, Synthesis, and Phenotypic Profiling of Pyrano-Furo-Pyridone Pseudo Natural Products
Angew. Chem. Int. Ed., 58 (2019), pp. 14715-14723, [10.1002/anie.201907853](https://doi.org/10.1002/anie.201907853)

T. Schneidewind, S. Kapoor, G. Garivet, G. Karageorgis, R. Narayan, G. Vendrell-Navarro, A.P. Antonchick, S. Ziegler, H. Waldmann. The pseudo natural product myokinasib is a myosin light chain kinase 1 inhibitor with unprecedented chemotype *Cell Chem. Biol.*, 26 (2019), pp. 512-523

M. Akbarzadeh, I. Deipenwisch, B. Schoelermann, A. Pahl, S. Sievers, S. Ziegler, H. Waldmann. Morphological profiling by means of the Cell Painting assay enables identification of tubulin-targeting compounds *Cell Chem. Biol.* <https://doi.org/10.1016/j.chembiol.2021.12.009>

Harrison PJ, Gupta A, Rietdijk J, Wieslander H, Carreras-Puigvert J, Georgiev P, et al. (2023) Evaluating the utility of brightfield image data for mechanism of action prediction. *PLoS Comput Biol* 19(7): e1011323. <https://doi.org/10.1371/journal.pcbi.1011323>

Corsello, S., Bittker, J., Liu, Z. et al. The Drug Repurposing Hub: a next-generation drug library and information resource. *Nat Med* 23, 405-408 (2017). <https://doi.org/10.1038/nm.4306>

Ctibor Škuta, Tomáš Müller, Milan Voršilák, Martin Popr, Trevor Epp, Katholiki E Skopelitou, Federica Rossella, Katja Herzog, Bahne Stechmann, Philip Gribbon, Petr Bartůněk, ECBD: European chemical biology database, *Nucleic Acids Research*, 2025;, gkae904, <https://doi.org/10.1093/nar/gkae904>

Piergiovanni, M., Mennecozzi, M., Barale-Thomas, E. et al. Bridging imaging-based in vitro methods from biomedical research to regulatory toxicology. Arch Toxicol 99, 1271-1285 (2025). <https://doi.org/10.1007/s00204-024-03922-z>

Reviewer 2:

Summary and General remarks

The review article presented by Erik Serrano and the community around CytoData Society is a timely and state-of-the-art update to their 2017 best practice article. In my opinion, the details, intricacies, and guides for beginners in the field of image-based profiling, as well as high-content/phenotypic screening, are comprehensively covered and extended in upcoming areas of research. What I particularly liked is where the authors centered on approaches/methods/software that have not yet yielded significant improvements over the more traditional way of analyzing cellular imagery. I would also like to highlight that I noticed the authors made a great effort to describe methods and advances in an application-agnostic way, wherever possible, including referencing peripheral articles as well. Set aside that I think this is an excellent read for anybody working or being interested in the booming field of phenotypic profiling, I do have some minor comments, which I list below:

We sincerely thank the reviewer for their positive and encouraging assessment of our manuscript. We are delighted that they found our review to be a timely, state-of-the-art, and comprehensive update. We especially appreciate their recognition of our efforts to present a balanced perspective and maintain an application-agnostic approach. We are grateful for their kind words and will address the minor comments they raised below.

Minor points

1. In Figure 1, take care that the microscope is not the picture of a specific instrument from some vendor. This should be more generalized.
 - 1.a. The same holds true for software like CellProfiler and DeepProfiler. Though these are excellent examples, there are also many other tools (as the authors explain later), but those two are highlighted particularly, and that shall be avoided.
 - 1.c. The text boxes in Figure 1 are redundant to the main text and can either be actual text boxes or cut from the figure, as well as the "downstream-analysis icon" that is too small for interpretation after all.

We thank the reviewer for their careful eye. We agree that generalized images are more appropriate. We are working with the editorial team to create artistic renderings of these draft figures, and we will ensure more general references are added, and that any additional text is clear.

2. The entire Section II would benefit significantly if the learnings since the time of Caicedo 2017 et al. were highlighted. Otherwise, it seems like repeating what many people already know very well.

We thank the reviewer for this constructive feedback.

We have reframed the entire section to use the 2017 paper as a foundational baseline and to better articulate the evolution of the pipeline. This approach is strategic: Instead of performing a detailed comparison in every single subsection, we've focused on explicitly contrasting the classical methods with modern advancements in the areas where the most significant paradigm shifts have occurred. For example, in subsections like "*Illumination Correction, Cell Segmentation, Feature Extraction, and Batch Correction*", we now directly reference the 2017 state-of-the-art and then detail how and why modern deep learning approaches have changed the methodology.

We agree this approach effectively highlights the key "learnings" since 2017, as the reviewer suggested, while maintaining the section's primary goal of describing the modern image-based profiling pipeline (some components of which are relatively unchanged since 2017). We believe these changes have improved the section's narrative and impact for the reader.

For example, in the introductory paragraph under "*Transforming microscopy images into high-dimensional image-based profiles*"

"The canonical multi-step workflow, thoroughly reviewed in (Caicedo *et al*, 2017), provides a foundational framework for this process. In the years since, while the fundamental steps remain similar, significant advancements, particularly driven by deep learning, have modernized each component of this pipeline (Way *et al*, 2023; Driscoll & Zaritsky, 2021). Deep learning approaches may sidestep some of these steps to process embeddings directly from raw microscopy images (Moshkov *et al*, 2024)."

In the illumination correction last sentence:

"Notably, however, many modern deep learning pipelines often bypass this explicit correction step, as models can learn to be robust to such illumination variations during training, representing a significant leap (Wang *et al*, 2021; Guo *et al*, 2025; Wang *et al*, 2023; Li *et al*, 2022; Rai *et al*, 2022)."

In the batch correction section:

"However, the increasing scale and complexity of modern datasets have pushed the field to adapt more sophisticated methods from other domains, as simple plate-level normalization is often insufficient. "

And in the feature selection section:

"This move toward learning compact representations directly from pixels, which often bypasses the need for explicit post hoc feature selection, marks one of the most significant evolutions from the classical, multi-step workflows described by (Caicedo *et al*, 2017; Tang *et al*, 2024; Chandrasekaran *et al*, 2020)."

3. Section II a-f, in my view, need a figure with small representative examples, such that also newbies in the field can relate the text to the problem at hand and thus gain a deeper understanding of the methods applied.

We appreciate the reviewer's suggestion to include additional figures. Our existing figure was intentionally designed to focus exclusively on single-cell profiles, emphasizing the distinction between this approach and traditional population-level profiling.

We carefully considered creating a new figure that provides small illustrative examples for each method discussed in Section II. However, we were concerned that including an additional figure would make the section dense and could actually detract from the manuscript's overall flow.

Therefore we decided to cite key papers and reviews that offer clear visual examples and detailed explanations. We believe this approach maintains the manuscript's clarity and focus while effectively guiding readers to authoritative visual resources. Nevertheless, we will keep approachability as a core concept when we approve the editorial figure drafts.

4. I'd suggest clustering section IV a little bit differently, such that the different current challenges and topics connect a little bit better e.g.: a,d,e,f,c,b,g

We thank the reviewer for this thoughtful suggestion. We agree that reorganizing this section improves its logical flow and narrative structure for the reader.

We have reordered the subsections to first introduce the major shifts in data types and modalities, and then discuss the cross-cutting analytical methods that apply to them. The new order is as follows:

- Single-cell image-based profiling
- Optical pooled screening
- Temporal and 3D image-based profiling
- Virtual staining for image-based profiling
- Integration of omics
- Batch correction for image-based profiling
- Profile similarity metrics

We agree this new structure tells a more coherent story. It begins with the fundamental move to single-cell resolution and its related topics, then broadens to other emerging data dimensions and types, and concludes with the universal analytical challenges: batch correction and profile comparison, that are relevant to all the preceding topics. We are grateful for the feedback that led to this improvement.

Otherwise, congratulations on a well-written, balanced, and comprehensive review.

We thank the reviewer for their encouragement.

Reviewer 3

Summary and general remarks

Serrano et al. survey the methodological and computational aspects of image-based profiling. The manuscript details the experimental pipeline and computational analyses followed by focusing on three areas. First, a comprehensive description of deep learning in cell profiling, where I especially enjoyed the clear description of weak- and self-supervised learning (Section 3). Second, "emerging" advances and challenges (Section 4) and third, surveying benchmarking and software tools. Given the growing number of cell profiling projects, this review is timely and valuable. However, I believe the manuscript would be strengthened by a more focused and organized structure.

We sincerely thank the reviewer for their thoughtful and encouraging feedback. We greatly appreciate their recognition of the clarity and depth of our discussion, particularly regarding weakly supervised and self-supervised learning, and their positive assessment of the review's timeliness and value to the growing image-based profiling community. We provide our point-by-point response below.

Major points

This review primarily focuses on the analysis of Cell Painting data, using it as the main example throughout the text. I believe it would be beneficial to state this explicitly in the introduction to manage reader expectations. While this focus allows for a detailed discussion of aspects specific to Cell Painting (and similar phenotypic-based screens), it may not fully represent other forms of high-content cell phenotyping. This is also reflected in the terminology used (e.g., "perturbation" for experimental conditions). A broader framing of "cell profiling" would, in my opinion, require substantial revisions to accurately reflect the state of the field.

The following comments are offered as constructive suggestions to strengthen the manuscript. It is, of course, up to the authors to determine if and how these can best be implemented.

In my opinion, the review's broad scope and organization could be streamlined for greater clarity and impact. While a timely overview of the field, the current structure jumps between topics, making the flow less coherent than it could be. I suggest reorganizing the technical discussion into two main sections following the experimental and computational pipeline: one dedicated to "Current Technical Aspects and Best Practices" and a second to "Emerging Topics and the Future". Furthermore, I recommend that the authors transition from a broad survey of the literature to a more focused discussion. Synthesizing the key findings into a more coherent narrative would transform the manuscript from a list of papers into an engaging and insightful review.

To facilitate this, the section on "Current Technical Aspects and Best Practices" could be structured around key questions that are highly relevant to the readership. These topics are touched upon, but a more structured discussion would enhance the manuscript's contribution:

We are deeply grateful for the reviewer's thoughtful assessment of our work. We addressed many of the reviewer's suggestions here but refrained from a large-scale reorganization of the manuscript, as such changes would be impractical at this stage. The review was developed through extensive collaboration among multiple contributors, and a major structural revision would require substantial coordination and feedback from all co-authors, and may not provide a commensurate benefit to our future readers.

We did, however, implement many targeted revisions as suggested (for example, completely restructuring the section IV). We improved flow, focus, and clarity within the existing framework, effectively addressing the reviewer's concerns while maintaining the manuscript's coherence and value proposition to our future readers.

1. Representations: A comparative analysis of deep learning versus classic feature extraction would be highly valuable. This could address in what situation each approach is most effective, the trade-off in interpretability, and the magnitude of performance gains from deep learning. This discussion could also be broadened to include the distinction and implications of single-cell versus patch-based representations.

We thank the reviewer for this suggestion. We note that providing a quantitative analysis of the differences between deep learning and classical feature extraction methods is highly desirable, but it is unfortunately out of scope for this review.

Previous studies use existing benchmarks to evaluate performance on certain biological tasks, but these are developed for specific conditions, which limits generalizability of conclusions (Doron *et al*, 2023; Sanchez-Fernandez *et al*, 2023; Kim *et al*, 2025; Wong *et al*, 2023; Moshkov *et al*, 2024; Kraus *et al*, 2025; Morelli *et al*, 2025; Pawlowski *et al*, 2016; Hofmarcher *et al*, 2019; Gao *et al*, 2025; Doan *et al*, 2021; Kraus *et al*, 2024). The question of how beneficial deep learning is with respect to classical features is still an open research problem (as we discuss) and more studies are needed to clarify, including evaluations on benchmarks where models are trained and applied to diverse datasets. An effort of this type would require significant additional effort.

We instead elaborate on trends in a (mostly) qualitative way, which provides a high-level overview of the concepts discussed in the deep learning section. We present this comparison at the end of the deep learning section in a subsection titled “*Comparative analysis of classical features and deep learning*”.

The substantially updated text is as follows:

Previous studies have benchmarked the improvements of using deep learning compared to classical image-based profiling features (Doron *et al*, 2023; Sanchez-Fernandez *et al*, 2023; Kim *et al*, 2025; Wong *et al*, 2023; Moshkov *et al*, 2024; Kraus *et al*, 2025; Morelli *et al*, 2025; Pawlowski *et al*, 2016; Hofmarcher *et al*, 2019; Gao *et al*, 2025; Doan *et al*, 2021; Kraus *et al*, 2024). However, the specific added benefit remains a case-by-case basis and an open research question. We provide a summary of performance metrics comparing deep learning and classical features (**Table 2**). Specifically, benchmarks include comparing performance in the main biological task, usability, interpretability of features, data requirements, and runtime. On average, these benchmarking studies showcased a relative improvement of ~20% for deep learning approaches over classical features. We note that this does not account for learning paradigm (SSL, WSL or SL), model architecture or other factors that would prevent this from being a robust quantitative measure of deep learning performance for image based profiling.

There are also other various trade-offs, unrelated to performance. For example, classical features are implemented with graphical user interfaces that facilitate configuration and streamlined preparation of data processing (Stirling *et al*, 2021; Schindelin *et al*, 2012). On the other hand, deep learning solutions rarely provide a common interface and will need to be adapted to a given user's data via custom scripts, requiring additional programming expertise and other challenges to implementation. Furthermore, classical features can be applied to all scales of data: They have no minimum nor maximum data requirements. Deep learning models, on the other hand, require thousands of images to adequately fine-tune or tens of thousands of images to train from scratch (Steiner *et al*, 2021). Nevertheless, a trained deep learning model can extract representations much quicker than classical approaches. For example, a 50x speed improvement towards generating embeddings via deep learning was reported in (Kim *et al*, 2025) Lastly, classical features likely require additional pre or post processing, such as segmentation, illumination correction, and feature selection and these costs should be considered. Conversely, certain deep learning approaches may sidestep some or all preprocessing steps. For instance, segmentation can be skipped by processing image patches (or whole FOVs) instead, which results in less computation work, while yielding similar performance (Kim *et al*, 2025; Xun *et al*, 2024). The distinction between patch-based and

single-cell representations carries important implications. Generating patch-based representations offer major speed advantages and perform well for population-level classification but lack single-cell resolution (Xun *et al*, 2024). In contrast, single-cell representations retain information on cell heterogeneity, enabling analyses at the cellular level, but depend on computationally intensive segmentation, which can introduce errors and propagate unpredictable downstream effects, potentially compromising key biological insights such as protein localization or cell-cycle state classification.

We also added a new Table 2 to summarize:

Table 2. Comparing classical features with deep learning on key performance metrics.

Metric	Classical features	Deep learning
Performance	Robust baseline results.	~20% improvement relative to classical features
Usability	Published pipelines and user-friendly software available to reproduce results for many profiling tasks.	Requires knowledge of the torch ecosystem and the creation of custom extraction scripts. Requires further programming and compute resources to fine-tune/train models.
Interpretability	Features are named measurements taken from images, which may provide insights in some cases.	Features are unnamed latent variables. Requires additional methods (e.g., generative models) or further data analysis.
Data requirements	No minimum data requirements.	Pre-trained models have no minimum data requirements. Fine-tuning and training may need thousands to tens of thousands of samples
Runtime	Varies. Needs CPU compute. GPU-acceleration is typically not available. For large scale studies, cluster-based parallelization is needed.	Up to 50x faster for predictions due to GPU acceleration.
Hidden costs	Illumination correction, segmentation and feature selection.	Optional segmentation. May require fine-tuning or training

2. Encoding Cell Heterogeneity: A discussion on the trade-offs between averaging representations at the well level versus preserving single-cell heterogeneity. What does the current literature say about the downstream performance gained by maintaining a population-level representation?

We'd thank the reviewer for their insightful question. In response, we've made three key revisions to the manuscript:

Revision 1: We added information about aggregation techniques beyond traditional mean and median to the "Aggregation and Annotation" section.

"Additionally, alternative aggregation strategies extend beyond medians, incorporating distribution-based descriptors, cluster-derived subpopulation proportions, and learned aggregation functions from deep representation models (Doron *et al*, 2023; Frey *et al*, 2025; van Dijk *et al*, 2024; Yao *et al*, 2024; Hu *et al*, 2025; Pearson *et al*, 2022; Garcia-Fossa *et al*, 2023; Rohban *et al*, 2019)"

Revision 2: We discuss the advantages of aggregate-level profiles. This new text highlights their usage in downstream tasks, and we've included citations that support these points.

"Aggregating data in well-level or treatment-level profiles offers advantages by creating a robust and stable representation that minimizes the impact of irrelevant single-cell variability and technical noise. Furthermore, it dramatically reduces the volume of data points, ensuring that downstream computational analyses, like comparing thousands of compound profiles, remain scalable and efficient (Caicedo *et al*, 2017; Chandrasekaran *et al*, 2020). For many downstream applications, such as predicting a compound's mechanism of action (MoA), these aggregated profiles have proven highly effective (Scheeder *et al*, 2018)."

Revision 3: We provide a concise summary of the core trade-offs between the two approaches in the concluding paragraph of "*Methods for analyzing single-cell profile*" section, supported by citations for each perspective.

"Specifically, the primary trade-off is between the improved biological signal detection and discovery of subtle phenotypes offered by preserving single-cell heterogeneity, versus the computational efficiency, scalability, and robustness to noise provided by aggregate-level representations (van Dijk *et al*, 2024; Palma *et al*, 2025; Garcia-Fossa *et al*, 2023; Stossi *et al*, 2024)."

3. Normalization and Batch Correction: It would be highly beneficial to discuss the impact of different normalization and batch correction methods on downstream performance. Are there significant alterations that researchers should be aware of?

We thank the reviewer for raising this point. We agree that a systematic comparison of how different normalization and batch correction methods impact downstream performance is a crucial need for the image-based profiling community.

However, we believe such a comparison is beyond the scope of this manuscript. Our primary goal was to provide a comprehensive survey of the many available methods and their underlying mechanisms. We reserve a rigorous benchmarking of different data processing methods for a separate (but related) effort. We hope our review serves as a crucial foundation that enables and motivates exactly this type of future work.

4. Best Practices for Downstream Analyses: The authors could propose a set of "best practice" measurements for downstream analyses, such as reproducibility or MoA classification. While some answers may not be conclusive, a structured discussion based on relevant literature would be a great contribution to the field.

We agree that a guide on best practices for downstream analysis would be a valuable resource for the field *in the future*.

For this review, we focused on surveying the broad and rapidly evolving landscape of computational methods rather than providing a fixed set of "best practices". Our primary goal was to promote the diversity of emerging approaches and challenges within the field.

We feel that in a field moving as quickly as image-based profiling, defining "best practices" is challenging, as today's top-performing method may be superseded tomorrow. In fact, another reviewer cautioned against premature standardization, as it could risk stifling the very innovation our review seeks to document (see Reviewer 1 comment 11). Prescribing a single set of "best" approaches for downstream analyses would be at odds with this dynamic nature of the current state of our field.

While we do not include a dedicated "best practices" section, we have made an effort throughout the manuscript to discuss the specific strengths, weaknesses, and appropriate contexts for the various methods described. We hope this approach provides researchers with the necessary information to select the most suitable analytical strategies for their own specific research questions. Nevertheless, we are clear in stating cases where there is knowledge of clear optimal strategies (e.g., image augmentation for deep learning).

5. For the "Emerging Topics and Future" section, a clearer justification for each topic is needed. It would be helpful to explain what makes these areas "emerging," why they are important now, and what specific advances have enabled their recent development. For example, the authors could make the case that the emergence of "foundation models" are a trigger for data sharing and data standardization.

We sincerely thank the reviewer for this suggestion. Our intention was to introduce emerging topics, and adding a clear justification for why they are emerging now would increase clarity and context. We added a sentence providing justification to all items in Box 2. The sentences (taken out of their context within the box; please see the manuscript for the full changes) are below:

Item 1: This topic is gaining urgency not only because of the challenges workflows would solve, but also because recent advances now make solutions attainable.

Item 2: This is rapidly changing, however, as recent advances in microscopy hardware, assays (e.g., live-cell painting) generative AI models, and scalable computation are making these complex modalities now feasible for high-throughput profiling

Item 3: This is an emerging topic as a critical bottleneck because the field's rapid adoption of powerful deep learning models, which often act as "black boxes," has outpaced our ability to interpret what they are learning.

Item 4: This challenge is emerging now as the field's focus is shifting from population-level analyses to high-resolution single-cell resolution.

Item 5: This challenge has become particularly urgent, as the recent release of massive, high-quality public datasets now provides the common ground necessary to compare the concurrent explosion of new computational methods and deep learning models.

We also substantially revised Box 2, Item 6:

"A major frontier and growing success in image-based profiling is the rapid adoption and evolution of deep learning models, which have moved beyond proof-of-concept to become common in the field. Increasingly, these models are being used to scale profiling efforts, improve generalizability across experiments, adapt to varying fluorescent channel configurations, and enhance interpretability of complex cellular phenotypes (Pratapa *et al*, 2021; Tang *et al*, 2024). The next decade will see similar advances using AI. Furthermore, foundation models trained on large, diverse imaging datasets are now being actively developed and deployed, reflecting a shift toward more robust, flexible, and general-purpose AI systems for image-based profiling (Kraus *et al*, 2024; Kenyon-Dean *et al*, 2024; Gupta *et al*, 2024; Ji *et al*, 2024). This topic is emerging as a critical frontier now because the field has finally overcome two major historical barriers that previously constrained progress: the lack of massive-scale public datasets and the high cost of computation. The recent emergence of large, standardized datasets (such as the JUMP-CP and RxRx) and more accessible high-performance computing are the specific advances that have directly enabled the development of these powerful, data-hungry models. In turn, the very success of these models now acts as a powerful trigger for further data standardization and sharing, as the community recognizes the value of combining resources to build even more generalizable AI. Continued advancement will democratize access to state-of-the-art models, support reproducible and interpretable analyses, and empower researchers to derive deeper, more transparent insights from image-based profiling data."

Minor points

1. The abstract appears to focus on a few specific topics that may not fully align with the broader scope of the review. It would be helpful to ensure the abstract accurately reflects the full content and emphasis of the manuscript.
2. In the Abstract the statement, "Despite these advancements, the field still faces significant challenges requiring innovative solutions", is very broad and not specific. I recommend either specifying what these challenges are or omitting the sentence for greater conciseness.

We thank the reviewer for the valuable feedback on the abstract. We have now substantially revised our abstract to more accurately and comprehensively reflect the manuscript's scope. Additions include clarifying significant challenges, discussions on the core processing pipeline, the variety of emerging methodologies, and the critical role of benchmarking.

"For over two decades, image-based profiling has revolutionized cell phenotype analysis. Image-based profiling processes rich, high-throughput, microscopy data into thousands of unbiased measurements that reveal phenotypic patterns powerful for drug discovery, functional genomics, and cell state classification. Here, we review the evolving computational landscape of image-based profiling, detailing the bioinformatics processes involved from feature extraction to normalization and batch correction. We discuss how deep learning has fundamentally reshaped the field. We examine key methodological advancements, such as single-cell analysis, the development of robust similarity metrics, and the expansion into new modalities like optical pooled screening, temporal imaging, and 3D organoid profiling. We also highlight the growth of public benchmarks and open-source software ecosystems as a key driver for fostering reproducibility and collaboration. Despite these advances, the field still faces substantial challenges, particularly in developing methods for emerging temporal and 3D data modalities, establishing robust quality control standards and workflows, and interpreting the processed features. By focusing on the technical evolution of image-based profiling rather than the

wide-ranging biological applications, our aim with this review is to provide researchers with a roadmap for navigating the progress and new challenges in this rapidly advancing domain.”

3. "Since then, the field has advanced rapidly, driven by progress in both experimental protocols (e.g., Cell Painting (Bray et al, 2016), Live Cell Painting (Garcia-Fossa et al, 2024))". To better reflect the recent advancements in experimental protocols, I suggest citing a more contemporary screening technique, such as those involving organoids, instead of Live Cell Painting, which has not yet been widely demonstrated for practical applications. Also, Cell Painting was already referenced in the 2017 paper and thus should probably be excluded in this context.

The reviewer raises a valid point that these original examples are not the most representative of advancements since 2017. We've now revised the sentence to highlight a more contemporary advancement.

“Since then, the field has advanced rapidly, driven by progress in both experimental protocols such as live-cell high-content assays (Moraes-Lacerda *et al*, 2025; Garcia-Fossa *et al*, 2025; Cottet *et al*, 2023), 3D spheroids and organoid models (Lukonin *et al*, 2021; Bian *et al*, 2021; Ringers *et al*, 2025), complementary profiling modalities (Dagher *et al*, 2025), and rapidly advancing computational tools, including the rise of deep learning, data integration methods, and reproducible software (**Box 1**).”

4. Box 1 - To make the benefits of data sharing more concrete, it would be valuable to highlight specific papers that have successfully reused the data mentioned in Box 1 #3.

We thank the reviewer for this suggestion. We've added a sentence that highlights specific examples of studies that have successfully reused data.

“The availability of these datasets has driven rapid development in a wide range of methodological research, including new computational tools such as scmorph (Wagner *et al*, 2025), Pycytominer (Serrano *et al*, 2025), DeepProfiler (Moshkov *et al*, 2024), PhenoProfiler (Li *et al*, 2025), CytoSummaryNet (van Dijk *et al*, 2024), and SPACe (Stossi *et al*, 2024), as well as extensive benchmarking of existing algorithms (Arevalo *et al*, 2024a; Li *et al*, 2025; Yan *et al*, 2025; Kim *et al*, 2025; Tian *et al*, 2023; Shpigler *et al*, 2024).”

5. In Table 1, it would be helpful to explicitly state whether deep learning features are uncorrelated (which is mentioned later). A brief explanation of the underlying reasons would greatly benefit the reader. This is one example of a few topics that partly appear early in the manuscript and then appear again later. I would highly recommend discussing a topic in full in one place.

We thank the reviewer for their helpful suggestion to improve Table 1. We've added a new row we call 'Feature redundancy'

Feature attribute	Classical approach (e.g., CellProfiler)	Deep learning approach (e.g., CNNs, ViTs)
Feature redundancy	Features are often highly correlated, leading to significant redundancy (e.g., multiple features measuring different	Embeddings are typically uncorrelated or have low redundancy. This happens when models are optimized to learn a compact representation where each

	aspects of "size"). This necessitates a feature selection step to remove redundant information.	feature captures an independent axis of variation, making feature selection less important.
--	---	---

In general, we took care to streamline many topics, discussing them in only one spot. For example, we substantially reworked the single-cell image-based profiling section to include processing and quality control in the same place.

6. For a more comprehensive overview, I suggest citing some of the other leading cell segmentation methods and tools available in the field.

We thank the reviewer for this suggestion. We agree that the original section on cell segmentation was not comprehensive and have substantially reworked it to provide a more thorough overview.

Specifically, the revised section now better reflects the historical progression of these methods, starting with classical techniques and detailing the impact of deep learning approaches, from foundational CNN and U-Net architectures to the latest transformer-based models. Because of the richness of the segmentation field, we have intentionally kept the descriptions at a high level, as a detailed technical breakdown of each specific segmentation algorithm is beyond the scope of this review.

Revised Cell segmentation section:

“Segmentation is a critical step in most image-based profiling workflows. Classical approaches, implemented in tools like CellProfiler (Stirling *et al*, 2021) and Ilastik (Berg *et al*, 2019), use image analysis techniques such as intensity thresholding and watershed algorithms to delineate individual cells from the background (Wang *et al*, 2024). However, the field is witnessing a shift to deep learning, a transition catalyzed by foundational architectures like U-Net, which uses convolutional neural networks (CNNs) to achieve superior segmentation accuracy (Ronneberger *et al*, 2015; O’Shea & Nash, 2015; Chen & Murphy, 2023; Schmidt *et al*, 2018). This advancement paved the way for cell segmentation tools such as StarDist (Schmidt *et al*, 2018; Weigert & Schmidt, 2022), and Cellpose (Stringer *et al*, 2020) that offer models capable of identifying cell borders with minimal parameter tuning. Building on the success in segmentation, deep learning models have been extended to tackle the more complex challenge of cell tracking, often by performing both tasks simultaneously in end-to-end frameworks (Chen *et al*, 2021; Schwartz *et al*, 2019). More recently, the advent of foundation models has introduced transformer-based architectures to segmentation. Tools like CellSAM (Israel *et al*, 2023) and CellViT (Hörst *et al*, 2024, 2025) leverage these models to further improve performance and generalizability across diverse imaging conditions. Some of these powerful architectures are being applied to end-to-end models that perform both segmentation and tracking of cells over time (O’Connor & Dunlop, 2025). Despite significant performance improvements over the last two decades, challenges remain, especially in generalizing models to new data types and imaging modalities. Accurately segmenting cells with low signal-to-noise ratios, complex

morphologies, or high cell densities, especially within large 3D datasets, continues to be a difficult task within the field (Maška *et al*, 2023; Edlund *et al*, 2021).”

7. In Section 2d on Feature Extraction, I recommend mentioning the more commonly used terms "representation" and "embedding," particularly in the context of deep learning.

We thank the reviewer for this suggestion. We agree that incorporating the terms "representation" and "embedding" improves clarity, especially for readers with a background in deep learning. Here are the changes that we have made in section 2d:

“These learned representations, often referred to as ‘embeddings’, are often extracted via architectures like convolutional neural networks (CNNs) or vision transformers (ViTs), frequently outperform classical features in downstream tasks, a topic we explore further in section three.”

8. The concept of "annotation" (Section 2e) may be new to many readers. For improved flow, it might be more effective to introduce this concept within the section dedicated to software.

In regards to the reviewer’s comment about “annotation” in Section 2e. While we agree that annotation is often performed using specific software, we believe it is a conceptually distinct and critical step in the image-based profiling pipeline that warrants to be included in this section. Annotation is simply the process of appending key metadata information to profiles. Our intention is to frame annotation as a fundamental bridge between quantitative feature extraction and downstream biological interpretation.

However, the reviewer's feedback prompted us to revisit the section to improve its focus. We have revised the text to reduce specific software mentions and technical jargon. The section now focuses on explaining annotation and its importance.

Below is the revised paragraph:

“In the annotation step, metadata is merged with the associated aggregated profiles, which provides contextual information, enabling meaningful downstream interpretation and other analyses (Way *et al*, 2023). Key metadata include screen layouts, plate maps, sample types, treatment details (e.g., siRNA/shRNA IDs, gene targets, CRISPR guides, compounds), control types, replicate counts, concentrations and time points, which are particularly vital for temporal studies (Cimini *et al*, 2023; Caicedo *et al*, 2017; Garcia-Fossa *et al*, 2024; Graham *et al*, 2025). Importantly, efforts toward maintaining metadata annotation ensure datasets are well-documented, shareable, and integratable across studies, thereby promoting reproducibility and long-term utility under Findable, Accessible, Interoperable, and Reusable (FAIR) principles (Wilkinson *et al*, 2016; Way *et al*, 2023; Williams *et al*, 2017). Metadata is essential for downstream processes including normalization, batch correction, and quality control filtering (Caicedo *et al*, 2017). Despite its importance, metadata annotation faces challenges such as incomplete records, manual errors, software incompatibilities, and lack of standardization, which risks loss of context when comparing across experiments.”

9. "such as incomplete records, manual errors, software incompatibilities, and lack of standardization, risking loss of context when comparing across experiments, for example" - The use of "such as" and "for example" in this sentence is redundant. I suggest removing one of the phrases to make the sentence more concise.

We appreciate the reviewer's thorough evaluation. We have updated as following:

“Despite its importance, metadata annotation faces challenges such as incomplete records, manual errors, software incompatibilities, and lack of standardization, which risks loss of context when comparing across experiments.”

10. The discussion of the computational pipeline feels unbalanced, with some steps (e.g., normalization) receiving extensive detail while others (e.g., cell segmentation) are only briefly mentioned. A better approach might be to provide a concise overview of the entire pipeline and then select a few key topics for a more in-depth discussion.

We appreciate the reviewer’s constructive feedback. After much consideration, we elect to maintain the current structure. We believe that our readers will appreciate dedicated sections describing each core step, even if they are unbalanced in their content. Our primary focus is on the downstream processing of image-based profiles, which is why steps like normalization are discussed more extensively, as they are central to this topic. We now prompt the reader with this expectation as follows:

“We focus our discussion on these advances, and, while we discuss each step, we primarily focus on the downstream image-based profiling rather than upstream steps (e.g., segmentation). It is also important to note that many deep learning approaches may skip some of these upstream steps to process embeddings directly from raw microscopy images (Moshkov *et al*, 2024; Donovan-Maiye *et al*, 2022; Zhou *et al*, 2024)”

Furthermore, to better balance the discussion, we have added more detail to the cell segmentation section (Reviewer 3, minor comment 6). However, we believe that an exhaustive technical review of segmentation algorithms is beyond the scope of this paper.

11. "One of the earliest and most widespread applications of deep learning has been cell segmentation, where neural networks replace classical image-processing algorithms to achieve greater accuracy, robustness, and scalability across diverse imaging modalities and experimental setups (Moshkov *et al*, 2024; Tang *et al*, 2024)." The references cited in this sentence do not appear to be core cell segmentation papers. I recommend replacing them with more appropriate citations.

We thank the reviewer for this helpful suggestion. We agree that the sentence is better supported by more foundational citations. We have now updated the reference list to include core papers that are more relevant to the development of deep learning for cell segmentation.

“One of the earliest and most widespread applications of deep learning has been cell segmentation, where neural networks replace classical image-processing algorithms to achieve greater accuracy, robustness, and scalability across diverse imaging modalities and experimental setups (see section II: Cell segmentation) (Ronneberger *et al*, 2015; Chen *et al*, 2025; He *et al*, 2017; Schmidt *et al*, 2018)”

12. When discussing interpretability in microscopy, you could consider the following papers from my lab: [PMID: 39122948], [PMID: 34077708], [PMID: 39191720], including one example of interpretability in screening in <https://www.biorxiv.org/content/10.1101/2024.06.01.595856v1>.

We thank the reviewer for pointing us to this relevant body of work, which we regret omitting. We have now revised the 'Interpreting deep learning features' section (Section IIIe) to incorporate these references. Specifically, we have added a more nuanced discussion on

explainability versus interpretability, included generative counterfactuals as a key example of feature space interpretation, and highlighted the application of attribution methods in a high-throughput screening context.

“Deep learning features present a unique challenge for interpretation, as they are generally anonymous latent variables that do not have explicit names or meaning. This contrasts with traditional image-based profiling, which relies on predefined features extracted using tools like CellProfiler. These features are categorized by image channel, feature type (e.g., shape, texture, correlation), and cell compartment (e.g., nucleus or cytoplasm) and are therefore more interpretable. The challenge of deciphering deep learning models has spurred the growth of explainable AI (XAI), where a key distinction is made between “explainability” (describing the cause for a decision) and “interpretability” (translating an explanation into domain-specific insight) (Rotem & Zaritsky, 2024). Methods for visual interpretability in deep learning for image-based profiling generally fall into two categories: pixel-level explanations and feature space interpretation. Pixel-level explanations reveal how individual pixels contribute to model predictions, using techniques such as GradCAM (Longo *et al*, 2024; Gopalakrishnan *et al*, 2024; Rotem & Zaritsky, 2024). In transformer-based models, the attention mechanism across layers can provide insights into which regions of the input image are most influential in shaping learned representations (Pfaendler *et al*, 2023). While these methods provide detailed insights at the individual image level, they are often difficult to generalize or aggregate across perturbations, limiting their utility for downstream biological interpretation. Extending these ideas, attribution methods like SHapley Additive exPlanations (SHAP) (Lundberg & Lee, 2017) can also be applied. For example, in a high-throughput context, these methods can explain why a treated cell's profile is 'anomalous' by identifying which specific inter-feature dependencies, learned from control cells, were “broken” by the treatment (Shpigler *et al*, 2024). In contrast, feature space interpretation focuses on understanding the model's embedding space by decoding learned features into images that visually represent cell phenotypes. Generative models have also been explored to gain understanding of how images are organized in the latent space of deep learning models, including variational autoencoders (Lafarge *et al*, 2019), generative adversarial networks (GAN) (Goldsborough *et al*, 2017), and conditional GANs (Goldsborough *et al*, 2017; Lamielle *et al*, 2023; Fonnegra *et al*, 2023). For example, an interpretability technique, sometimes called "morphing," uses generative models to create counterfactual *in silico* images by intentionally altering the model's latent features, which can make the learned phenotypic properties visually apparent (Rotem *et al*, 2024; Zaritsky *et al*, 2021). However, further research in this field is needed to deepen our understanding of phenotypic variation of cells in microscopy images and to better connect image-derived representations with other sources of biological knowledge. This remains a fertile and promising frontier for advancing image-based profiling.”

13. The acronym "IBP" appears suddenly in Figure 3 without prior definition. I recommend either defining it at its first use in the text or consistently using the full phrase "Image-based profiling. My personal preference is for the latter.

We thank the reviewer for pointing out this artifact. We have followed their recommendation and have updated Figure 3 accordingly.

14. I found it hard to follow the flow of section 4 and recommend the authors to revise. I found the style of this section to deviate from the style in the earlier parts of the text.

We thank the reviewer for their thoughtful feedback on Section 4. Our initial intent was to present each emerging method as a self-contained topic to highlight its specific contributions to

the field. Based on this suggestion (as well as similar comments from other reviewers), we have revised the section to improve flow, clarity, and consistency. We make changes to most subsections, and substantially reworked others (e.g., single-cell image-based profiling). Please see the updated manuscript for all changes.

15. Temporal information - there are several papers that used live imaging for screening, and might be worth discussing. For example PMID: 33835701, PMID: 32191874.

We thank the reviewer for this valuable suggestion and for pointing us to these highly relevant papers. We have now revised the 'Temporal and 3D image-based profiling' section to include a discussion of these studies, highlighting them as prime examples of how dynamic, time-lapse data is being successfully applied in a high-throughput screening context:

“Additionally, these technologies now enable high-throughput screens where temporal information, such as signaling dynamics, serves as the primary readout instead of a single static endpoint (Goglia *et al*, 2020). The complex trajectories of cells have necessitated analytical tools like CODEX, which uses deep learning to explore and classify diverse signaling dynamics landscapes (Jacques *et al*, 2021).”

16. The statement that "Most 3D features are hand drawn features..." is not accurate. There are many examples of advanced 3D feature extraction from labs like Danuser, Liberali and institutions like the Allen Institute for Cell Science ([PMID: 36599983]). If the focus is on perturbation-based screens, I suggest discussing recent 3D cell or organoid screening efforts.

We thank the reviewer for this important correction and for providing valuable literature. We have now corrected this sentence in the manuscript.:

“Most 3D features are not hand drawn features using traditional computer vision principles (Ong *et al*, 2025). “

We also expanded the section to include a discussion of recent, 3D cell and organoid screening efforts, citing relevant work that highlights the progress in this area.

“Indeed, rapid development in the field is beginning to realize this potential, with new protocols adapting 2D assays like Cell Painting for 3D spheroids and enabling high-content morphological profiling of these models (Jeremiasso *et al*, 2024; Bozal *et al*, 2024; Ringers *et al*, 2025; Huang *et al*, 2024; Lukonin *et al*, 2021; Viana *et al*, 2023; Ramm *et al*, 2022; Betge *et al*, 2022). This progress is further supported by the creation of dedicated high-content screening software that uses automated AI-driven analysis to facilitate robust, high-throughput phenotyping in 3D models (Diosdi *et al*, 2025; Kok *et al*, 2025; Madan *et al*, 2025).”

17. The section on virtual staining could be enhanced by including additional applications in cell profiling, such as those described in these works:
<https://www.biorxiv.org/content/10.1101/2024.09.30.615654v1>, <https://arxiv.org/abs/2407.06979>,
<https://ieeexplore.ieee.org/document/10230501>, <https://arxiv.org/abs/2303.08863>,
<https://www.biorxiv.org/content/10.1101/2024.05.31.596710v1>,
<https://www.biorxiv.org/content/10.1101/2024.12.19.629451v1>.

A recent review from our team is available here PMID: 38838549. Diffusion models are worth discussing. In my opinion these papers offer insight beyond perceptual loss.

We thank the reviewer for these valuable suggestions and for highlighting additional works that deepen the discussion on virtual staining. We have revised this section to incorporate the suggested references, including recent studies and the review, and expanded our discussion to cover diffusion models and their relevance beyond perceptual loss.

“This strategy is distinct from directly profiling label-free images; although it is possible virtual staining improves information content in the images, so far its central motivation is to translate label-free data into a format that is compatible with the vast ecosystem of established tools, datasets, and pre-trained models originally developed for fluorescent assays like Cell Painting (Cross-Zamirski *et al*, 2022; Seal *et al*, 2024b; Tonks *et al*, 2023). To perform virtual staining, several architectures have been proposed, including CNNs, GANs, and diffusion models in UNet frameworks (Xing *et al*, 2024; Ronneberger *et al*, 2015; Elmalam *et al*, 2024; Zaritsky *et al*, 2021; Navidi *et al*, 2024). Image-based profiles can then be derived from these virtually stained images, enabling downstream analyses using existing pipelines.”

“However, recent work has extensively used image-based profiling pipelines to rigorously evaluate the biological faithfulness of virtual staining (Follain *et al*, 2024; Cross-Zamirski *et al*, 2023; Le & Lundberg, 2024; De Vries *et al*, 2025). For example, a multi-level evaluation framework that uses CellProfiler-derived features and high-throughput screening quality metrics like the robust Z prime (RZ') factor to demonstrate that while virtual nuclei and cytoplasm stains are reliable, other more complex structures like DNA-damage spots are not accurately reproduced (Tonks *et al*, 2023). Other studies have used morphological profiles to show that class-guided diffusion models, which incorporate experimental metadata, can improve performance in downstream tasks like mechanism-of-action prediction (Cross-Zamirski *et al*, 2023). “

18. Table 2 and Table 4 are missing from the provided PDF.

We apologize for the confusion regarding the missing tables. We submitted them as separate supplementary files with our original manuscript, but it appears they may not have been readily accessible to the reviewers. We have ensured that all tables are correctly uploaded with our revised submission and are clearly referenced in the main text. These tables are now Table EV1 and Table EV2

19. To provide greater clarity and practical guidance, I recommend including a more detailed discussion on the importance of batch correction and the magnitude of performance differences between various methods.

We thank the reviewer for this suggestion.

Regarding the request to discuss the magnitude of performance differences between methods, we agree this is a critical question for the field. However, we feel that a comprehensive, quantitative benchmark would require an extensive systematic review, which falls outside the scope of our current manuscript. Our primary goal is to provide a broad survey of the available methods rather than a deep performance comparison.

That said, we do recognize the importance of such benchmarks. To this end, our manuscript already highlights the initial efforts in this area by citing the work of (Arevalo *et al*, 2024b), who systematically evaluated several batch correction methods on a large-scale image-based profiling dataset. We believe that our review, by cataloging the landscape of available tools, will

help motivate and provide a foundation for more of these essential benchmarking studies in the future.

20. I believe a separate Discussion section is not necessary for a review article. I would suggest integrating the main discussion points into their relevant sections for improved flow.

We thank the reviewer for the suggestions to improve the manuscript's flow.

We agree that the second paragraph of the "Conclusion" section was largely a summary of topics detailed earlier in the paper. To reduce repetition, we have removed it.

In line with this change, we have retitled the section to "Future challenges" to better reflect its new, more focused purpose.

21. Box 2 currently frames cell segmentation as a limitation on engineered features. I suggest broadening this perspective to acknowledge that the cell is a fundamental unit of function, and therefore, single-cell representations hold intrinsic value and offer unique opportunities for biologically meaningful interpretability.

We thank the reviewer for this important suggestion. We agree that segmentation should be framed not only as a technical challenge but also as a critical step for preserving the biological meaning of single-cell representations. As suggested, we revised *Box 2* to highlight that accurate segmentation maintains the integrity of the cell as the fundamental unit of function and enables biologically interpretable single-cell profiles

“Importantly, beyond being a technical necessity for single-cell analysis, accurate segmentation preserves the integrity of the cell as the fundamental unit of function (Israel et al, 2025). This enables single-cell representations that capture biologically meaningful variability and foster interpretable connections between morphology and mechanism (Chen & Murphy, 2023).”

22. References: "Shpigler A, Kolet N, Golan S, Weisbart E & Zaritsky A (2024) Anomaly detection for high-content image-based phenotypic cell profiling. *Bioinformatics*". This is a biorxiv manuscript from my lab. It was not published in *Bioinformatics*. Please be careful with accurate references.

We thank the reviewer for catching this and sincerely apologize for the incorrect citation. The updated citation now reads:

“Shpigler A, Kolet N, Golan S, Weisbart E & Zaritsky A (2025) Anomaly detection for high-content image-based phenotypic cell profiling. *Cell Syst*”

At the time of resubmission, the article is in press. Once published, we will update the citation accordingly.

References

Akbarzadeh M, Deipenwisch I, Schoelermann B, Pahl A, Sievers S, Ziegler S & Waldmann H (2022) Morphological profiling by means of the Cell Painting assay enables identification of tubulin-targeting compounds. *Cell Chem Biol* 29: 1053–1064.e3

- Arevalo J, Su E, Ewald JD, van Dijk R, Carpenter AE & Singh S (2024a) Evaluating batch correction methods for image-based cell profiling. *Nature Communications* 15: 1–12
- Arevalo J, Su E, Ewald JD, van Dijk R, Carpenter AE & Singh S (2024b) Evaluating batch correction methods for image-based cell profiling. *Nat Commun* 15: 6516
- Arora A, Alderman JE, Palmer J, Ganapathi S, Laws E, McCradden MD, Oakden-Rayner L, Pfohl SR, Ghassemi M, McKay F, *et al* (2023) The value of standards for health datasets in artificial intelligence-based applications. *Nat Med* 29: 2929–2938
- Berg S, Kutra D, Kroeger T, Straehle CN, Kausler BX, Haubold C, Schiegg M, Ales J, Beier T, Rudy M, *et al* (2019) ilastik: interactive machine learning for (bio)image analysis. *Nature Methods* 16: 1226–1232
- Betge J, Rindtorff N, Sauer J, Rauscher B, Dingert C, Gaitantzi H, Herweck F, Srouf-Mhanna K, Miersch T, Valentini E, *et al* (2022) The drug-induced phenotypic landscape of colorectal cancer organoids. *Nature Communications* 13: 1–15
- Bian X, Li G, Wang C, Liu W, Lin X, Chen Z, Cheung M & Luo X (2021) A deep learning model for detection and tracking in high-throughput images of organoid. *Comput Biol Med* 134: 104490
- Bond C, Santiago-Ruiz AN, Tang Q & Lakadamyali M (2022) Technological advances in super-resolution microscopy to study cellular processes. *Mol Cell* 82: 315–332
- Bozal SB, Sjogren G, Costa AP, Brown JS, Roberts S, Baker D, Gabriel P Jr, Ristau BT, Samuels M, Flynn WF, *et al* (2024) Development of an automated 3D high content cell screening platform for organoid phenotyping. *SLAS Discov* 29: 100182
- Bragantini J, Theodoro I, Zhao X, Huijben TAPM, Hirata-Miyasaki E, VijayKumar S, Balasubramanian A, Lao T, Agrawal R, Xiao S, *et al* (2024) Ultrack: pushing the limits of cell tracking across biological scales. *bioRxiv*
- Caicedo JC, Cooper S, Heigwer F, Warchal S, Qiu P, Molnar C, Vasilevich AS, Barry JD, Bansal HS, Kraus O, *et al* (2017) Data-analysis strategies for image-based cell profiling. *Nat Methods* 14: 849–863
- Carpenter AE, Jones TR, Lamprecht MR, Clarke C, Kang IH, Friman O, Guertin DA, Chang JH, Lindquist RA, Moffat J, *et al* (2006) CellProfiler: image analysis software for identifying and quantifying cell phenotypes. *Genome Biology* 7: 1–11
- Celik S, Hütter J-C, Carlos SM, Lazar NH, Mohan R, Tillinghast C, Biancalani T, Fay MM, Earnshaw BA & Haque IS (2024) Building, benchmarking, and exploring perturbative maps of transcriptional and morphological data. *PLOS Computational Biology* 20: e1012463
- Chandrasekaran SN, Ackerman J, Alix E, Ando DM, Arevalo J, Bennion M, Boisseau N, Borowa A, Boyd JD, Brino L, *et al* (2023) JUMP Cell Painting dataset: morphological impact of 136,000 chemical and genetic perturbations. *bioRxiv*
- Chandrasekaran SN, Ceulemans H, Boyd JD & Carpenter AE (2020) Image-based profiling for drug discovery: due for a machine-learning upgrade? *Nature Reviews Drug Discovery* 20: 145

- Chen C, Mat Isa NA & Liu X (2025) A review of convolutional neural network based methods for medical image classification. *Comput Biol Med* 185: 109507
- Chen H & Murphy RF (2023) Evaluation of cell segmentation methods without reference segmentations. *Mol Biol Cell* 34: ar50
- Chen Y, Song Y, Zhang C, Zhang F, O'Donnell L, Chrzanowski W & Cai W (2021) Celltrack R-CNN: A novel end-to-end deep neural network for cell segmentation and tracking in microscopy images. In *2021 IEEE 18th International Symposium on Biomedical Imaging (ISBI)* IEEE
- Chen Z, Pham C, Wang S, Doron M, Moshkov N, Plummer BA & Caicedo JC (2023) CHAMMI: A benchmark for channel-adaptive models in microscopy imaging.
- Christoforow A, Wilke J, Binici A, Pahl A, Ostermann C, Sievers S & Waldmann H (2019) Design, Synthesis, and Phenotypic Profiling of Pyrano-Furo-Pyridone Pseudo Natural Products. *Angew Chem Int Ed Engl* 58: 14715–14723
- Cimini BA, Chandrasekaran SN, Kost-Alimova M, Miller L, Goodale A, Fritchman B, Byrne P, Garg S, Jamali N, Logan DJ, *et al* (2023) Optimizing the Cell Painting assay for image-based profiling. *Nature Protocols* 18: 1981–2013
- Cimini BA, Way G, Becker T, Weisbart E, Chandrasekaran SN, Tromans-Coia C, Shafqat Abbasi H, Carpenter A & Singh S (2019) Image-based Profiling Handbook.
- Corsello SM, Bittker JA, Liu Z, Gould J, McCarren P, Hirschman JE, Johnston SE, Vrcic A, Wong B, Khan M, *et al* (2017) The Drug Repurposing Hub: a next-generation drug library and information resource. *Nat Med* 23: 405–408
- Cottet M, Marrero YF, Mathien S, Audette K, Lambert R, Bonneil E, Chng K, Campos A & Andrews DW (2023) Live Cell Painting: New nontoxic dye to probe cell physiology in high content screening. *SLAS Discov* 29: 100121
- Cross-Zamirski JO, Anand P, Williams G, Mouchet E, Wang Y & Schönlieb C-B (2023) Class-guided image-to-image diffusion: Cell painting from Brightfield images with class labels. *arXiv [csCV]*
- Cross-Zamirski JO, Mouchet E, Williams G, Schönlieb C-B, Turkki R & Wang Y (2022) Label-free prediction of cell painting from brightfield images. *Scientific Reports* 12: 1–13
- Dagher M, Ongo G, Robichaud N, Kong J, Rho W, Teahulos I, Tavakoli A, Bovaird S, Merjaneh S, Tan A, *et al* (2025) nELISA: a high-throughput, high-plex platform enables quantitative profiling of the inflammatory secretome. *Nat Methods*
- Danial JSH (2025) Super-resolution microscopy for structural biology. *Nature Methods* 22: 1636–1652
- De Vries M, Dent LG, Curry N, Rowe-Brown L, Bousgouni V, Fourkioti O, Naidoo R, Sparks H, Tyson A, Dunsby C, *et al* (2025) Geometric deep learning and multiple-instance learning for 3D cell-shape profiling. *Cell Syst* 16: 101229
- van Dijk R, Arevalo J, Babadi M, Carpenter AE & Singh S (2024) Capturing cell heterogeneity in representations of cell populations for image-based profiling using contrastive learning.

- Diosdi A, Toth T, Harmati M, Istvan G, Schrettner B, Hapek N, Kovacs F, Kriston A, Buzas K, Pampaloni F, *et al* (2025) HCS-3DX, a next-generation AI-driven automated 3D-oid high-content screening system. *Nat Commun* 16
- Djaffardjy M, Marchment G, Sebe C, Blanchet R, Bellajhame K, Gaignard A, Lemoine F & Cohen-Boulakia S (2023) Developing and reusing bioinformatics data analysis pipelines using scientific workflow systems. *Comput Struct Biotechnol J* 21: 2075–2085
- Doan M, Barnes C, McQuin C, Caicedo JC, Goodman A, Carpenter AE & Rees P (2021) Deepometry, a framework for applying supervised and weakly supervised deep learning to imaging cytometry. *Nat Protoc* 16: 3572–3595
- Donovan-Maiye RM, Brown JM, Chan CK, Ding L, Yan C, Gaudreault N, Theriot JA, Maleckar MM, Knijnenburg TA & Johnson GR (2022) A deep generative model of 3D single-cell organization. *PLoS Comput Biol* 18: e1009155
- Doron M, Moutakanni T, Chen ZS, Moshkov N, Caron M, Touvron H, Bojanowski P, Pernice WM & Caicedo JC (2023) Unbiased single-cell morphology with self-supervised vision transformers. *bioRxiv*: 2023.06.16.545359
- Driscoll MK & Zaritsky A (2021) Data science in cell imaging. *J Cell Sci* 134
- Edlund C, Jackson TR, Khalid N, Bevan N, Dale T, Dengel A, Ahmed S, Trygg J & Sjögren R (2021) LIVECell-A large-scale dataset for label-free live cell segmentation. *Nat Methods* 18: 1038–1045
- Elmalam N, Ben Nedava L & Zaritsky A (2024) In silico labeling in cell biology: Potential and limitations. *Curr Opin Cell Biol* 89: 102378
- Ershov D, Phan M-S, Pylvänäinen JW, Rigaud SU, Le Blanc L, Charles-Orszag A, Conway JRW, Laine RF, Roy NH, Bonazzi D, *et al* (2022) TrackMate 7: integrating state-of-the-art segmentation algorithms into tracking pipelines. *Nat Methods* 19: 829–832
- Fay MM, Kraus O, Victors M, Arumugam L, Vuggumudi K, Urbanik J, Hansen K, Celik S, Cernek N, Jagannathan G, *et al* (2023) RxRx3: Phenomics Map of Biology. *bioRxiv*
- Follain G, Ghimire S, Pylvänäinen JW, Vaitkevičiūtė M, Wurzinger D, Guzmán C, Conway JRW, Dibus M, Oikari S, Rilla K, *et al* (2024) Fast label-free live imaging reveals key roles of flow dynamics and CD44-HA interaction in cancer cell arrest on endothelial monolayers. *bioRxiv*
- Fonnegra R, Sanian M, Chen Z, Paavolainen L & Caicedo J (2023) Analysis of cellular phenotypes with unbiased image-based generative models. In *NeurIPS 2023 Generative AI and Biology (GenBio) Workshop*
- Foroughi Pour A, White BS, Park J, Sheridan TB & Chuang JH (2022) Deep learning features encode interpretable morphologies within histological images. *Sci Rep* 12: 9428
- Forsgren E, Cloarec O, Jonsson P, Lovell G & Trygg J (2024) A scalable, data analytics workflow for image-based morphological profiles. *Chemometr Intell Lab Syst* 254: 105232
- Frey B, Holmberg D, Byström P, Bergman E, Georgiev P, Johansson M, Hennig P, Rietdijk J,

- Rosén D, Carreras-Puigvert J, *et al* (2025) Single-cell morphological profiling reveals insights into cell death. *bioRxiv*: 2025.01.15.633042
- Gao X, Zhang F, Guo X, Yao M, Wang X, Chen D, Zhang G, Wang X & Lai L (2025) Attention-based deep learning for accurate cell image analysis. *Sci Rep* 15: 1265
- Garcia-Fossa F, Cruz MC, Haghighi M, de Jesus MB, Singh S, Carpenter AE & Cimini BA (2023) Interpreting Image-based Profiles using Similarity Clustering and Single-Cell Visualization. *Curr Protoc* 3: e713
- Garcia-Fossa F, Moraes-Lacerda T, Rodrigues-da-Silva M, Diaz-Rohrer B, Singh S, Carpenter AE, Cimini BA & de Jesus MB (2024) Live Cell Painting: image-based profiling in live cells using Acridine Orange. *bioRxiv*: 2024.08.28.610144
- Garcia-Fossa F, Moraes-Lacerda T, Rodrigues-da-Silva M, Diaz-Rohrer B, Singh S, Carpenter AE, Cimini BA & de Jesus MB (2025) Live-cell painting: Image-based profiling in live cells using acridine orange. *Mol Biol Cell* 36: mr7
- Goglia AG, Wilson MZ, Jena SG, Silbert J, Basta LP, Devenport D & Toettcher JE (2020) A live-cell screen for altered Erk dynamics reveals principles of proliferative control. *Cell Syst* 10: 240–253.e6
- Goldsborough P, Pawlowski N, Caicedo JC, Singh S & Carpenter AE (2017) CytoGAN: Generative Modeling of Cell Images. *bioRxiv*: 227645
- Gopalakrishnan V, Ma J & Xie Z (2024) Grad-CAMO: Learning Interpretable Single-Cell Morphological Profiles from 3D Cell Painting Images.
- Gordonov S, Hwang MK, Wells A, Gertler FB, Lauffenburger DA & Bathe M (2016) Time series modeling of live-cell shape dynamics for image-based phenotypic profiling. *Integr Biol (Camb)* 8: 73–90
- Graham RE, Zheng R, Wagner J, Unciti-Broceta A, Hay DC, Forbes SJ, Gadd VL & Carragher NO (2025) Single-cell morphological tracking of cell states to identify small-molecule modulators of liver differentiation. *iScience* 28: 111871
- Guo M, Wu Y, Hobson CM, Su Y, Qian S, Krueger E, Christensen R, Kroeschell G, Bui J, Chaw M, *et al* (2025) Deep learning-based aberration compensation improves contrast and resolution in fluorescence microscopy. *Nat Commun* 16: 313
- Gupta A, Wefers Z, Kahnert K, Hansen JN, Leineweber W, Cesnik A, Lu D, Axelsson U, Navarro FB, Karaletsos T, *et al* (2024) SubCell: Vision foundation models for microscopy capture single-cell biology. *bioRxiv*: 2024.12.06.627299
- He K, Gkioxari G, Dollár P & Girshick R (2017) Mask R-CNN. *arXiv [csCV]*
- Hofmarcher M, Rumetshofer E, Clevert D-A, Hochreiter S & Klambauer G (2019) Accurate prediction of biological assays with high-throughput microscopy images and convolutional networks. *J Chem Inf Model* 59: 1163–1171
- Hörst F, Rempe M, Becker H, Heine L, Keyl J & Kleesiek J (2025) CellViT++: Energy-efficient and adaptive cell segmentation and classification using foundation models. *arXiv [csCV]*

- Hörst F, Rempe M, Heine L, Seibold C, Keyl J, Baldini G, Ugurel S, Siveke J, Grünwald B, Egger J, *et al* (2024) CellViT: Vision Transformers for precise cell segmentation and classification. *Med Image Anal* 94: 103143
- Huang K, Li M, Li Q, Chen Z, Zhang Y & Gu Z (2024) Image-based profiling and deep learning reveal morphological heterogeneity of colorectal cancer organoids. *Comput Biol Med* 173: 108322
- Hu B, Canon S, Eloë-Fadrosh EA, Anubhav, Babinski M, Corilo Y, Davenport K, Duncan WD, Fagnan K, Flynn M, *et al* (2021) Challenges in bioinformatics workflows for processing microbiome omics data at scale. *Front Bioinform* 1: 826370
- Hu H, Sanghi S & Quon G (2025) Predicting emergent phenotypes from single cell populations using CELLECTION. *bioRxiv*
- Image Feature Variability Results For Five Open-Source Feature Extraction Libraries
- Israel U, Marks M, Dilip R, Li Q, Schwartz M, Pradhan E, Pao E, Li S, Pearson-Goulart A, Perona P, *et al* (2023) A foundation model for cell segmentation. *arXiv [q-bioQM]*
- Jacques M-A, Dobrzyński M, Gagliardi PA, Sznitman R & Pertz O (2021) CODEX, a neural network approach to explore signaling dynamics landscapes. *Mol Syst Biol* 17: e10026
- Jeremiase B, van Ineveld RL, Bok V, Kleinnijenhuis M, de Blank S, Alieva M, Johnson HR, van Vliet EJ, Zeeman AL, Wellens LM, *et al* (2024) A multispectral 3D live organoid imaging platform to screen probes for fluorescence guided surgery. *EMBO Mol Med* 16: 1495–1514
- Ji Y, Tejada-Lapuerta A, Schmacke NA, Zheng Z, Zhang X, Khan S, Rothenaigner I, Tschuck J, Hadian K & Theis FJ (2024) Scalable and universal prediction of cellular phenotypes. *bioRxiv*
- Kalinin AA, Arevalo J, Serrano E, Vulliard L, Tsang H, Bornholdt M, Muñoz AF, Sivagurunathan S, Rajwa B, Carpenter AE, *et al* (2025) A versatile information retrieval framework for evaluating profile strength and similarity. *bioRxiv*: 2024.04.01.587631
- Kenyon-Dean K, Wang ZJ, Urbanik J, Donhauser K, Hartford J, Saberian S, Sahin N, Bendidi I, Celik S, Fay M, *et al* (2024) ViTally Consistent: Scaling Biological Representation Learning for Cell Microscopy.
- Kim V, Adaloglou N, Osterland M, Morelli FM, Halawa M, König T, Gnut D & Marin Zapata PA (2025) Self-supervision advances morphological profiling by unlocking powerful image representations. *Scientific Reports* 15: 1–15
- Kok RNU, Spoelstra WK, Betjes MA, van Zon JS & Tans SJ (2025) Label-free cell imaging and tracking in 3D organoids. *Cell Rep Phys Sci* 6: 102522
- Kraus O, Comitani F, Urbanik J, Kenyon-Dean K, Arumugam L, Saberian S, Wognum C, Celik S & Haque IS (2025) RxRx3-core: Benchmarking drug-target interactions in High-Content Microscopy. *arXiv [q-bioQM]*
- Kraus O, Kenyon-Dean K, Saberian S, Fallah M, McLean P, Leung J, Sharma V, Khan A, Balakrishnan J, Celik S, *et al* (2024) Masked autoencoders for microscopy are scalable learners of cellular biology. *arXiv [csCV]*

- Lafarge MW, Caicedo JC, Carpenter AE, Pluim JPW, Singh S & Veta M (2019) Capturing Single-Cell Phenotypic Variation via Unsupervised Representation Learning. *Proc Mach Learn Res* 103: 315–325
- Lamiable A, Champetier T, Leonardi F, Cohen E, Sommer P, Hardy D, Argy N, Massougbdji A, Del Nery E, Cottrell G, *et al* (2023) Revealing invisible cell phenotypes with conditional generative modeling. *Nat Commun* 14: 6386
- Le T & Lundberg E (2024) High-resolution in silico painting with generative models. *bioRxiv*
- Li B, Zhang B, Zhang C, Zhou M, Huang W, Wang S, Wang Q, Li M, Zhang Y & Song Q (2025) PhenoProfiler: Advancing Phenotypic Learning for Image-based Drug Discovery.
- Li C, Rai MR, Ghashghaei HT & Greenbaum A (2022) Illumination angle correction during image acquisition in light-sheet fluorescence microscopy using deep learning. *Biomed Opt Express* 13: 888–901
- Longo L, Lapuschkin S & Seifert C (2024) Explainable Artificial Intelligence: Second World Conference, xAI 2024, Valletta, Malta, July 17–19, 2024, Proceedings, Part II Springer Nature
- Lukonin I, Zinner M & Liberali P (2021) Organoids in image-based phenotypic chemical screens. *Exp Mol Med* 53: 1495–1502
- Lundberg S & Lee S-I (2017) A unified approach to interpreting model predictions. *arXiv [csAI]*
- Madan A, Saini R, Dhiman N, Juan S-H & Satapathy MK (2025) Organoids as next-generation models for tumor heterogeneity, personalized therapy, and cancer research: Advancements, applications, and future directions. *Organoids* 4: 23
- Maška M, Ulman V, Delgado-Rodriguez P, Gómez-de-Mariscal E, Nečasová T, Guerrero Peña FA, Ren TI, Meyerowitz EM, Scherr T, Löffler K, *et al* (2023) The Cell Tracking Challenge: 10 years of objective benchmarking. *Nat Methods* 20: 1010–1020
- Moraes-Lacerda T, Rodrigues-Da-Silva M, Singh S & De Jesus MB (2025) Image-Based Profiling in Live Cells Using Live Cell Painting. *Bio Protoc* 15: e5464
- Morelli FM, Kim V, Hecker F, Geibel S & Marín Zapata PA (2025) uniDINO: Assay-independent feature extraction for fluorescence microscopy images. *Comput Struct Biotechnol J* 27: 928–936
- Moshkov N, Bornholdt M, Benoit S, Smith M, McQuin C, Goodman A, Senft RA, Han Y, Babadi M, Horvath P, *et al* (2024) Learning representations for image-based profiling of perturbations. *Nature Communications* 15: 1–17
- Murthy RS, Stassen SV, Siu DMD, Lo MCK, Yip GGK & Tsia KK (2024) Generalizable morphological profiling of cells by interpretable unsupervised learning. *bioRxiv*
- Navidi Z, Ma J, Miglietta EA, Liu L, Carpenter AE, Cimini BA, Haibe-Kains B & Wang B (2024) MorphoDiff: Cellular morphology painting with diffusion models. *bioRxiv*
- O'Connor OM & Dunlop MJ (2025) Cell-TRACTR: A transformer-based model for end-to-end segmentation and tracking of cells. *PLoS Comput Biol* 21: e1013071

- Ong HT, Karatas E, Poquillon T, Greci G, Furlan A, Dilasser F, Mohamad Raffi SB, Blanc D, Drimaracci E, Mikec D, *et al* (2025) Digitalized organoids: integrated pipeline for high-speed 3D analysis of organoid structures using multilevel segmentation and cellular topology. *Nat Methods*: 1–12
- O'Shea K & Nash R (2015) An Introduction to Convolutional Neural Networks. *arXiv [csNE]*
- Palma A, Theis FJ & Lottfollahi M (2025) Predicting cell morphological responses to perturbations using generative modeling. *Nat Commun* 16: 505
- Pawlowski N, Caicedo JC, Singh S, Carpenter AE & Storkey A (2016) Automating Morphological Profiling with Generic Deep Convolutional Networks. *Bioinformatics*
- Pearson YE, Kremb S, Butterfoss GL, Xie X, Fahs H & Gunsalus KC (2022) A statistical framework for high-content phenotypic profiling using cellular feature distributions. *Commun Biol* 5: 1409
- Pfaendler R, Hanimann J, Lee S & Snijder B (2023) Self-supervised vision transformers accurately decode cellular state heterogeneity. *Systems Biology*
- Pratapa A, Doron M & Caicedo JC (2021) Image-based cell phenotyping with deep learning. *Curr Opin Chem Biol* 65: 9–17
- Qiu M, Zhou B, Lo F, Cook S, Chyba J, Quackenbush D, Matzen J, Li Z, Mak PA, Chen K, *et al* (2020) A cell-level quality control workflow for high-throughput image analysis. *BMC Bioinformatics* 21: 280
- Rai MR, Li C & Greenbaum A (2022) Quantitative analysis of illumination and detection corrections in adaptive light sheet fluorescence microscopy. *Biomed Opt Express* 13: 2960–2974
- Ramm S, Vary R, Gulati T, Luu J, Cowley KJ, Janes MS, Radio N & Simpson KJ (2022) High-throughput live and fixed cell imaging method to screen Matrigel-embedded organoids. *Organoids* 2: 1–19
- Ringers C, Holmberg D, Flobak Å, Georgiev P, Jarvius M, Johansson M, Larsson A, Rosen D, Seashore-Ludlow B, Visnes T, *et al* (2025) High-content morphological profiling by Cell Painting in 3D spheroids. *bioRxiv*
- Rohban MH, Abbasi HS, Singh S & Carpenter AE (2019) Capturing single-cell heterogeneity via data fusion improves image-based profiling. *Nat Commun* 10: 2082
- Ronneberger O, Fischer P & Brox T (2015) U-Net: Convolutional Networks for Biomedical Image Segmentation. *Medical Image Computing and Computer-Assisted Intervention – MICCAI 2015*: 234–241
- Rotem O, Schwartz T, Maor R, Tauber Y, Shapiro MT, Meseguer M, Gilboa D, Seidman DS & Zaritsky A (2024) Visual interpretability of image-based classification models by generative latent space disentanglement applied to in vitro fertilization. *Nat Commun* 15: 7390
- Rotem O & Zaritsky A (2024) Visual interpretability of bioimaging deep learning models. *Nat Methods* 21: 1394–1397

- Sanchez-Fernandez A, Rumetshofer E, Hochreiter S & Klambauer G (2023) CLOOME: contrastive learning unlocks bioimaging databases for queries with chemical structures. *Nat Commun* 14: 7339
- Scheeder C, Heigwer F & Boutros M (2018) Machine learning and image-based profiling in drug discovery. *Current Opinion in Systems Biology* 10: 43
- Schindelin J, Arganda-Carreras I, Frise E, Kaynig V, Longair M, Pietzsch T, Preibisch S, Rueden C, Saalfeld S, Schmid B, *et al* (2012) Fiji: an open-source platform for biological-image analysis. *Nat Methods* 9: 676–682
- Schmidt U, Weigert M, Broaddus C & Myers G (2018) Cell detection with star-convex polygons. In *Medical Image Computing and Computer Assisted Intervention – MICCAI 2018* pp 265–273. Cham: Springer International Publishing
- Schneidewind T, Kapoor S, Garivet G, Karageorgis G, Narayan R, Vendrell-Navarro G, Antonchick AP, Ziegler S & Waldmann H (2019) The pseudo natural product Myokinasib is a myosin light chain kinase 1 inhibitor with unprecedented chemotype. *Cell Chem Biol* 26: 512–523.e5
- Schwartz MS, Moen E, Miller G, Dougherty T, Borba E, Ding R, Graf W, Pao E & Van Valen D (2019) Caliban: Accurate cell tracking and lineage construction in live-cell imaging experiments with deep learning. *bioRxiv*
- Seal S, Carreras-Puigvert J, Singh S, Carpenter AE, Spjuth O & Bender A (2024a) From pixels to phenotypes: Integrating image-based profiling with cell health data as BioMorph features improves interpretability. *Molecular Biology of the Cell*
- Seal S, Trapotsi M-A, Spjuth O, Singh S, Carreras-Puigvert J, Greene N, Bender A & Carpenter AE (2024b) Cell Painting: a decade of discovery and innovation in cellular imaging. *Nature Methods* 22: 254–268
- Serrano E, Chandrasekaran SN, Buntun D, Brewer KI, Tomkinson J, Kern R, Bornholdt M, Fleming SJ, Pei R, Arevalo J, *et al* (2025) Reproducible image-based profiling with Pycytominer. *Nat Methods* 22: 677–680
- Shpigler A, Kolet N, Golan S, Weisbart E & Zaritsky A (2024) Anomaly detection for high-content image-based phenotypic cell profiling. *bioRxiv*
- Škuta C, Müller T, Voršilák M, Popr M, Epp T, Skopelitou KE, Rossella F, Stechmann B, Gribbon P & Bartůněk P (2025) ECBD: European chemical biology database. *Nucleic Acids Res* 53: D1383–D1392
- Steiner A, Kolesnikov A, Zhai X, Wightman R, Uszkoreit J & Beyer L (2021) How to train your ViT? Data, Augmentation, and Regularization in Vision Transformers. *arXiv [csCV]*
- Stirling DR, Swain-Bowden MJ, Lucas AM, Carpenter AE, Cimini BA & Goodman A (2021) CellProfiler 4: improvements in speed, utility and usability. *BMC Bioinformatics* 22: 1–11
- Stossi F, Singh PK, Marini M, Safari K, Szafran AT, Rivera TA, Candler CD, Mancini MG, Mosa EA, Bolt MJ, *et al* (2024) SPACE: an open-source, single-cell analysis of Cell Painting data. *Nature communications* 15

- Stringer C, Wang T, Michaelos M & Pachitariu M (2020) Cellpose: a generalist algorithm for cellular segmentation. *Nature Methods* 18: 100–106
- Subramanian A, Alperovich M, Yang Y & Li B (2022) Biology-inspired data-driven quality control for scientific discovery in single-cell transcriptomics. *Genome Biol* 23: 267
- Sypetkowski M, Rezanejad M, Saberian S, Kraus O, Urbanik J, Taylor J, Mabey B, Victors M, Yosinski J, Sereshkeh AR, *et al* (2023) RxRx1: A dataset for evaluating experimental batch correction methods. In *2023 IEEE/CVF Conference on Computer Vision and Pattern Recognition Workshops (CVPRW)* pp 4285–4294. IEEE
- Tang Q, Ratnayake R, Seabra G, Jiang Z, Fang R, Cui L, Ding Y, Kahveci T, Bian J, Li C, *et al* (2024) Morphological profiling for drug discovery in the era of deep learning. *Brief Bioinform* 25: bbae284
- Tian G, Harrison PJ, Sreenivasan AP, Carreras-Puigvert J & Spjuth O (2023) Combining molecular and cell painting image data for mechanism of action prediction. *Artif Intell Life Sci*: 100060
- Tonks S, Hsu C-Y, Hood S, Musso R, Hopely C, Titus S, Krull A, Doan M & Styles I (2023) Evaluating virtual staining for high-throughput screening. In *2023 IEEE 20th International Symposium on Biomedical Imaging (ISBI)* pp 1–5. IEEE
- Vaswani A, Shazeer N, Parmar N, Uszkoreit J, Jones L, Gomez AN, Kaiser L & Polosukhin I (2017) Attention Is All You Need.
- Viana MP, Chen J, Knijnenburg TA, Vasan R, Yan C, Arakaki JE, Bailey M, Berry B, Borensztein A, Brown EM, *et al* (2023) Integrated intracellular organization and its variations in human iPS cells. *Nature* 613: 345–354
- Vulliard L, Hancock J, Kamnev A, Fell CW, da Silva J F, Loizou JI, Nagy V, Dupré L & Menche J (2022) BioProfiling.jl: profiling biological perturbations with high-content imaging in single cells and heterogeneous populations. *Bioinformatics (Oxford, England)* 38
- Wagner J, Warden H, Khamseh A & Beentjes SV (2025) scmorph: a Python package for analysing single-cell morphological profiles. *The Journal of Open Source Software (JOSS)*
- Wang J, Wang X, Zhang P, Xie S, Fu S, Li Y & Han H (2021) Correction of uneven illumination in color microscopic image based on fully convolutional network. *Opt Express* 29: 28503–28520
- Wang S, Liu X, Li Y, Sun X, Li Q, She Y, Xu Y, Huang X, Lin R, Kang D, *et al* (2023) A deep learning-based stripe self-correction method for stitched microscopic images. *Nat Commun* 14: 5393
- Wang Y, Zhao J, Xu H, Han C, Tao Z, Zhou D, Geng T, Liu D & Ji Z (2024) A systematic evaluation of computational methods for cell segmentation. *Brief Bioinform* 25
- Way GP, Sailem H, Shave S, Kasprowicz R & Carragher NO (2023) Evolution and impact of high content imaging. *SLAS Discov* 28: 292–305
- Weigert M & Schmidt U (2022) Nuclei Instance Segmentation and Classification in Histopathology Images with Stardist. In *2022 IEEE International Symposium on Biomedical*

Imaging Challenges (ISBIC) IEEE

- Weisbart E, Kumar A, Arevalo J, Carpenter AE, Cimini BA & Singh S (2024) Cell Painting Gallery: an open resource for image-based profiling. *Nature Methods* 21: 1775–1777
- Wilkinson, Dumontier M, Aalbersberg IJ, Appleton G, Axton M, Baak A, Blomberg N, Boiten JW, da Silva Santos LB, Bourne PE, *et al* (2016) The FAIR Guiding Principles for scientific data management and stewardship. *Scientific data* 3
- Williams E, Moore J, Li SW, Rustici G, Tarkowska A, Chessel A, Leo S, Antal B, Ferguson RK, Sarkans U, *et al* (2017) The Image Data Resource: A bioimage data integration and publication platform. *Nat Methods* 14: 775–781
- Wong DR, Logan DJ, Hariharan S, Stanton R, Clevert D-A & Kiruluta A (2023) Deep representation learning determines drug mechanism of action from cell painting images. *Digit Discov* 2: 1354–1367
- Xing X, Murdoch S, Tang C, Papanastasiou G, Cross-Zamirski J, Guo Y, Xiao X, Schönlieb C-B, Wang Y & Yang G (2024) Can generative AI replace immunofluorescent staining processes? A comparison study of synthetically generated cellpainting images from brightfield. *Comput Biol Med* 182: 109102
- Xun D, Wang R, Zhang X & Wang Y (2024) Microsnoop: A generalist tool for microscopy image representation. *Innovation (Cambridge (Mass))* 5
- Yan C, Zhang Y, Feng J, Hua H, Ruan Z, Li Z, Li S, Yan C, Li P, Liu J, *et al* (2025) Triple-effect correction for Cell Painting data with contrastive and domain-adversarial learning. *Nat Commun* 16: 6886
- Yao H, Hanslovsky P, Huetter J-C, Hoeckendorf B & Richmond D (2024) Weakly supervised set-consistency learning improves morphological profiling of single-cell images. *arXiv [csCV]*
- Yoshida SR, Maity BK & Chong S (2023) Visualizing protein localizations in fixed cells: Caveats and the underlying mechanisms. *J Phys Chem B* 127: 4165–4173
- Zaritsky A, Jamieson AR, Welf ES, Nevarez A, Cillay J, Eskiocak U, Cantarel BL & Danuser G (2021) Interpretable deep learning uncovers cellular properties in label-free live cell images that are predictive of highly metastatic melanoma. *Cell Syst* 12: 733–747.e6
- Zhou Y, Sollmann J & Chen J (2024) Deep-learning-based image compression for microscopy images: An empirical study. *Biol Imaging* 4: e16

12th Dec 2025

Manuscript Number: MSB-2024-12819R

Title: Progress and new challenges in image-based profiling

Author: Erik Serrano

John Peters

Jesko Wagner

Rebecca Graham

Zhengkao Chen

Brian Feng

Gisele Miranda

Alexandr Kalinin

Loan Vulliard

Jenna Tomkinson

Cameron Mattson

Michael Lippincott

Ziqi Kang

Divya Sitani

Dave Bunten

Srijit Seal

Neil Carragher

Anne Carpenter

Shantanu Singh

Paula Marin Zapata

Juan Caicedo

Gregory Way

Dear Greg,

Thank you for sending us your revised manuscript. We have now heard back from the three reviewers who were asked to re-evaluate your study. As you will see, the reviewers are overall satisfied with the modifications made. Before we can formally accept your manuscript, we would ask you to address the following minor issues:

1. Reviewer #1 has raised a new point regarding the discussion of technical issues in cell tracking. Although we normally do not allow new concerns to be introduced after a round of major revision, we consider this comment addressable. We therefore ask that you provide a response and make reasonable revisions to address it.

On a more editorial level:

1. Please ensure that the funding information in the manuscript text matches the details entered in the submission system. The following funding information is missing in the submission system and need to be added: NIH under award number 5T15LM007359; the Medical Research Council (MC_ST_00035), (MR/R015635/1), the National Science Foundation (Grant No.2348683); an advised fund of Silicon Valley Community Foundation (funder DOI 10.13039/100014989).

2. Please provide a new (preferably institutional) email address for Alexandr A. Kalinin. We currently receive the following automatic message:

"As of October 1, I am no longer at the Broad Institute. If this is a personal message you can reach me at alxndrkalinin [at] gmail.com. If this is business concerning the Imaging Platform, please write to imagingadmin@broadinstitute.org".

3. The tables and EV tables are not organized correctly. Please choose one of the following options:

a. Move the current Tables 1-3 into the manuscript file, placing them after the "Legends" section. The legends for these tables may remain in the manuscript.

Because the current Table 4 and Table EV2 are relatively complex, they should be reclassified as EV datasets using the nomenclature Dataset EV1-EV2. The source file names, titles, legends, and manuscript callouts should be updated accordingly. Include the legends as a separate tab/sheet within each Excel file. Remove the legends for EV tables and EV datasets from the manuscript file.

b. Rename Tables 1, 2, 3, and Table EV1 as EV tables using the nomenclature Table EV1-EV4. Update the source file names, titles, legends, and manuscript callouts accordingly. Include the legends above each table within its respective Excel file. Because Table 4 and Table EV2 are complex, they should be reclassified as EV datasets using the nomenclature Dataset EV1-

EV2. Update the source file names, titles, legends, and manuscript callouts accordingly, and include the legends as a separate tab/sheet in each Excel file. Remove the legends for all of these EV tables/datasets from the manuscript text.

I have forwarded your figures to our designer, who will contact you directly regarding the redrawn images once they are ready.

Use the following link to submit your revised paper:

Thank you for submitting this interesting paper to Molecular Systems Biology.

Kind regards,
Jingyi

Jingyi Hou, PhD
Senior Editor
Molecular Systems Biology

*** PLEASE NOTE *** As part of the EMBO Press transparent editorial process initiative (see our Editorial at <https://dx.doi.org/10.1038/msb.2010.72> , Molecular Systems Biology will publish online a Review Process File to accompany accepted manuscripts. When preparing your letter of response, please be aware that in the event of acceptance, your cover letter/point-by-point document will be included as part of this File, which will be available to the scientific community. More information about this initiative is available in our Instructions to Authors. If you have any questions about this initiative, please contact the editorial office (msb@embo.org).

Reviewer #1:

I would like to thank the authors for their comprehensive response and the thoughtful discussion. I also want to express my gratitude for the careful and extensive revisions made to the manuscript. I have reviewed both the response and the revised version of the paper. The authors have fully addressed all of the key points I raised previously.

In my review of the current version of the manuscript, I have identified a further point I unfortunately have overlooked in my first revision. For this, I want to apologize to the authors. However, I believe it is an important point I would like to bring to the discussion with the authors.

The authors raise: "Most single-cell tracking tools have a graphical user interface (GUI), which is helpful, but lacks scalability for HTS, making these tools virtually unusable at large scale...". They mention this as a major limitation of tracking software such as TrackMate. This sentence is confusing, as it makes it seem that the presence of a GUI limits the scalability of cell tracking analysis. Written like this does not seem to accurately represent the development efforts going towards software such as TrackMate, and on the other hand, misses for me the core technical challenge that underlies the cell tracking problem.

Towards the first point, the developers of TrackMate spent a lot of effort to make it modular in the latest version, and in the last decade, the developers in the Fiji/ImageJ community have spent a lot of time untangling the functions from the GUI. In fact, this was one driver of the ImageJ2 project. Although I currently do not employ cell tracking in my work, I have a hard time believing TrackMate or its functions cannot be easily run headlessly (<https://imagej.net/plugins/trackmate/scripting/scripting>) or that the technical dependence on a GUI would hamper users in general from performing cell-tracking at scale.

In my viewpoint, the main reason why tools such as TrackMate are in widespread use is that cell tracking is still a very hard problem to fully automate. Currently, there seems to be no general tracking workflow that works for the majority of tasks; they all need to be finely tuned from segmentation/detection to particle-linking. TrackMate specifically addresses this with its modularity to be on a software engineering level agnostic to segmentation/detection and particle-linking, and the GUI, as well as the visualization tools, facilitate the fine tuning of workflow components.

Then, even if low tracking error rates are achieved, these still compound into useless tracks over time. A particular issue with cell tracking is also that cells typically look very similar to each other, thus correcting automatically based on cell features, for instance, is hard (contrasting this with tracking in natural images e.g., people - different clothes, faces, appearance, etc). Thus, the GUI is indispensable for many cell tracking problems as it is the prerequisite for manual curation. Arguably, this is a big issue

with the developmental biology, from where my tracking experience comes from, as this community is mainly interested in accurate tracking of cell lineages within developing tissues. However, the same issues will apply to the profiling field. Thus, the presence of the GUI, visualization, and analysis of software such as TrackMate makes tracking feasible for many users in the first place. Which would mean for me that the difficulty with fine-tuning workflows to reduce tracking errors to acceptable levels and the need for manual curation via GUIs is making cell-tracking hard to scale.

I am sorry again to raise this in the second round, but I believe this is critical to define the core technical issue of cell tracking. Creating software that can run headlessly seems to me comparatively trivial.

Once again, I would like to emphasize the high quality of the manuscript, even prior to revision, the timeliness, and the importance of this work. I consider this review an essential contribution to the field.

I discovered a minor error that can be easily corrected:

It is "European Chemical Biology Database (Skuta et al, 2025)" not "European Chemical Biology (Skuta et al, 2025)"

Reviewer #2:

Dear Editorial Team,

My suggestions have been adequately addressed.

Kind regards

Reviewer #3:

The authors have addressed my concerns. In my opinion, the manuscript is ready for publication and will be a nice contribution to the field.

Assaf Zaritsky

Ben-Gurion University of the Negev, Israel

Department of Biomedical Informatics
1890 N Revere Ct.
Aurora, CO 80045

Dear Jingyi,

Thank you again for facilitating the peer review of our review article "*Progress and new challenges in image-based profiling*" (MSB-2024-12819R). We have received this second round of comments, and we have addressed all concerns (including editorial comments).

We resubmit following expectations and guidelines of the journal, but please let us know if there are any other required changes.

Below, we respond to each reviewer's comments in **blue** and add details of what we changed in **orange**. We detail the specific changes to the manuscript text in **black**. We also include an edited manuscript with tracked changes.

Sincerely,

Gregory P. Way, PhD
Assistant Professor
Department of Biomedical Informatics
University of Colorado Anschutz

Editorial Comments

On a more editorial level:

1. Please ensure that the funding information in the manuscript text matches the details entered in the submission system. The following funding information is missing in the submission system and need to be added: NIH under award number 5T15LM007359; the Medical Research Council (MC_ST_00035), (MR/R015635/1), the National Science Foundation (Grant No.2348683); an advised fund of Silicon Valley Community Foundation (funder DOI 10.13039/100014989).

We regret to have omitted this information, and we thank the editor for bringing this to our attention. We have now added this to the manuscript info in the submission.

2. Please provide a new (preferably institutional) email address for Alexandr A. Kalinin. We currently receive the following automatic message:
"As of October 1, I am no longer at the Broad Institute. If this is a personal message you can reach me at [alxndrkalinin \[at\] gmail.com](mailto:alxndrkalinin@gmail.com). If this is business concerning the Imaging Platform, please write to imagingadmin@broadinstitute.org".

We have updated the author's email to alex.kalinin@biohub.org.

3. The tables and EV tables are not organized correctly. Please choose one of the following options:

- a) Move the current Tables 1-3 into the manuscript file, placing them after the "Legends" section. The legends for these tables may remain in the manuscript.

Because the current Table 4 and Table EV2 are relatively complex, they should be reclassified as EV datasets using the nomenclature Dataset EV1-EV2. The source file names, titles, legends, and manuscript callouts should be updated accordingly. Include the legends as a separate tab/sheet within each Excel file. Remove the legends for EV tables and EV datasets from the manuscript file.

We have opted for this strategy. Specifically, we updated Table 4 to "Dataset EV2" and Table EV2 to "Dataset EV1". In addition to these file changes, we have added a new sheet to each EV table/dataset containing the table name and caption. We have also updated the titles of each xlsx:

From	To
Table4-Common-metrics-used	Dataset-EV2-Summary of evaluation metrics used in imaging-based profiling
EV1-Batch-correction-methods	EV1-Batch correction methods used in image-based profiling
EV2-Large-public-datasets-for-image-based-profiling	Dataset-EV1-Publicly-available datasets for image-based profiling of human cells

- b) . Rename Tables 1, 2, 3, and Table EV1 as EV tables using the nomenclature Table EV1-EV4. Update the source file names, titles, legends, and manuscript callouts accordingly. Include the legends above each table within its respective Excel file.

Because Table 4 and Table EV2 are complex, they should be reclassified as EV datasets using the nomenclature Dataset EV1-EV2. Update the source file names, titles, legends, and manuscript callouts accordingly, and include the legends as a separate tab/sheet in each Excel file. Remove the legends for all of these EV tables/datasets from the manuscript text.

Reviewer comments:

Reviewer #1:

I would like to thank the authors for their comprehensive response and the thoughtful discussion. I also want to express my gratitude for the careful and extensive revisions made to

the manuscript. I have reviewed both the response and the revised version of the paper. The authors have fully addressed all of the key points I raised previously.

In my review of the current version of the manuscript, I have identified a further point I unfortunately have overlooked in my first revision. For this, I want to apologize to the authors. However, I believe it is an important point I would like to bring to the discussion with the authors.

The authors raise: "Most single-cell tracking tools have a graphical user interface (GUI), which is helpful, but lacks scalability for HTS, making these tools virtually unusable at large scale...". They mention this as a major limitation of tracking software such as TrackMate. This sentence is confusing, as it makes it seem that the presence of a GUI limits the scalability of cell tracking analysis. Written like this does not seem to accurately represent the development efforts going towards software such as TrackMate, and on the other hand, misses for me the core technical challenge that underlies the cell tracking problem.

Towards the first point, the developers of TrackMate spent a lot of effort to make it modular in the latest version, and in the last decade, the developers in the Fiji/ImageJ community have spent a lot of time untangling the functions from the GUI. In fact, this was one driver of the ImageJ2 project. Although I currently do not employ cell tracking in my work, I have a hard time believing TrackMate or its functions cannot be easily run headlessly (<https://imagej.net/plugins/trackmate/scripting/scripting>) or that the technical dependence on a GUI would hamper users in general from performing cell-tracking at scale.

In my viewpoint, the main reason why tools such as TrackMate are in widespread use is that cell tracking is still a very hard problem to fully automate. Currently, there seems to be no general tracking workflow that works for the majority of tasks; they all need to be finely tuned from segmentation/detection to particle-linking. TrackMate specifically addresses this with its modularity to be on a software engineering level agnostic to segmentation/detection and particle-linking, and the GUI, as well as the visualization tools, facilitate the fine tuning of workflow components.

Then, even if low tracking error rates are achieved, these still compound into useless tracks over time. A particular issue with cell tracking is also that cells typically look very similar to each other, thus correcting automatically based on cell features, for instance, is hard (contrasting this with tracking in natural images e.g., people - different clothes, faces, appearance, etc). Thus, the GUI is indispensable for many cell tracking problems as it is the prerequisite for manual curation. Arguably, this is a big issue with the developmental biology, from where my tracking experience comes from, as this community is mainly interested in accurate tracking of cell lineages within developing tissues. However, the same issues will apply to the profiling field. Thus, the presence of the GUI, visualization, and analysis of software such as TrackMate makes tracking feasible for many users in the first place. Which would mean for me that the difficulty with fine-tuning workflows to reduce tracking errors to acceptable levels and the need for manual curation via GUIs is making cell-tracking hard to scale.

I am sorry again to raise this in the second round, but I believe this is critical to define the core technical issue of cell tracking. Creating software that can run headlessly seems to me comparatively trivial.

Once again, I would like to emphasize the high quality of the manuscript, even prior to revision, the timeliness, and the importance of this work. I consider this review an essential contribution to the field.

We thank the reviewer for raising this point. We agree that characterizing the GUI as the primary limitation was inaccurate, given the advanced headless scripting capabilities of modern tools like TrackMate.

We have revised the text:

“While widely used single-cell tracking tools, like TrackMate, offer scripting capabilities for automated 'headless' execution, achieving sufficiently low error rates across large-scale datasets remains a major technical hurdle (Cayuela López et al. 2023). In smaller experiments, tracking errors (e.g., identity swaps or lost tracks) are typically corrected through manual curation using the software's graphical user interface (GUI). However, in high-throughput screening, where datasets contain billions of cells, such manual intervention is impossible. Consequently, the compounding nature of tracking errors remains the primary barrier to scalability. Some limitations of single-cell tracking can be addressed through thoughtful experimental design, for instance, using shorter time intervals to better track highly motile cells (Yang et al. 2020)”

I discovered a minor error that can be easily corrected:

It is "European Chemical Biology Database (Skuta et al, 2025)" not "European Chemical Biology (Skuta et al, 2025)"

We thank the reviewer for pointing out this oversight on our part. We have corrected the name.

Reviewer #2:

Dear Editorial Team,

My suggestions have been adequately addressed.

Kind regards

Reviewer #3:

The authors have addressed my concerns. In my opinion, the manuscript is ready for publication and will be a nice contribution to the field.

16th Dec 2025

Manuscript number: MSB-2024-12819RR

Title: Progress and new challenges in image-based profiling

Dear Greg,

Thank you again for submitting your revised manuscript. I am pleased to inform you that your paper has been accepted for publication.

I have forwarded your figures to our designer, who will contact you directly regarding the redrawn images once they are ready.

After you approve the redrawn images, your manuscript will be exported to our production team to begin publication processing. It will undergo copy editing, and you will receive page proofs prior to publication.

Please note that you will be contacted by Springer Nature Author Services to complete licensing and payment information. When prompted by the Author Services system, please use the following token for payment: Token MTG5NJGWMJQXNW.

Should you be planning a Press Release on your article, please get in contact with embo_production@springernature.com after the manuscript is exported, in order to coordinate publication and release dates.

Thank you for your valuable contribution to Molecular Systems Biology!

Kind regards,
Jingyi

Jingyi Hou, PhD
Senior Editor
Molecular Systems Biology

>>> Please note that it is Molecular Systems Biology policy for the transcript of the editorial process (containing referee reports and your response letter) to be published as an online supplement to each paper. If you do NOT want this, you will need to inform the Editorial Office via email immediately. More information is available here: <https://link.springer.com/partners/embo-press/editorial-policies#Peer%20review>